# A dynamic partitioning mechanism polarizes membrane protein distribution

Tatsat Banerjee [1,2] ✉, Satomi Matsuoka [3,4], Debojyoti Biswas [5], Yuchuan Miao [1,6], Dhiman Sankar Pal [1], Yoichiro Kamimura[3], Masahiro Ueda [3,4], Peter N. Devreotes [1,6] ✉ & Pablo A. Iglesias [1,5] ✉

The plasma membrane is widely regarded as the hub of the numerous signal transduction activities. Yet, the fundamental biophysical mechanisms that spatiotemporally compartmentalize different classes of membrane proteins remain unclear. Using multimodal live-cell imaging, here we first show that several lipid-anchored membrane proteins are consistently depleted from the membrane regions where the Ras/PI3K/Akt/F-actin network is activated. The dynamic polarization of these proteins does not depend upon the F-actin-based cytoskeletal structures, recurring shuttling between membrane and cytosol, or directed vesicular trafficking. Photoconversion microscopy and single-molecule measurements demonstrate that these lipid-anchored molecules have substantially dissimilar diffusion profiles in different regions of the membrane which enable their selective segregation. When these diffusion coefficients are incorporated into an excitable network-based stochastic reaction-diffusion model, simulations reveal that the altered affinity mediated selective partitioning is sufficient to drive familiar propagating wave patterns. Furthermore, normally uniform integral and lipid-anchored membrane proteins partition successfully when membrane domain-specific peptides are optogenetically recruited to them. We propose "dynamic partitioning" as a new mechanism that can account for large-scale compartmentalization of a wide array of lipid-anchored and integral membrane proteins during various physiological processes where membrane polarizes.

Numerous signal transduction and cytoskeletal molecules spatially and temporally self-organize into distinct regions on the plasma membrane to establish polarity which regulates cell morphology and migration mode[1,2]. The asymmetric localization and activation of these biomolecules are necessary for proper physiological responses[3–5]. For example, when a migrating cell experiences an external cue, receptors trigger G-protein activation which in turn initiates a signaling cascade such as the activation of Ras/Rap, PI3K, Akt, Rac, and Cdc42. These collectively result in the activation of cytoskeletal network activities mediated by Scar/WAVE, Arp2/3, etc., which leads to actin polymerization and eventual protrusion formation[3,6–9]. All these events take place at the cell's leading edge in "front" regions of the plasma

[1]Department of Cell Biology and Center for Cell Dynamics, School of Medicine, Johns Hopkins University, Baltimore, MD, USA. [2]Department of Chemical and Biomolecular Engineering, Whiting School of Engineering, Johns Hopkins University, Baltimore, MD, USA. [3]Laboratory for Cell Signaling Dynamics, RIKEN Center for Biosystems Dynamics Research, Suita, Osaka, Japan. [4]Laboratory of Single Molecule Biology, Graduate School of Frontier Biosciences, Osaka University, Suita, Osaka, Japan. [5]Department of Electrical and Computer Engineering, Whiting School of Engineering, Johns Hopkins University, Baltimore, MD, USA. [6]Department of Biological Chemistry, School of Medicine, Johns Hopkins University, Baltimore, MD, USA. ✉e-mail: tatsatb@jhu.edu; pnd@jhmi.edu; pi@jhu.edu

membrane[3–7,10]. On the other hand, components that antagonize this activation process, such as PTEN, activated RhoA/ROCK, and myosin II assembly vacate the activated regions and maintain the basal quiescent state or "back"-state of the membrane elsewhere[3,5–8,10–15]. The dynamic "front-" and "back" regions that form in the inner leaflet of the plasma membrane appear as complementary propagating waves on the ventral surface of cells. A similar complementary, asymmetric organization is conserved across phylogeny, in a wide array of physiological and developmental processes, such as random migration, phagocytosis, macropinocytosis, cytokinesis, and apical/basal polarity formation[4,6,16,17].

Multiple different mechanisms have been proposed to explain such symmetry breaking processes that can lead to polarization of plasma membrane and compartmentalization of membrane proteins. First, dynamic cortical patterning has been attributed to "shuttling" or reversible recruitment of peripheral membrane proteins from cytosol to membrane and subsequent spatiotemporally controlled release of such proteins from membrane to cytosol[4,6,11,18–20]. While the shuttling-based mechanism does operate for a variety of proteins involved in protrusion formation and ventral wave propagation, it cannot explain the polarization of integral, lipid-anchored, or otherwise tightly bound membrane proteins since their membrane association and dissociation rates are much slower than the time scale of these dynamic events. Second, various "fence and picket" models of membrane organization, which rely on actin-based cytoskeletal "fences" to compartmentalize the plasma membrane and impede long-range diffusion of proteins, have been suggested to describe the stable polarized distributions of proteins in the membrane[21,22]. However, now it has been repeatedly demonstrated in *Dictyostelium*, neutrophil, and epithelial cells that, either under the influence of external cues or during spontaneous activation, multiple components of the signal transduction network can get activated and display robust dynamic polarization and pattern formation even when cytoskeletal dynamics is abolished[4,23–37]. Third, intracellular sorting by directed vesicular transport has been shown to generate asymmetry of different types of membrane proteins during amoeboid migration of leukocytes and during neuronal polarity formation[38–44]. However, to generate and reorient dynamic asymmetry of so many molecules, as it occurs for the signal transduction cascade, via directed vesicular trafficking, in a repeated fashion, it would require an enormous amount of energy, and again, the sorting and transport process would be expected to require intact cytoskeletal dynamics.

If polarized distributions of membrane proteins were to arise spontaneously and be maintained dynamically within the plasma membrane due to their native biophysical characteristics, many of these inconsistencies would be resolved, but such a mechanism has not been envisioned or investigated. In this study, we first identified multiple proteins, including three key lipid-anchored proteins of the signaling network (the $\beta\gamma$ subunit of heterotrimeric G-protein, a Akt/SGK-related kinase, and a RasGTPase) and two synthetic lipidated peptides, which surprisingly exhibited dynamic symmetry breaking during ventral wave propagation and protrusion formation. We found that these proteins maintained their polarized dynamics even in the absence of cytoskeletal activity. Combining global receptor activation, photoconversion microscopy, optogenetics, and single-molecule imaging with computational simulations, we discovered that lipid-anchored and integral membrane proteins align to polarized compartments simply by differentially diffusing in different domains of the membrane. The affinity alteration-mediated, spatially heterogeneous mobility-based way of compartmentalization is independent of recurrent recruitment/release-based "shuttling", external cytoskeletal barriers, and vesicular trafficking. We term this distinct mechanism "dynamic partitioning" and propose that it can explain the general compartmentalization and polarization phenomena of numerous integral, lipid-anchored, and other tightly

bound associated membrane proteins in various physiological and developmental scenarios.

## Results

To examine the spatiotemporal dynamics of different peripheral, lipid-anchored, and integral membrane proteins of signal transduction and cytoskeleton networks, we visualized protrusion formation during migration and cortical wave propagation on the substrate-attached surface of electrofused giant *Dictyostelium* cells. As previously reported[25,28,31,45–48], we observed a coordinated propagation of waves of F-actin polymerization biosensor, LimE$_{\Delta coil}$ ('LimE') and PI(3,4,5)P3 biosensor PH$_{Crac}$ (Supplementary Fig. 1a). An analogous coordination was clear in the confocal section of the membrane of the migrating cell where both localized to the new protrusions (Supplementary Fig. 1b). It has been established[3,6,9,49] that either in the case of protrusion formation or cortical wave propagation, the inner leaflet of plasma membrane is consistently segregated into two distinct states: a "front" or protrusion state and "back" or basal state. Front-state regions of the membrane are defined by the Ras/PI3K/Akt activation and subsequent actin polymerization, whereas molecules that antagonize their activation such as PTEN/PI(4,5)P2/Myosin-II mark the back-state regions. Supplementary Fig. 1c and Supplementary Fig. 1d demonstrate the complementary spatiotemporal dynamics of PTEN and PIP3 in ventral waves and migrating cell protrusions, respectively. A similar complementary localization was exhibited by another peripheral back protein CynA with respect to PIP3 (Supplementary Fig. 1e, f). To quantitate such dynamic complementarity in localization, throughout this study for these and additional proteins, we have computed Pearson's correlation coefficient (r) with respect to PIP3 (see Methods for details) which acts as a reliable proxy for signaling network activation, i.e. the spatiotemporal zone of the "front" state of the membrane. As evident from the heatmap, standard peripheral back-proteins PTEN (Supplementary Fig. 1g) and CynA (Supplementary Fig. 1h) maintain a high degree of consistent complementarity with respect to PIP3 on the membrane. As discussed earlier, this kind of polarized patterning (Supplementary Fig. 1i) can be attributed to a spatially restricted recruitment of front molecules from cytosol to particular domains of membrane that are transitioning from back to front state (Supplementary Fig. 1j). The opposite sequence of events is thought to drive the switch from front to back state (Supplementary Fig. 1j).

### Different localization of multiple lipid-anchored membrane proteins in front- state and back-state regions

To gain further insight into the dynamic compartmentalization and patterning of different classes of membrane proteins, we first examined the spatiotemporal profiles of multiple fluorescently-tagged lipidated membrane proteins with respect to PIP3 levels during ventral wave propagation and protrusion formation in live *Dictyostelium* cells. First, we imaged Akt/SGK homolog PKBR1 which maintains its membrane association via a N-terminal myristoylation moiety. Surprisingly, PKBR1 was substantially depleted in the front-state regions of the membrane that was enriched in PIP3 ventral waves (Fig. 1a). Line kymographs (Fig. 1b) and videos (Supplementary Movie 1) demonstrated the consistency of complementarity with respect to front state regions. Correspondingly, PKBR1 was depleted from the protrusions in migrating cells (Supplementary Fig. 2a). Pearson's r heatmap for PKBR1 (Fig. 1c) establishes that the localization dynamics of PKBR1 resembles the asymmetric localization of standard back proteins like PTEN and CynA (Supplementary Fig. 1g, h). Second, we recorded the dynamics of the $\beta\gamma$ subunit of heterotrimeric G-Protein which associates with membrane via the prenylation on G$\gamma$. G$\beta\gamma$ was consistently confined to the back-state regions of the membrane during ventral wave propagation (Fig. 1d–f and Supplementary Movie 2) and was localized away from protrusions in migrating cells (Supplementary Fig. 2b). Next, we imaged the membrane profile of RasG, which like

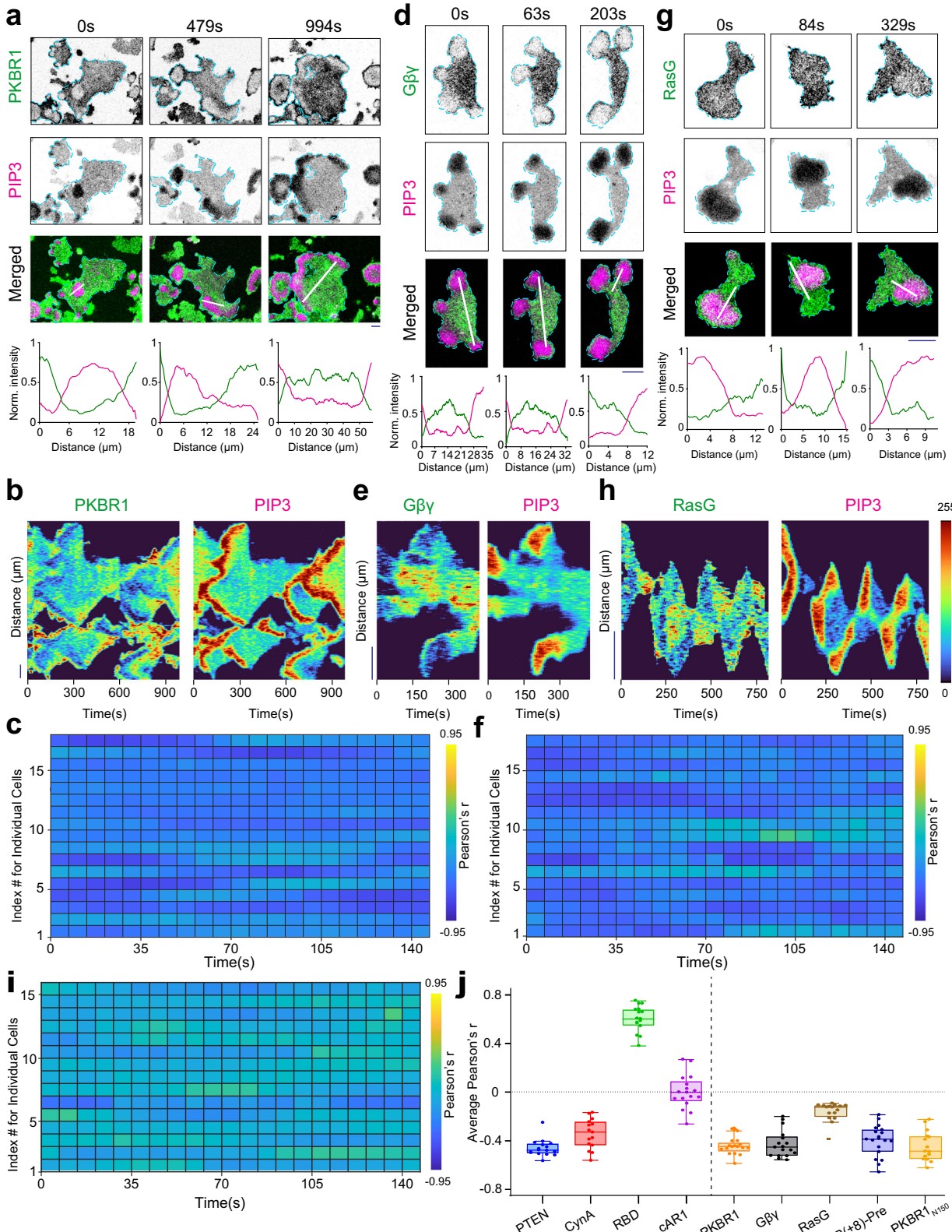

many other small GTPases, maintains its membrane targeting via a prenylation moiety at the C-terminal. We found that RasG maintained a consistent preference towards back-state regions of the membrane (Fig. 1g, h and Supplementary Movie 3) during continuous propagation of ventral wave and protrusion formation (Supplementary Fig. 2c), much like PKBR1 and Gβγ, albeit to a bit lesser degree (Fig. 1i). We next wondered whether these asymmetric distribution profiles are more

generalizable. To this end, we created two membrane-targeting synthetic peptides, one of which is myristoylated and other one is prenylated and recorded their spatiotemporal dynamics over membrane. Consistent with our previous result, the 18 amino acid prenylated peptide R(+8)-Pre that carries +8 positive charge[9,50], displayed dynamic exclusion from the front-state regions of membrane in ventral waves (Supplementary Fig. 2d, e). Another myristoylated peptide consisting

**Fig. 1 | Asymmetric dynamics of multiple lipid-anchored membrane proteins during ventral wave propagation and protrusion formation. a, d, g** Representative live-cell time-lapse images of cortical waves on the ventral surface of a *Dictyostelium* cell co-expressing PIP3 biosensor $PH_{Crac}$-mCherry along with PKBR1-KikGR (**a**), or KikGR-G$\beta$ (**d**), or GFP-RasG (**g**), demonstrating dynamic depletion of PKBR1, G$\beta\gamma$, and RasG from the activated regions of the membrane (which are marked by PIP3). Line-scan intensity profiles are shown in the bottommost panels. Throughout the study, line-scan intensity profiles are shown in bottommost or rightmost panels. Times are always indicated in seconds in top or left. Unless otherwise mentioned, all scale bars are 10 µm. **b, e, h** Representative line-kymographs of wave patterns shown in cell (**a**), (**d**), and (**g**), respectively, showing the consistency of complementary localization of PKBR1 (**b**), G$\beta\gamma$ (**e**), and RasG (**h**) with respect to front-state marker PIP3 over time. The intensities in all kymographs are plotted with "Turbo" colormap (shown in right). **c, f, i** Quantification of consistency and extent of complementarity of PKBR1 (**c**)/G$\beta\gamma$ (**f**)/ RasG (**i**) with respect to PIP3 in terms of Pearson's correlation coefficient (r). Number of cells: $n_c$=17 (**c**), 17 (**f**), 15 (**i**); $n_f$ = 20 frames were analyzed (7 s/frame) for each of $n_c$ cells. Unless otherwise mentioned, throughout the study, the Pearson's correlation coefficients (*r*) were computed with respect to PIP3 and $n_f$=20 frames were analyzed (7 s/frame) for each cell. Heatmaps were plotted in "Parula" colormap. **j** Time averaged Pearson's *r* of PTEN ($n_c$=16), CynA ($n_c$=15), RBD ($n_c$ = 16), cAR1 ($n_c$ = 17), PKBR1 ($n_c$ = 17), G$\beta\gamma$ ($n_c$ = 17), RasG ($n_c$ = 15), R(+8)-Pre ($n_c$ = 19), and PKBR1$_{N150}$ ($n_c$ = 15), where $n_c$ denotes the number of cell. To generate each data point, 20 frames (imaged at 7 s/frame) were averaged over time for each of these cells ($n_c$). Boxes extend from the 25th to 75th percentiles, median is at the center, and whiskers and outliers are graphed as per Tukey's convention (as computed by Graphpad Prism). Source data are provided as a Source Data file.

of the first 150 amino acids of PKBR1, designated *PKBR1$_{N150}$*, also showed strong complementary localization with respect to ventral waves of front-state markers and was analogously depleted from protrusions in migrating cell (Supplementary Fig. 2f–h, and Supplementary Movie 4). Although PIP3 level is a standard surrogate for marking front-state or protrusion, PIP3 was shown to be not essential for making protrusions and is often involved in other physiological processes. Hence, to assess the selective localization of our lipid-anchored proteins further, we performed a few additional experiments. First, we coexpressed LimE (the biosensor for newly polymerized F-actin) or RBD (the biosensor for activated Ras) with our lipid-anchored proteins and recorded their localization during protrusion formation. We found that LimE and RBD are enriched inside the protrusion or front-state regions, as expected, whereas our lipid-anchored proteins were selectively depleted from those membrane domains (Supplementary Fig. 3a–e). Second, we treated the cells with PI3K inhibitor LY294002 (which depletes the PIP3 level from the protrusions[9,51,52]) and found that lipid-anchored proteins were still consistently depleted from the F-actin-rich protrusions or front-state regions of the membrane (Supplementary Fig. 3f–i). Finally, we performed chemotaxis assay where a cAMP chemotactic gradient was introduced to the field of developed *Dictyostelium* cells. We found that during chemotactic movement of the polarized cells, PKBR1 and G$\beta\gamma$ were depleted in the front-regions of the membrane (which were marked with F-actin based protrusions) and were consistently enriched in the back of the cell (Supplementary Fig. 4a–c and Supplementary Movie 5). In summary, during ventral waves propagation, random migration, and chemotaxis, to the extent that has been tested, all of these five lipidated proteins exhibited preference toward the back-state regions of the membrane, resembling dynamics of standard back proteins such as PTEN and CynA which shuttle between membrane and cytosol. Again, these distributions contrast the dynamics of front protein/sensors such as $PH_{Crac}$ and $RBD_{Raf1}$ (Supplementary Fig. 5a–c). These distributions are also clearly distinct from the profile of surface receptor cAR1 (Supplementary Fig. 5d–f and Supplementary Movie 6) or other lipid-anchored proteins such as Lyn and Palm/Pre[9], all of which exhibits nearly homogeneous distribution over the membrane. Time-averaged Pearson's *r* values for all the five lipid-anchored proteins that we examined yielded negative values, whereas $RBD_{Raf1}$ and cAR1 values were positive and near zero, respectively (Fig. 1j).

## The cytoskeletal dynamics independent asymmetric distribution of lipid-anchored proteins

Even though F-actin polymerization wave peaks move with the waves of Ras-activation and PIP3 accumulation (Supplementary Fig. 1a), the signal transduction events can be triggered and the membrane can be spontaneously segregated into front- and back-states in the absence of F-actin as well[26,28,31,32,53,54]. To test whether the spatiotemporal separation of the lipid-anchored proteins depend on the existence of actin-barrier between front and back states, we first treated *Dictyostelium* cells with Latrunculin A. When periodic circulating waves were induced in these cells, typical symmetry breaking of PI3K activities was observed (Fig. 2a–f and Supplementary Fig. 6a–k). Standard peripheral back-associated membrane proteins, PTEN and CynA, were depleted from the circulating PIP3 crescents which marked the front-state regions of the membrane (Supplementary Fig. 6a–d). The 360° membrane kymographs[9,25] demonstrates the dynamics and consistency of CynA and PTEN depletion from front-states of the cell membrane (Supplementary Fig. 6b, d). Importantly, PKBR1 (Fig. 2a, b) and G$\beta\gamma$ (Fig. 2c, d) also consistently adjusted their localization towards the back-state regions of membrane throughout the time span of the experiment. We also observed that even on the ventral surface of these cytoskeleton impaired cells, the asymmetric waves of PKBR1 can propagate, maintaining consistent complementarity with respect to PIP3-rich domains (Supplementary Fig. 6e). RasG largely maintained its back state distribution as well (Supplementary Fig. 6f), although fidelity was slightly reduced (Supplementary Fig. 6g). The prenylated peptide R(+8)-Pre and myristoylated peptide *PKBR1$_{N150}$* dynamically localized away from PIP3 crescent-marked front-states in a highly consistent fashion (Fig. 2e, f, Supplementary Fig. 6h, i). Since we observed essentially the same dynamics for PKBR1 and *PKBR1$_{N150}$*, we will hereafter report only the findings on PKBR1. As a control, we recorded membrane wave patterns in cells co-expressing the GPCR cAR1 and the PIP3 sensor. As in ventral waves and migrating cells (Supplementary Fig. 5d–f), cAR1 exhibited uniform membrane distribution in cytoskeleton-inhibited cells as well (Supplementary Fig. 6j, k), demonstrating that membrane integrity remained intact in these experiments. To test the generality of this cytoskeleton independent compartmentalization of lipid-anchored membrane proteins, we next used RAW 264.7 macrophage cells where we observed ventral wave propagation by inducing frustrated phagocytosis, followed by osmotic shock[9,55,56] (please see "Methods" for details). Consistent with our previous report[9], we observed that during ventral wave propagation, R(+8)-Pre exhibited a consistent spatiotemporal complementary with respect to PIP3 waves (Supplementary Fig. 7a and Supplementary Movie 7). When we treated the cell with ROCK inhibitor Y-27632 (which also blocks membrane flow in macrophages[57]) and Latrunculin A together, before inducing frustrated phagocytosis, cells did not spread (Supplementary Fig. 7b). However, importantly, when we first treated the cells with Y-27632, allowed the cells to spread, and then added Latrunculin A, we observed strong complementarity between R(+8)-Pre and PIP3 – whenever a new PIP3 rich membrane region was created, R(+8)-Pre consistently moved away from that specific region (Supplementary Fig. 7c and Supplementary Movie 8). Taken together, our data so far establish that, in different physiological scenarios and in different cell systems, even in the absence of cytoskeletal dynamics, our lipid-anchored proteins consistently localized to the back-state regions of the membrane, maintaining significant exclusion from the membrane regions where the signal transduction network is activated to create front-states (Supplementary Fig. 7d).

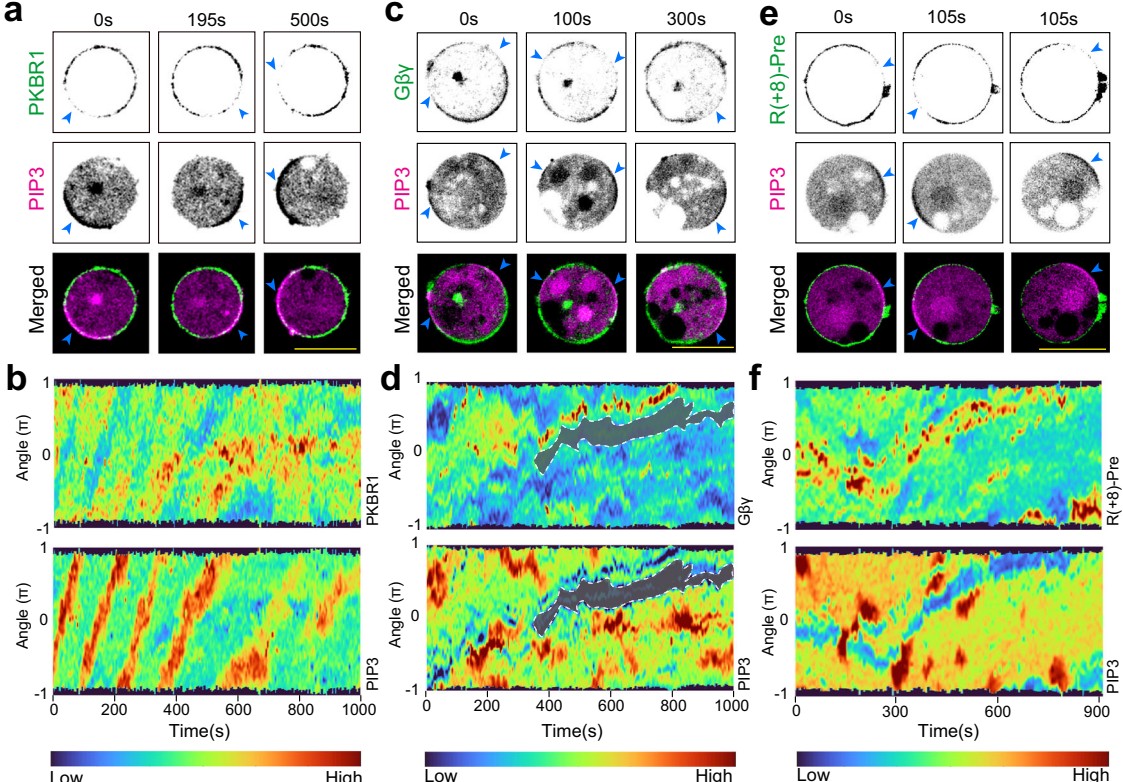

**Fig. 2 | Dynamic polarization of multiple lipid-anchored membrane proteins in cytoskeleton-impaired cells. a**, **c**, **e** Representative live-cell time-lapse images of *Dictyostelium* cell co-expressing $PH_{Crac}$-mCherry along with PKBR1-KikGR (**a**), KikGR-G$\beta\gamma$ (**c**), or GFP-R(+8)-Pre (**e**) showing depletion of PKBR1, G$\beta\gamma$, and R(+8)-Pre from the activated/front-states of the membrane, which are marked by the traveling PIP3 crescents (indicated with blue arrowheads). In all cases, cells were pre-treated with 5 μM Latrunculin-A (final concentration) to inhibit actin polymerization and waves were induced. **b**, **d**, **f** The 360° membrane kymographs (see "Methods" for details) of asymmetric wave propagation in cells shown in (**a**), (**b**), and (**c**), respectively. Note that the depletion of PKBR1 (**b**), G$\beta\gamma$ (**d**), and R(+8)-Pre (**f**) from the front-state crescents of PIP3 is highly consistent over the entire time course of the experiment.

## "Shuttling"-type and lipid-anchored back membrane proteins responds differently to receptor activation

Since previously identified back-state associated proteins were reported to dissociate from membrane and move to the cytosol upon signal transduction network activation (Supplementary Fig. 1j), the back-state association of our newly found lipid-anchored proteins (which presumably do not dissociate from membrane) was surprising. To test further whether these lipidated proteins remain associated with membrane during dynamic compartmentalization, we used chemoattractant-induced receptor activation (in Latrunculin A treated immobilized cells) which is an established process of uniformly converting the membrane into activated or front state[6,18,23,24,29,58–61]. Figure 3a, b and Supplementary Fig. 8a–d (and associated Supplementary Movie 9) illustrate that the front sensors, in this case $PH_{Crac}$, was transiently recruited from the cytosol to the membrane whereas back proteins, in this case CynA (Fig. 3a, b, Supplementary Fig. 8a, and Supplementary Movie 9) and PTEN (Supplementary Fig. 8b–d), were released from the membrane to cytosol upon global stimulation. After a short period of time, the system was adapted and the original localizations were eventually restored (Fig. 3a and Supplementary Fig. 8a, b, d). The time courses of shuttling were not identical for front and back proteins, but the complementarity in their reversible translocation was consistent.

Although the lipid-anchored proteins, as shown above, displayed similar spatiotemporal pattern of standard peripheral back proteins, they did not dissociate from the membrane during global chemoattractant stimulation. Throughout the time course of the experiment, PKBR1 (Fig. 3c, d, and Supplementary Movie 10), G$\beta\gamma$ (Supplementary Fig. 8e, f, and Supplementary Movie 11), R(+8)-Pre (Fig. 3e, f, and Supplementary Movie 12), as well as RasG (Fig. 3g, h)

remained bound to the membrane. The front-state indicator $PH_{Crac}$ consistently translocated to the membrane demonstrating robust receptor activation in each cases (Fig. 3c–h and Supplementary Fig. 8e, f). In fact, in this particular assay, the kinetics of all of these lipidated proteins resembled that of cAR1 (Supplementary Fig. 8g, h, and and Supplementary Movie 13), which, as shown earlier, exhibited symmetric distribution during protrusion formation and ventral wave propagation. Again, to test whether this same phenomena is conserved in mammalian cells, we globally activated C5a receptor in RAW 264.7 macrophages. We observed that upon C5aR agonist stimulation, PIP3 biosensor $PH_{Akt}$ consistently translocated to the membrane, indicating signaling activation, and eventually came back to the cyotosl, indicating adaptation (Supplementary Fig. 8i, j, and Supplementary Movie 14). R(+8)-Pre, on the other hand, just like in *Dictyostelium* cells, maintained membrane association throughout the time-course of the experiment (Supplementary Fig. 8i, j, and Supplementary Movie 14).

Together, these data suggest the need for a new model of the compartmentalization process that can drive the polarized distribution of lipid-anchored membrane proteins, since unlike shuttling-based pattern forming peripheral membrane proteins, they do not transiently dissociate from the membrane during network activation, yet can exhibit consistent and dynamic asymmetric distribution in different scenarios (Fig. 3i). We speculated that, even though these lipid-anchored proteins remain membrane associated, they nevertheless bind selectively to the two different membrane states. These differential affinities would possibly change their effective diffusion or mobility in different state-regions of the membrane and that, in turn, would drive a novel partitioning based compartmentalization or polarization process.

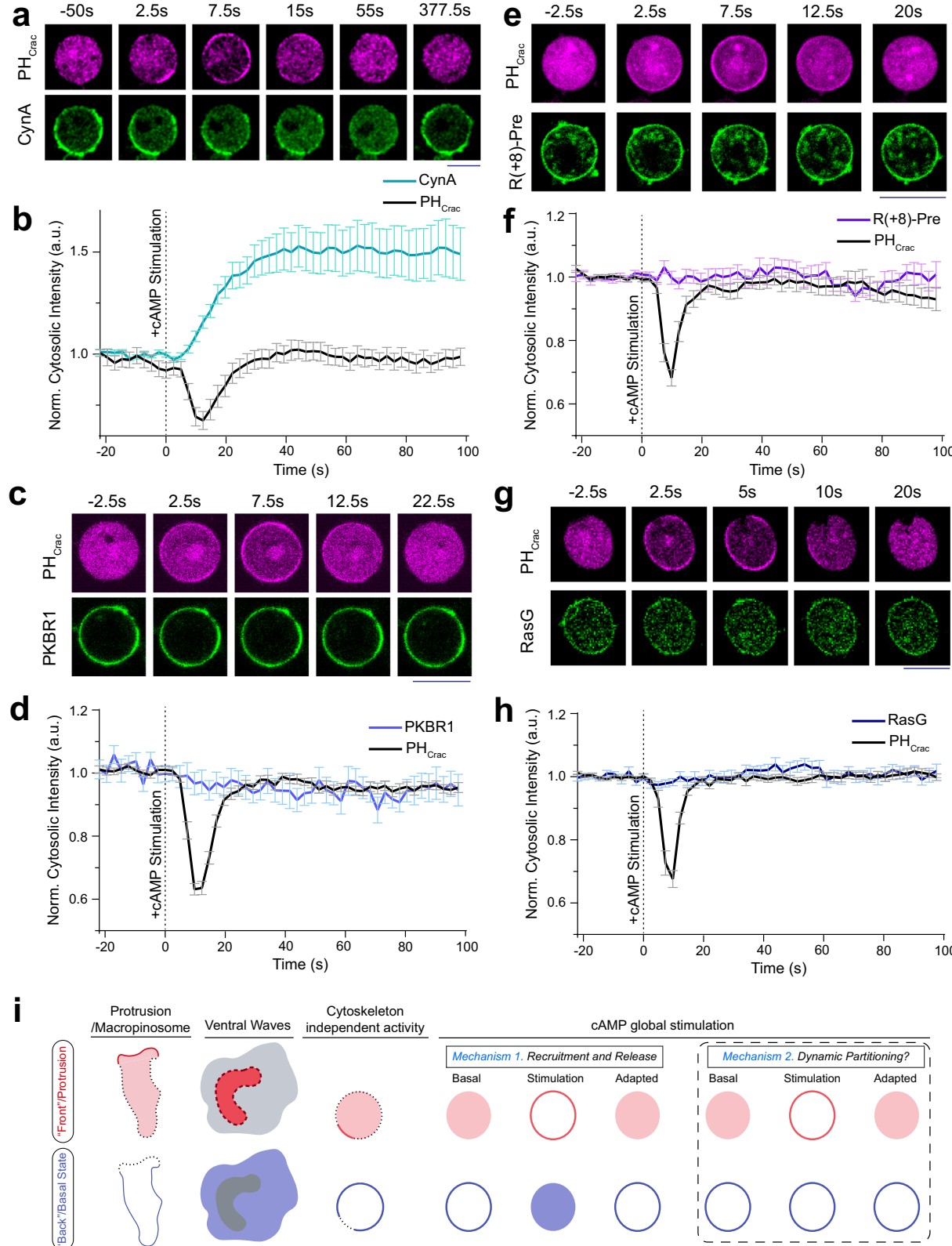

## Photoconversion microscopy suggests a novel partitioning mechanism for the asymmetric distribution of lipid-anchored membrane proteins

To test our hypothesis, first we fused photoconvertible proteins (such as KikGR or Dendra2) with our lipid-anchored or standard peripheral back membrane proteins and then studied their movements during ventral wave propagation (on the substrate-attached surface of

electrofused cells) by using selective photoconversion microscopy which offers high degree of spatiotemporal control in investigating binding and diffusion kinetics[62]. As a control, we started with Lifect-Dendra2 expressing cells and photoconverted a section of molecules on the propagating waves (Supplementary Fig. 9a). As previously surmised by Fluorescent recovery after photobleaching (FRAP) experiments[63,64], we recognized that actin-polymerization waves

**Fig. 3 | Profiles of back-associated peripheral, lipid-anchored, and integral membrane proteins during global receptor activation. a** Representative live-cell images of *Dictyostelium* cells co-expressing PH*$_{Crac}$*-mCherry and CynA-KikGR upon global cAMP stimulation, demonstrating that upon transient global activation of cAR1 receptors, PH*$_{Crac}$* gets uniformly recruited to membrane whereas CynA gets dissociated from the membrane and translocates to cytosol. Both responses adapted over time, although CynA adaptation took longer time. In all global stimulation experiments, at time t=0 s, 10 µM (final concentration) cAMP was added. **b** Time series plot of normalized cytosolic intensities of CynA and PH*$_{Crac}$*, showing the kinetics of the response upon global stimulation with cAMP (also see Supplementary Fig. 8a which demonstrates the time-course of adaptation for CynA). In all these figures, vertical dashed lines are used to indicate the time of stimulation. Mean ± SEM are shown for n$_c$=15 cells. **c–h** Response of *Dictyostelium* cells co-expressing PH*$_{Crac}$*-mCherry and PKBR1-KikGR (**c, d**) / GFP-R(+8)-Pre (**e, f**) / GFP-RasG (**g, h**) upon global cAMP stimulation. Live-cell images (**c, e, g**) and temporal profile of normalized cytosolic intensities (**d, f, h**) are shown demonstrating the transient recruitment of PH*$_{Crac}$* to membrane whereas lipid-anchored proteins such as PKBR1, R(+8)-Pre, and RasG remained steadily membrane bound throughout the entire time course of the experiment. Mean ± SEM are shown for n$_c$ = 17 cells (**d**), n$_c$= 15 cells (**f**), and n$_c$ = 15 cells (**h**). **i** Left three panels of the schematic summarizing the front-back complementarity in migrating cell protrusions, ventral wave propagation, and cytoskeleton independent signaling events. In right panels, schematic is showing two different responses observed during global receptor activation experiments, suggesting the existence of two different mechanisms that drive dynamic compartmentalization process. In contrast to "shuttling" based polarization of peripheral membrane proteins (Scenario 1), the lipid-anchored or integral membrane proteins (Scenario 2) do not dissociate, but possibly spatio-temporally rearranges over the plane of membrane to exhibit asymmetric distribution during different physiological processes. Source data are provided as a Source Data file.

propagate via continuous exchange of the actin binding protein molecules between the cytosol and the membrane. The photoconverted red Lifeact molecules dissociated and vanished from the plane of membrane within 30 s as green Lifeact wave continued to propagate presumably through recruitment of new green Lifeact molecules from cytosol (Supplementary Fig. 9a). Next, to distinguish between the shuttling vs. lipid-anchored back proteins, we decided to photoconvert a patch of molecules just in the front of a propagating "shadow" wave (a moving zone depleted of back-state proteins), i.e., where membrane is on the verge of switching from back- to front-state (Fig. 4a–d). Note that Ras/PI3K/Akt/Rac1/F-actin network is activated in the shadow region. If the pattern that a particular component is displaying, is generated via recurring shuttling or directed endocytosis and vesicle fusion, then as the shadow wave reaches the photoconverted region, the photoconverted molecules would vanish (Scenario 1 in Fig. 4a). On the other hand, if a protein self-organizes into patterns via partitioning mechanism, the photoconverted molecules would stay in the plane of membrane and move laterally to rearrange to other back-state regions of the membrane (Scenario 2 in Fig. 4a). To make sure that shadow waves had traveled to the photoconverted area and any loss/rearrangement of signal is indeed due to switching of back-state to front-state we performed optical flow analysis (as per Horn-Schunk method[65,66]) with segmented masks of shadow waves and photoconverted region and computed the angle between their resultant vectors (Fig. 4b, see "Methods" for details).

We found that, as expected, photoconverted PTEN and CynA molecules vanished as shadow waves crossed the photoconversion regions, i.e. when back-states switched to front-states in the membrane (Fig. 4e; Supplementary Fig. 9b, c; Supplementary Movie 15, Supplementary Movie 16). On the other hand, photoconverted PKBR1 and *Gβγ* molecules stayed on the membrane and moved laterally on the plane of the membrane to rearrange themselves in existing back-state regions as waves propagated (Fig. 4c–e; Supplementary Fig. 10a, b; Supplementary Movie 17, Supplementary Movie 18). This consistent association of the majority of photoconverted PKBR1 and *Gβγ* molecules on the membrane not only excludes the possibility of shuttling, but also rules out the necessity of directed vesicular trafficking in symmetry breaking of these lipid-anchored proteins. While much slower trafficking pathways can still exist for these proteins, the photoconversion assay demonstrates that it cannot significantly contribute to highly dynamic spatiotemporal organization of these proteins on the membrane, as it happens during ventral wave propagation or protrusion formation. The automated optical flow analysis (third and fourth panels in Fig. 4c, d; Supplementary Fig. 9b, c; Supplementary Fig. 10a, b) proves that the partitioning of PKBR1 and *Gβγ* as well as shuttling of PTEN and CynA were due to the shadow wave propagating through the photoconverted domain of the membrane and not due to a random event on the membrane (Fig. 4f, g; Supplementary Fig. 10c–e). Together, our data suggests that lipid-anchored

proteins undergo compartmentalization and form patterns via a dynamic rearrangement process within the plane of the plasma membrane.

## Single-molecule imaging in front and back-state regions of the cell membrane

To gain insight into the underlying molecular reaction and diffusion process that drives dynamic rearrangement, we measured the diffusion coefficient of individual molecules of lipidated protein PKBR1 by single-molecule imaging and compared their dynamics with that of the individual molecules of the typical back-state associate peripheral protein PTEN[32,67]. We first verified that PKBR1-Halo-TMR consistently localizes to the back-state regions of the membrane during ventral wave propagation (Fig. 5a). To keep track of the instantaneous front-back state demarcation on the membrane, we used multiscale imaging where we detected the broad PIP3 waves and PKBR1-Halo-TMR single molecules simultaneously (Fig. 5b, Supplementary Movie 19). These appeared as diffusing fluorescent puncta (Fig. 5c, d, Supplementary Movie 19). We confirmed that the single fluorescent puncta of PKBR1-Halo-TMR seen on the TIRF are indeed single molecules with single-step photobleaching curves (Supplementary Fig. 11a) and fluorescence intensity distribution of puncta (Supplementary Fig. 11b). Displacement profiles for each individual PKBR1 molecules were measured in front- as well as back-state regions of the membrane via single-particle tracking (Fig. 5d, e).

Lifetimes of membrane binding were computed using the time duration between appearance and disappearance of individual fluorescent spots and then fitting with three exponential components (Supplementary Fig. 11c, Supplementary Table 1). The effect of fluorophore photobleaching was excluded by using the photobleaching rates measured under respective experimental conditions. The mean lifetime analysis (Supplementary Fig. 11c–e, Supplementary Table 1) suggests that, within each region, the majority of PKBR1 and PTEN molecules remains membrane bound during the time-course of the single-molecule measurements. The major difference, as also shown by the photoconversion studies, is that PTEN leaves the membrane as the active zone approaches whereas PKBR1 remains membrane bound, which presents the issue of how PKBR1 delocalizes.

To investigate the difference in diffusion of PKBR1 molecules in front vs. back state regions, we performed short-range diffusion (SRD) analysis by estimating mean diffusion coefficient for arbitrary 0.5 s during the diffusion trajectory using mean-squared displacement (see Methods and ref. 68 for details) (Fig. 5f). The histograms showed two peaks at around 0.01–0.02 and 0.4–0.5 µm²/s for front and back-states, and it was clear that, compared to front-state associated group, back-state associated cohort had a significantly larger slower-mobile fraction (Fig. 5f). To quantitate the diffusion coefficients, we performed the displacement distribution analysis[69] where probability density functions were fit to distributions of

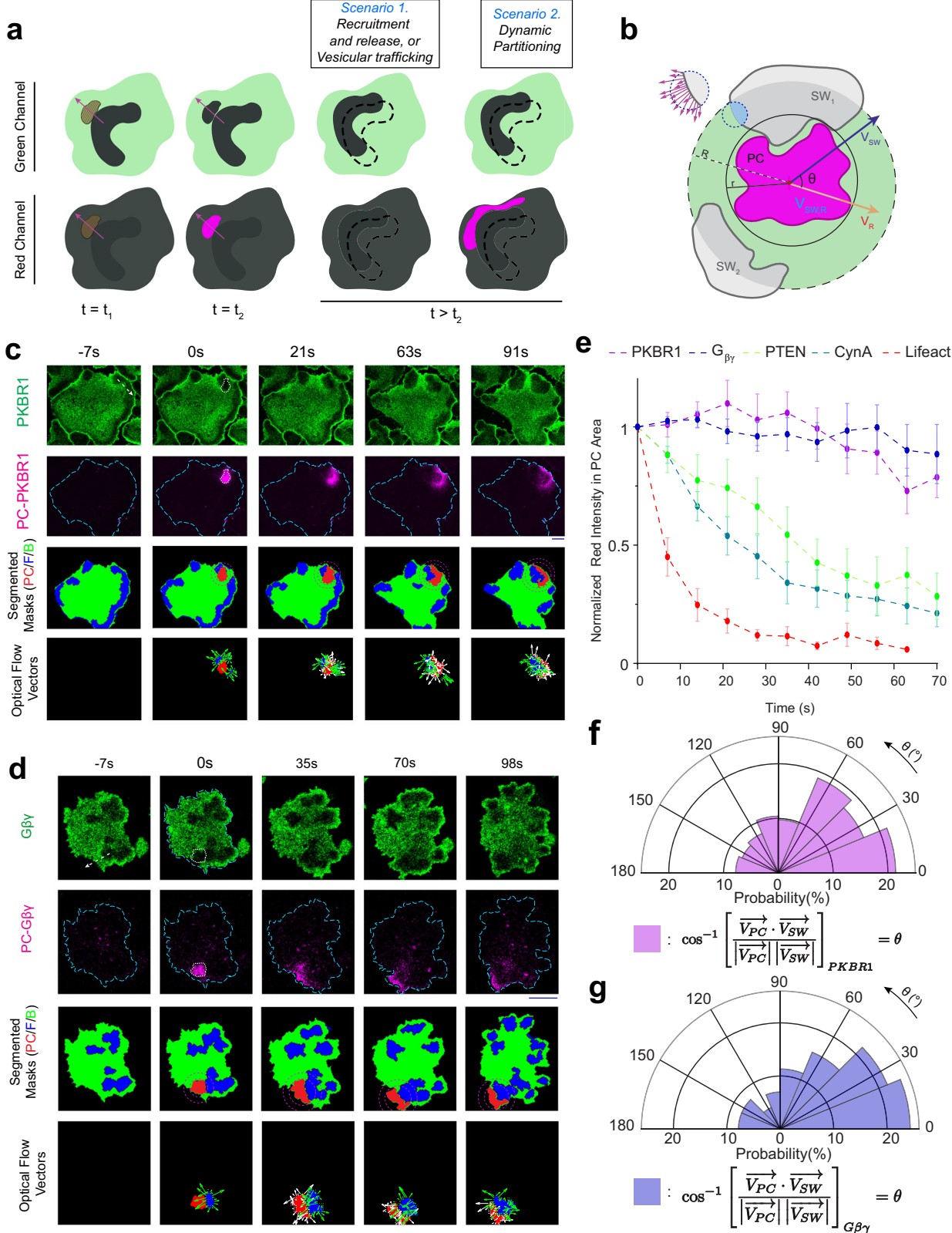

displacement with the shortest lag time, Δt = 33 ms (Fig. 5g). The distribution clearly showed that, compared to the front-state localized cohort, the back-state localized PKBR1 molecules generally exhibited shorter displacements (Fig. 5g). The diffusion coefficient of the slowest mobile fraction was 0.02 $\mu m^2$/s irrespective of the membrane state (Supplementary Table 2). Taking into account that the total amount of front-state bound PKBR1 was 0.56-fold of that of

back-state bound one according to the quantification in TIRFM images (Supplementary Fig. 11f), the fraction of the slowest mobility in the front state group was about 5%, whereas the fraction in the back state group was near 20% (Supplementary Fig. 11g). The fast versus slow diffusion coefficients differed by about 30-fold. These data show that more PKBR1 molecules accumulate in the back-state region because their diffusion is slower in that region. That is,

**Fig. 4 | Photoconversion microscopy based protein tracking assay of different back-associated lipid-anchored proteins. a** Setup of photoconversion experiment and possible mechanisms of wave propagation on the substrate-attached surface. In the cells where a photoconvertible fluorescent protein tagged backprotein was expressed, waves of activated regions appear as dynamic dark shadows (dark gray regions). The 405 nm laser was selectively illuminated in an area ahead of such shadow waves. Purple arrows: Wave propagation direction; tan-colored hatched region: photoconversion area. The dynamics of the molecules which were converted from green to red (magenta region in bottom panels) were tracked and analyzed for different proteins. **b** Schematic of optical flow vector analysis. PC: Photoconverted area shown in magenta, SW: Shadow waves (i.e. the front-state/activated region waves of the membrane, as they appear in the cells expressing a back proteins) shown in light-gray. Inner circle encloses photoconverted area whereas outer circle shows the area up to which shadow waves were considered for optical flow analysis (R = 0.2r--0.3r). $V_{SW}$: Resultants of all shadow wave vectors inside the outer circle (a zoomed in part is shown with violet flow vectors). $V_R$: Resultant optical flow vectors of photoconverted region PC. **c, d** Live-cell time-lapse images of *Dictyostelium* cells expressing PKBR1-KikGR (**c**) or KikGR-G$\beta$ (**d**) showing very little dissociation of PC-PKBR1 and PC-G$\beta\gamma$ molecules from the

membrane as waves propagated through the initial illumination area, indicating a spontaneous dynamic partitioning and lateral propagation mechanism. Third horizontal rows are showing masks generated by automated segmentation; PC(red): Photoconverted area, F(blue): Front-state regions (which appeared as shadow waves in green channel imaging), B (green): Back-state regions shown in green. Inner and Outer Magenta circles: as described in (**b**). The last horizontal rows are showing optical flow vectors along with segmented photoconversion area and associated -shadow wave regions. Shadow-wave region's and photoconverted region's optical flow vectors are shown in green and white, respectively. **e** Time-series plot of normalized intensity of the photoconverted membrane molecules demonstrating that intensity of lipid-anchored membrane proteins (PKBR1, G$\beta\gamma$) do not change as waves propagate whereas intensities of typical shuttling-type peripheral membrane proteins (PTEN, CynA, Lifeact) decrease sharply within 70 s. Data are mean ± SEM. $n_c$ (number of cells) = 14 (for PKBR1), 10 (for G$\beta\gamma$), 11 (for PTEN), 13 (for CynA), 11 (for Lifeact). **f, g** Polar histograms depicting the probability distribution of angle between resultant of optical flow vectors of front-state shadow-waves ($V_{SW}$) and of the photoconverted regions ($V_{PC}$). (**f**): PKBR1, $n_f$ = 154 frames; (**g**) G$\beta\gamma$, $n_f$ = 97 frames. Source data are provided as a Source Data file.

because of the relative diffusion rates the flux of PKBR1 molecules within the plane of the membrane is biased toward the back state. We term this spatially heterogeneous diffusion process, "dynamic partitioning", and propose that this mechanism underlies pattern formation for lipid-anchored or otherwise tightly associated membrane proteins during different physiological processes.

## Stochastic simulation of an excitable system demonstrates that "dynamic partitioning" and "shuttling" can generate similar propagating wave patterns

To test whether molecules that can only diffuse on the plane of the membrane can theoretically still exhibit spatially asymmetric dynamic wave patterns in silico, we incorporated reaction and diffusion dynamics involving lipid-anchored (LP) and exchangeable peripheral membrane proteins (PP) into a previously reported excitable network model (Fig. 6a) that has been used to explain ventral wave propagation and cell migration phenotypes[9,28,31,70,71]. The model consists of three system states: front (F), back (B), and refractory (R) (Fig. 6a). It was demonstrated earlier that, such excitable network, consisting of a mutually inhibitory action (between F and B), feedforward interaction (between F and R), and delayed negative feedback loop (between R and F) can give rise to firing of the system i.e. a complete excursion in the phase space, when stochastic noise can cross the threshold of the network (see Methods for details). This, in turn, generates defined patterns in two dimensions which underlies protrusion formation[28,31,71]. To include realistic random stochastic noise, we simulated an unstructured mesh based spatiotemporal reaction diffusion system (Supplementary Fig. 12a) using the URDME framework (see Methods for details). The URDME-based stochastic spatiotemporal simulations demonstrated that when the system fires, F and B exhibited a complementary pattern whereas R exhibited slightly delayed activity profile compared to F (Fig. 6b, c, Supplementary Movie 20). We incorporated binding reactions into the reaction-diffusion model where all the back proteins bound more strongly to the B- than the F-state (Fig. 6a). Upon dissociation, shuttling peripheral proteins (PP) were released to the cytosol, but lipidated proteins (LP) remained on the membrane (Fig. 6a). We considered the "slow" and "fast" distributions of diffusion coefficients measured from single-molecule imaging to define the diffusion dynamics of the tightly membrane-bound and membrane-associated free states of the membrane protein molecules, respectively. In both front and back regions, we determined the fraction of molecules of membrane-bound and membrane-associated free states by fitting probability distribution data from the single-molecule experiments (see Methods for details). All the reaction parameters are listed in Supplementary Table 4.

Importantly, in the two-dimensional representations of the simulations, we observed that PP and LP showed similar pattern which resembled the compartmentalization pattern of B (Fig. 6c, d, and Supplementary Movie 20). Simulations also demonstrated that at each node, different fractions of LP or PP can interconvert (Supplementary Fig. 12b). Although the total concentration of membrane-bound LP did not change, the total concentration of membrane-bound PP decreased as the system fired and recovered when system was restored to the basal state (Fig. 6b). It was also interesting to note that, LP molecules accumulated in the areas just ahead of advancing-waves (Fig. 6c, Supplementary Movie 20), reminiscent of the spatial profile of lipid-anchored proteins in the photoconversion assay (Fig. 4c, d). To gain further insight on this, we simulated photoconversion where we converted a fraction of molecules of PP and LP right in front of a propagating F-wave (Supplementary Fig. 12c). As observed in the experiments, when F-state wave hit the photoconversion area, the membrane-associated PP molecules vanished whereas membrane-associated LP molecules stayed and partitioned into B-state (Supplementary Fig. 12c). Next, to check whether the spatially asymmetric pattern formation of LP is dependent on the difference in diffusion between membrane bound and membrane unbound forms, we next forced the diffusion coefficient of these two forms to be equal (Supplementary Fig. 13a–c and Supplementary Movie 21). Under this condition, the spatial heterogeneity in the LP channel abrogated (Supplementary Fig. 13a and Supplementary Movie 21), while other dynamics of the system remained unchanged (Supplementary Fig. 13b, c) and PP still faithfully aligned to the asymmetric pattern of the B-state (Supplementary Fig. 13a and Supplementary Movie 21). These simulations together demonstrate that the inherent heterogeneity in the membrane can give rise to differential diffusion-driven dynamic partitioning of lipid-anchored membrane proteins which can be sufficient to induce compartmentalization and generate patterns that are similar to those generated by peripheral membrane proteins which shuttle between membrane and cytosol.

## Optogenetic alteration of membrane region-specific binding affinity can induce compartmentalization of uniform lipidated and integral membrane proteins

As the experimental data and computational modeling together demonstrated that higher affinity towards specific domains of the plasma membrane can slow down the mobility of different lipid-anchored membrane proteins and can result in their polarized distributions, we wondered whether normally uniform membrane proteins can generate asymmetric patterns if their membrane region-specific affinity can be artificially manipulated. To test this idea, we

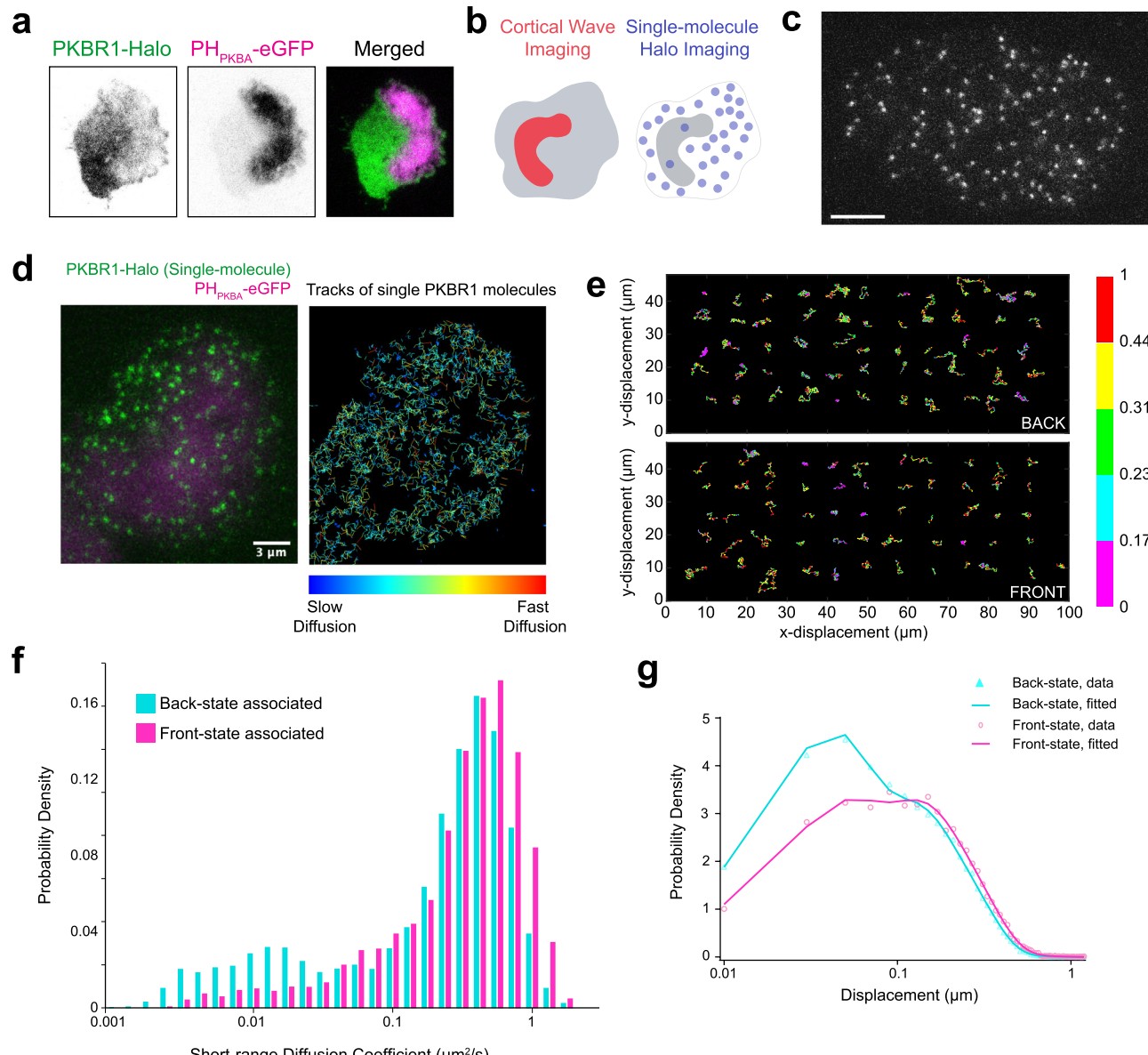

**Fig. 5 | Single-molecule imaging experiments to measure the different diffusion coefficients in front and back-state regions of the plasma membrane.**
**a** Representative live-cell image of *Dictyostelium* cells co-expressing PKBR1-Halo and PIP3 sensor PH*PKBA*-eGFP showing a complimentary localization profile of PKBR1 and PIP3 during wave propagation. **b** Set up of single-molecule imaging experiments where the coordinates of dynamic front-states were spatiotemporally tracked by imaging ventral waves using PH*PKBA*, and single-molecules of Halo tagged (TMR conjugated) PKBR1 was imaged in the other channel. **c** A representative TIRF microscopy image of *Dictyostelium* cell showing PKBR1-Halo-TMR molecules (scale bar: 5 μm). Also see Supplementary Fig. 11a, b for single-molecule characterization. **d** Left: A representative multiscale TIRF microscopy image of *Dictyostelium* cell where magenta is showing the front-state regions with high PIP3 level whereas green is showing the single PKBR1-Halo-TMR molecules throughout the membrane. Right: The trajectories of single PKBR1 molecules movement detected during 2s in the cell shown in left. The colormap indicates the amount of

movement. Note that, PKBR1 molecules are less in front-state regions of the membrane and they are moving slowly inside the back-state regions of the membrane which can explain their increased accumulation inside back-state regions. **e** Color-coded Trajectories of single PKBR1 molecules undergoing lateral diffusion on the membrane inside the back- (upper panel) and front-state regions (lower panel). Color bar on the right is depicting the amount of displacement between two consecutive frames (numbers in right colorbar are displacements in μm). **f** Histograms of the short range diffusion coefficients of front-state associated PKBR1 molecules (magenta) and back-state associated PKBR1 molecules (cyan) showing a significant fraction of back-state PKBR1 molecules exhibit a highly slower lateral diffusion compared to their front counterparts. **g** Probability density distribution of the displacement of single PKBR1 molecules during 33 ms in front- and back-states of the membrane indicating displacement in back-state is comparatively less. Source data are provided as a Source Data file.

devised a biophysical perturbation strategy, building upon the CRY2PHR/CIBN-based optogenetic system which can rapidly translocate a protein of interest from cytosol to membrane, in a light-gated fashion (Fig. 7a and Supplementary Fig. 14a). We decided to fuse CIBN with different integral and lipid-anchored membrane proteins which do not physiologically exhibit polarized distribution and then to selectively increase their back-state specific binding affinity, we

planned to recruit a cytosolic CRY2PHR-fused protein of interest which has a back-state affinity on its own. Although several options exist that have a selective back-state binding affinity (e.g. PTEN, PH domain of phospholipase C $\delta$1, PH domain of CynA, etc), as a proof of concept, we chose a short positively charged peptide (Fig. 7a and Supplementary Fig. 14a). The peptide, R+, consisting of +8 charge, was obtained by deleting the CAAX motif from the R(+8)-Pre which is a farnesylated

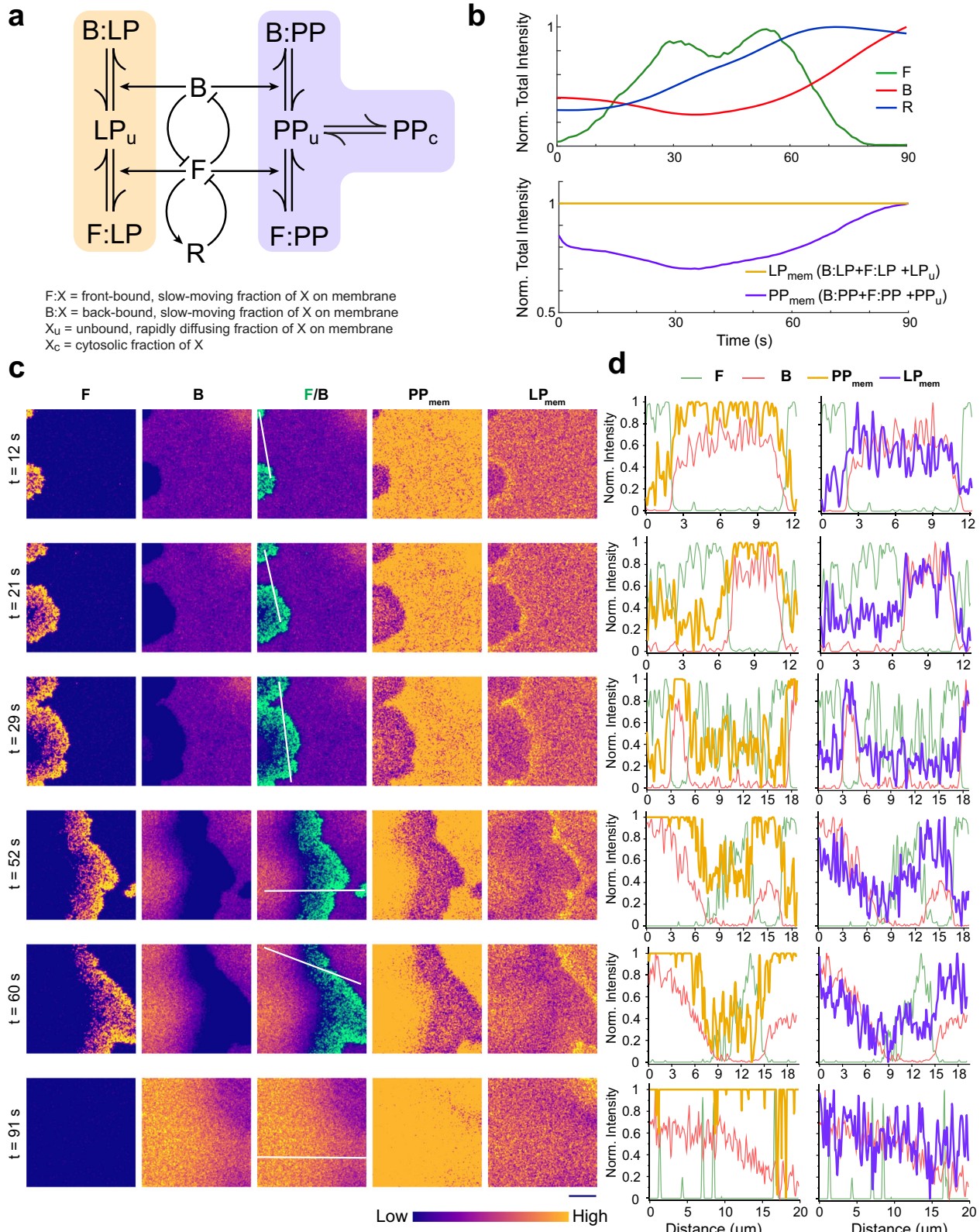

F:X = front-bound, slow-moving fraction of X on membrane
B:X = back-bound, slow-moving fraction of X on membrane
$X_u$ = unbound, rapidly diffusing fraction of X on membrane
$X_c$ = cytosolic fraction of X

peptide that exhibits strong preferential back-state localization, as shown in Supplementary Fig. 3d, e (also as documented earlier[9]). First, in migrating *Dictyostelium* cells, we recruited this peptide to the transmembrane GPCR cAR1 which normally exhibits symmetric distribution over membrane, as demonstrated in Fig. 2d–f (Fig. 7a). Upon light-induced recruitment, we observed that, within a minute, the recruited peptide, which is a proxy for cAR1 localization as well, started

exhibiting polarized distribution by dynamically partitioning to the back-state regions of the membrane (Fig. 7b, Supplementary Fig. 14b, Supplementary Movie 22). Each time a new protrusion formed, cAR1 spatiotemporally readjusted its localization within the back-state regions of the membrane, presumably due to its synthetically increased affinity towards the back-state regions which has slowed its diffusion there (Fig. 7b, Supplementary Fig. 14b, Supplementary

**Fig. 6 | Spatiotemporal stochastic simulation of an excitable network that incorporates the dynamics of lipid-anchored and peripheral back-state associated membrane proteins. a** Schematic showing excitable network, coupled with reactions involving peripheral membrane proteins (PP) and lipid-anchored membrane proteins (LP). Excitable network consists of three membrane states: F (front), B (back), and R (refractory). Membrane-associated, freely moving unbound species (denoted with `u' subscripts) binds with two different states of the membrane) to form strongly membrane-bound, slowly moving species (denoted with B: and F: notations for back-region bound and front-region bound species, respectively). Unlike PP, LP cannot shuttle between membrane and cytosol. **b** Temporal profiles of normalized total intensity for different species (F, B, R, total LP, and membrane-associated PP). Although the bound and unbound fraction of LP varies locally (see

Supplementary Fig. 12b), the total amount of LP on the membrane remains unchanged over time, whereas due to shuttling between membrane and cytosol, the total membrane fraction (combining bound and unbound) of PP fluctuates. **c** Simulated spatiotemporal profiles of F, B, combined F/B, and total membrane fractions of PP and LP. As wave propagation was initiated (from the left edge of the simulation domain), due to stochastic firing of the excitable network, both PP and LP exhibited compartmentalization and became dynamically aligned with the back-state. Dynamic profiles are shown in Matplotlib "plasma" colormap, as shown below. **d** Normalized spatial intensity profiles of total membrane fraction of LP, PP, F, and B along the white lines in (c). Note that like experimental observations, simulated LP profiles show the slight accumulation in the areas just ahead of the advancing-waves.

Movie 22). Analogously, highly dynamic symmetry breaking of cAR1 was observed during the ventral wave propagation at the substrate-attached surface of the cell where cAR1 was consistently depleted from the front-state waves, marked by high levels of PIP3 and F-actin polymerization (Fig. 7c, d). Next, to examine whether such selective affinity alteration can be sufficient to asymmetrically distribute typically uniform membrane proteins in mammalian cells as well, we used HL-60 neutrophil cells, which exhibit a defined front-back polarity upon differentiation. There we recruited the same peptide R+ to the uniformly distributed CIBN-fused Lyn11, a myristoylated and palmitoylated protein (Supplementary Fig. 14a). Light-driven recruitment in the polarized neutrophils resulted in consistent alignment of Lyn11 to the back-state membrane regions of the cell which closely tracked with the localization dynamics of the recruited peptide (Fig. 7e, Supplementary Movie 23). As a control, we recruited uncharged CRY2PHR-mCherry either to cAR1-CIBN (in *Dictyostelium*) or to Lyn11-CIBN (in HL-60 cells); neither of these recruitment's altered the uniform localization of cAR1 or Lyn11 over the membrane (Supplementary Fig. 14c, d, Supplementary Movie 22, Supplementary Movie 24). To further test whether cytoskeleton-driven rearward membrane flow is playing any major role in this optogenetic-recruitment induced back-localization of Lyn11, we treated the cells with ROCK inhibitor Y-27632 and repeated the CRY2PHR-mCherry-R+ global recruitment experiment in HL-60 cells. We observed that, even when rearward membrane flow is impaired, Lyn11 consistently vacated the cell front or protrusion areas and accumulated at the back (Supplementary Fig. 14e, f). Quantification in terms of line-scan analysis, Pearson's correlation coefficients, and front-to-back intensity ratios (Fig. 7f, Supplementary Fig. 14g–i) demonstrates that R+ recruitment can induce polarization of cAR1 or Lyn11 in plasma membrane, whereas uncharged control recruitments do not alter the uniform distribution.

## Discussion

We have shown that a variety of lipidated membrane proteins, such as *Gβγ*, PKBR1, and RasG, as well as several synthetic lipidated peptides, which were reported[29,72–79] or might be expected to distribute uniformly on the membrane, instead align to dynamic self-organizing membrane domains. Heretofore, these front- and back-state membrane regions, which are defined by the orchestrated opposing signal transduction activities, were assumed to be created by "shuttling" of proteins themselves or enzymes that differentially modify lipid head groups, as exemplified by PI3K and PTEN, which display cytosol/membrane exchange, and PIP3, which is regulated by modifications by these enzymes. However, photoconversion showed that the new examples we examined exchanged only slowly with cytosolic pools, prompting us to seek an alternative explanation. Careful observation showed that the photoconverted proteins, which remained on the membrane, gradually "sorted" into the evolving patterns. We theorized that partitioning would occur if the diffusion coefficient were different in the front- versus back-state regions. Single molecule measurement data of PKBR1 and PTEN bore this out, with a more than 3-fold higher probability of the back-state associated PKBR1 molecules displaying

a nearly 30-fold smaller diffusion coefficient. Computational modeling, based on those observations demonstrated that dynamic partitioning versus shuttling could result in similar patterns, although with some distinguishing characteristics.

Dynamic partitioning mechanism that we establish here (Fig. 8) is distinct from multiple mechanisms that have been previously proposed to explain compartmentalization which bring about polarization or traveling waves on the cell cortex. In addition to the examples of self-organizing patterns in *Dictyostelium* mentioned above, "shuttling" or relocalization of proteins between the cytosol and membrane has been shown to drive pattern formation during the propagation of Hem-1 (of SCAR/WAVE complex) waves in migrating neutrophils[63], Cdc42/FBP17 waves in tumor mast cells[20,80], Actin-polymerization/Rho-actvity/RhoGAP (RGA-3/4) waves in *Xenopus* (frog) eggs and *Patiria* (starfish) embryo[81,82], Actin-polymerization/PI3K waves in epithelial breast cancer cells[83], myosin IB/actin polymerization waves in *Dictyostelium*[64] as well as waves of multiple signaling components in *Dictyostelium*[25,31,45]. In distinction to shuttling, "fence and picket" models of membrane organization have been proposed to explain polarized distributions in the membrane[21,22]. The models rely on actin-based cytoskeletal "fences" to hinder long-range diffusion of transmembrane proteins as well as peripheral proteins on inner and outer surface of the membrane and thereby compartmentalize the plasma membrane. Such cytoskeleton-driven diffusional barrier, originally proposed in fibroblast-like cells[84,85], were shown to organize the differential distribution of receptors in front vs back regions of the membrane in the phagocytic macrophages[86,87], and to induce polarized distribution of different transmembrane and lipid-anchored proteins in somatodendritic vs. axonal membrane domains in neurons[88–90]. Finally, intracellular sorting by directed vesicular transport has been invoked to explain asymmetry of proteins in plasma membrane. Such spatially and temporally regulated vesicular transport, which is normally cytoskeleton dependent, was shown to polarize integral membrane proteins in migrating *Dictyostelium* and neutrophil cells[38–41]. Similarly, vesicular trafficking in axonal initial segment, often in conjunction with cytoskeleton mediated diffusional restrictions, were shown to contribute in polarizing transmembrane receptors in neurons[42–44].

Our results showed that the dynamic patterns of the lipidated proteins that we examined required a completely different explanation. Instead of shuttling, anchoring, or trafficking, these molecules simply diffuse more rapidly in front- versus back-state regions of the membrane. In essence, the slower mobility rates in the back-state regions serves as a molecular trap, concentrating molecules there at the expense of the front state regions (Fig. 8). We propose, without concrete evidence as yet, that the slower diffusion rates in the back-state regions results from complex formation with entities in the back-state region. Importantly, in different physiological scenarios, back-state regions were shown to be distinct from the front-state regions in terms of lipid composition and physical properties[6,9,50,86,91]. For example, the back-state regions of the membrane maintains higher negative surface charge, compared to the front-state regions[9]. Hence the

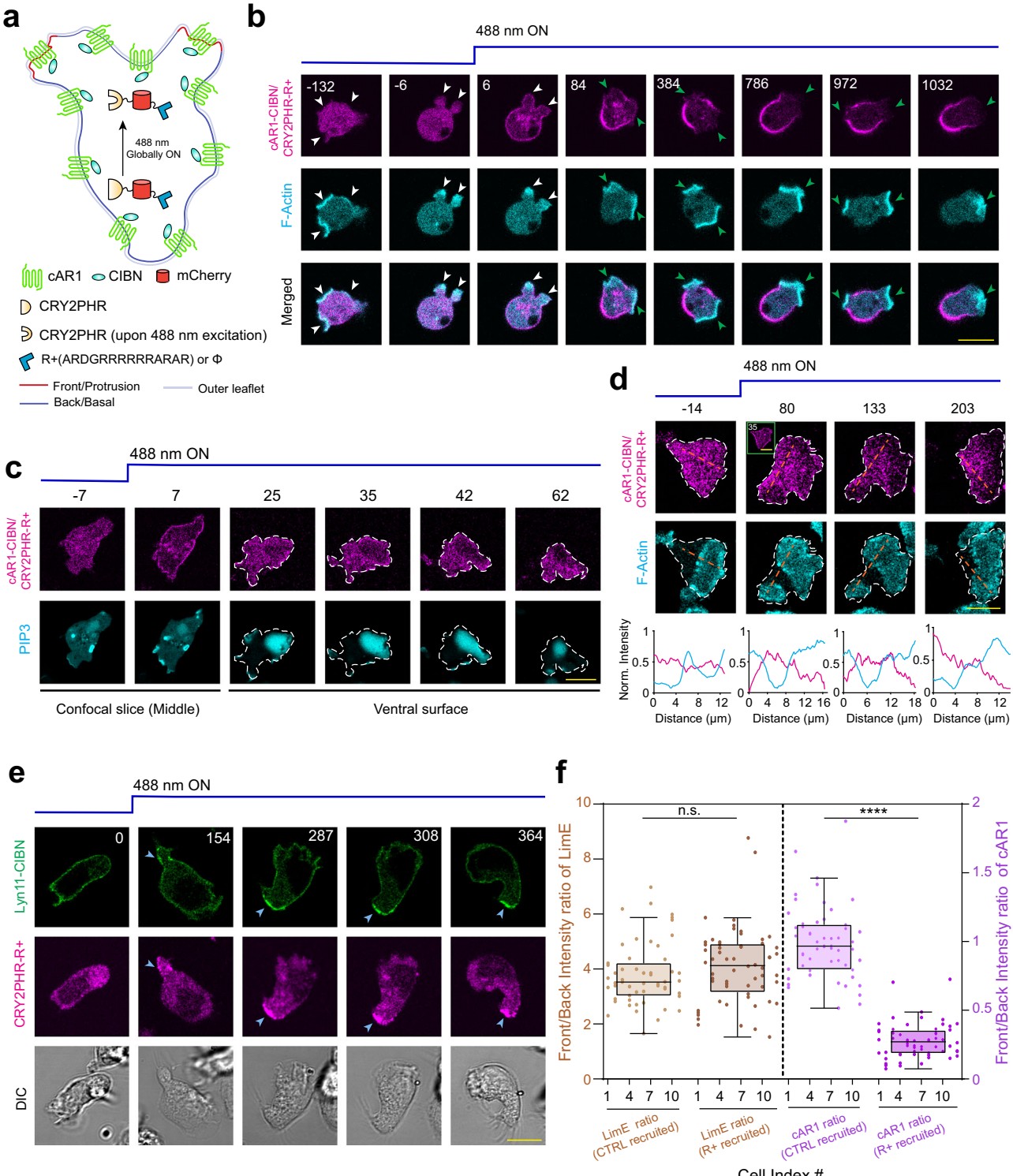

dynamic partitioning of R(+8)-Pre is likely driven by its positive stretch of charge. Since RasG and PKBR1 also have a stretch of polybasic residues in the C-terminus and N-terminus respectively, we can further speculate that those motifs are possibly important to partition these proteins to the back-state. While electrostatic interaction of Gβγ with the membrane via its charged motif is also known for other cell systems[92], it remains to be seen whether it plays any key role in *Dictyostelium* cells. Recently, it was also found that transmembrane proteins with widely different structures exhibited three free diffusion states with similar diffusion constants irrespective of their structural variability, following Saffman-Delbrück model[93]. We speculate that it is

also possible that slowly moving domains of transmembrane proteins accumulate in the back-state regions and that helps in complex formation, either by binding with lipid-anchored proteins or by retaining back-state region specific anionic phospholipids. Since these broad regions propagate rapidly across the membrane, the putative complexes must be formed and disbanded rapidly and reversibly. In the literature, a number of mechanisms have been proposed to explain this kind of supramolecular complex formation in the cytosol or in membrane, such as liquid-liquid phase separation[94,95], molecular crowding or trapping with scaffold proteins[96], and the formation of lipid rafts[21,97,98]. Selective formation of these sort of complexes can

**Fig. 7 | Effect of the acute manipulation of membrane region specific affinity of different lipid-anchored and integral membrane proteins. a** Schematic for increasing the back-state region specific affinity of uniformly distributed trans-membrane protein cAR1. Upon 488 nm irradiation, the cytosolic CRY2PHR, which is fused to positively charged peptide (R+) or blank (ϕ, CTRL), gets globally recruited to CIBN-fused cAR1. **b** Live-cell images of *Dictyostelium* co-expressing cAR1-CIBN, CRY2PHR-mCherry-R+, and Lifeact-HaloTag(Janelia Fluor 646), before and after global 488 nm illumination (in all cases, laser was turned on at time t = 0 s). White arrowheads: F-actin rich protrusions before or right after recruitment; Green arrowheads: F-actin rich protrusions from where cAR1-CIBN/recruited CRY2PHR-mcherry-R+ was excluded. **c, d** Live-cell images of ventral wave propagation in *Dictyostelium* cell co-expressing cAR1-CIBN, CRY2PHR-mCherry-R+, along with PH$_{Crac}$-YFP (**c**) or Lifeact-HaloTag(Janelia Fluor 646) (**d**), before and after global 488 nm irradiation. First two time point images in (**c**) and the inset image of second time point in (**d**) are showing confocal slices around the middle z-section (proving

successful recruitment); other images are focusing on the substrate-attached surface of cell to visualize wave propagation. **e** Live-cell images of differentiated HL-60 cells, before and after recruitment of cytosolic CRY2PHR-mCherry-R+ to membrane bound Lyn11-CIBN-GFP. Blue arrowheads: The uropods or back-state regions of neutrophils where CRY2PHR-mCherry-R+ was localized upon recruitment, which in turn polarized the membrane distribution of Lyn11 there. **f** Box and whisker plots and aligned dot plots of front-state regions to back-state region intensity ratio of F-actin biosensor LimE (tan) and cAR1-CIBN (purple), after the recruitment of CRY2PHR-mCherry-R+ or CRY2PHR-mCherry(CTRL). For each of the $n_c$ = 11 (for CTRL) or $n_c$ = 12 (for R+) cells, intensity ratio values for $n_f$ = 5 frames were plotted ($n_c$: number of cells; $n_f$: number of frames); *p* values (two-sided, by Mann–Whitney-Wilcoxon test): n.s.: 0.1391, ****: ≤0.0001. Boxes extend from the 25th to 75th percentiles, median is at the center, and whiskers and outliers are graphed as per Tukey's convention (as computed by Graphpad Prism). Source data are provided as a Source Data file.

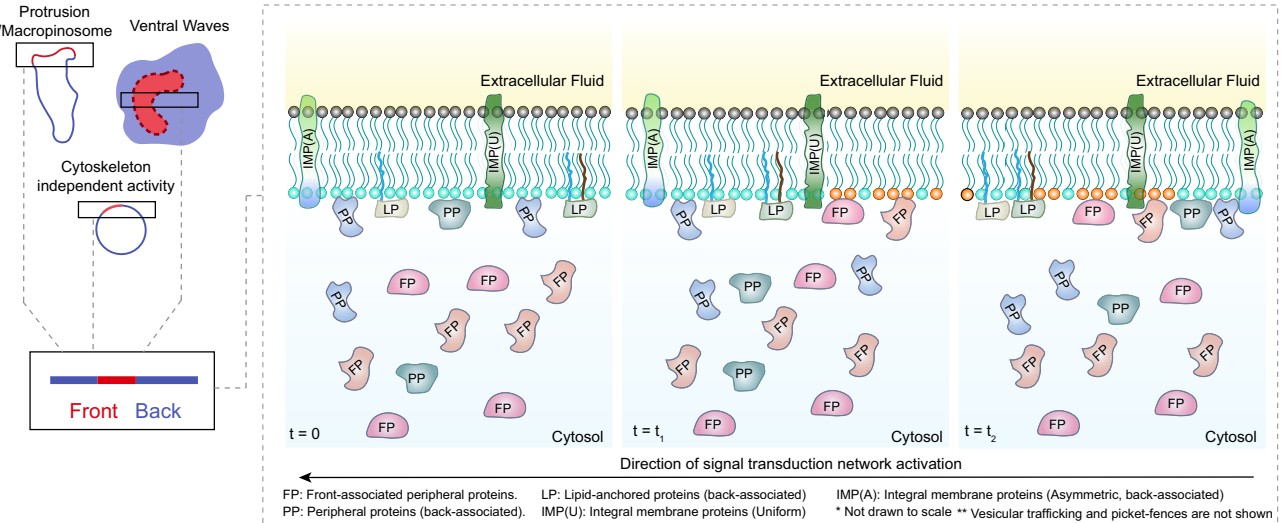

**Fig. 8 | Schematic illustration showing the effect of dynamic partitioning and shuttling in plasma membrane organization.** Note that both front- (FP) and back-associated (PP) peripheral membrane proteins, can shuttle on and off from membrane since their membrane binding is weaker. However, back-associated lipid-anchored proteins (LP), which cannot translocate to and from cytosol, changes its diffusion profile to move faster inside the front-states of the membrane, to exhibit polarized distribution. Similar partitioning can drive symmetry breaking of integral

membrane proteins (IMP(A)) or tightly-bound peripheral membrane proteins (not shown) as well. The headgroups of the inner leaflet lipid molecules that are enriched in front-state (such as PIP3, DAG, etc.) are shown in orange. The headgroups of inner leaflet lipid molecules that are enriched in back or basal-state (such as PI(4,5) P2, PI(3,4)P2, PS, PA, etc.) are shown in cyan. At time t=0, the entire membrane is in the resting/basal/back state. At $t = t_1$, signaling network activation was started at the right end which propagated to left at $t = t_2$.

result in altered mobility in different compartments. However, these mechanisms can contribute to dynamic partitioning that we suggested here, only if the complex formation processes were switched on and off in large micron-scale regions on the membrane quickly, in a tightly orchestrated fashion. Incidentally, although we are primarily reporting on the lipid-anchored proteins here, transmembrane proteins, or tightly bound peripheral proteins, might also be patterned by dynamic portioning. In fact, we were able to pattern the G-protein coupled chemoattractant receptor, cAR1, by recruiting a fragment of a back-seeking protein, to its cytoplasmic segment.

The observation that these lipidated proteins conform to the protrusion dynamics or propagating waves patterns was surprising since most of these proteins were repeatedly reported to be uniformly distributed over membrane in earlier literature[29,72–79]. Our study now has revealed a new mechanism of compartmentalization, but at this point we can only speculate on the function of the asymmetry. Most of the proteins which dynamically associate with the back-state, such as *Gβγ*, PKBR1, and RasG are paradoxically activated in front-state regions during chemotaxis[28,29,59,78,99–102]. While the G-protein activation is modest, resembling the external gradient, the activations of PKBR1 and RasG are amplified within the cell compared the external gradient. The activated proteins are not merely swept to the back since the

redistribution by dynamic partitioning occurs unabated in the absence of cell movement or cytoskeletal activity. It is possible that the movement to the back serves to counteract activation at the front, thereby controlling activity. Taking this speculation a bit further, perhaps activation leads to complex formation, as suggested earlier, which then causes the proteins to drift to the back as they become inactive. Further investigation will be needed to determine the true purpose of these dramatic redistributions.

The fluid mosaic model of the plasma membrane has been a powerful guiding premise for over 40 years. The original concept envisioned a bilayer "sea" in which integral membrane proteins could diffuse homogeneously and serve as binding sites for peripheral proteins. The changes produced on the membrane by the protrusions and propagating waves of signaling and cytoskeletal events suggest that the fluid mosaic "sea" is dynamically divided into extremely broad complementary regions, which segregate activities. The differences are defined by changes on the inner leaflet of the bilayer. These comprise the actions of differentially bound membrane receptors, selectively activated or inactivated G-proteins and kinases, markedly different lipid headgroups, and as described here, differential diffusion of lipid-anchored and integral membrane proteins. Remarkably, all these enzymatic actions and membrane organizations exhibit tight

spatiotemporal coordination, even in the absence of external cues or cytoskeletal scaffolding. When cells does experience an external stimulus or undergoes through a specific developmental programming, cell essentially just align these actions to respond correctly, as observed in case of stable front-state and back-state formation during polarized cell migration towards an chemotactic or galvanotactic gradient. It is reasonable to assume that these dynamic partitioning events in plasma membrane can contribute to other functions in numerous physiological processes in different types of cells where membrane gets compartmentalized.

## Methods

### Cell culture

The wild-type *Dictyostelium discoideum* cells of axenic strain AX2 (obtained from lab stock; cells were originally obtained from R R Kay laboratory, MRC Laboratory of Molecular Biology, UK) as well as $G\beta^-$ *Dictyostelium* cells (previously generated in our lab[72,103,104]) were cultured in standard HL-5 media supplemented with penicillin and streptomycin at 22 °C. To maintain stable expression of different constructs, Hygromycin (50 μg/mL) and/or G418 (30 μg/mL) and/or Blasticidin (15 μg/mL) were added to the media as per the resistance of the vectors containing genes of interest. Cells were subcultured after every 2–5 days using proper techniques to maintain a healthy confluency of 70–90%. Cells were usually maintained in adherent culture on petri dishes and they were transferred to a shaking culture (~200 rpm speed) for ~3–7 days before electrofusion or development experiments. All the experiments were performed within 1 month of thawing the cells from the frozen stocks.

HL-60 human neutrophil-like cells were obtained from O D Weiner laboratory (University of California San Francisco) and cultured in RPMI 1640 medium with L-glutamine and 25 mM HEPES (ThermoFisher Scientific; 22400089), supplemented with 15% heat-inactivated fetal bovine serum (ThermoFisher Scientific; 16140071) and 1% penicillin-streptomycin (ThermoFisher Scientific; 15140122). Cells were passaged upon reaching a density of $1–2 \times 10^6$ cells/mL and were subcultured at a density of $0.15 \times 10^6$ cells/mL. Approximately after every 3 days, cells were subcultured using standard technique. To differentiate the HL-60 cells into neutrophils, 1.3% DMSO was added to cells (which were maintained at a density $0.15 \times 10^6$ cells/mL) and cells were incubated for 6-8 days before nucleofection and subsequent microscopy.

RAW 264.7 macrophages-like cells were obtained from N Gautam laboratory (Washington University School of Medicine in St. Louis) who initially obtained from the Washington University Tissue Culture Support Center and American Type Culture Collection (ATCC, TIB-71). Cells were cultured in DMEM that contains 4500 mg/l glucose L-glutamine, sodium pyruvate, and sodium bicarbonate (Sigma-Aldrich, D6429), supplemented with 10% heat-inactivated fetal bovine serum (ThermoFisher Scientific, 16140071) and 1% penicillin-streptomycin (ThermoFisher Scientific, 15140122). Upon reaching 70–90% confluency, adherent cells were gently lifted using cell scrapers and subcultured using 1:5-1:10 split ratio. All neutrophil and macrophage cells were maintained under humidified conditions at 37 °C and 5% $CO_2$ and all experiments were performed using low passage number cells.

### DNA constructs

The constructs of GFP-R(+8)-Pre and CRY2PHR-mCherry-R+ (*Dictyostelium* and mammalian) were generated by annealing the forward and reverse pairs of appropriate synthetic oligonucleotides, followed by restriction enzyme mediated digestion and subcloning into proper *Dictyostelium* or mammalian expression vectors. All other constructs were made by PCR amplification of appropriate ORFs, followed by standard restriction enzyme-based subcloning to enable integration into suitable vectors. All oligonucleotides were acquired from Sigma-Aldrich. All the sequences were verified by the diagnostic restriction digests and by standard Sanger sequencing (JHMI Synthesis

& Sequencing Facility). The following plasmid constructs were made in this study. Selected plasmids will be deposited on dictyBase/Dicty Stock Center[105,106] and/or Addgene and rest will be available from the authors upon direct request: (a) PKBR1-KikGR (pDM358), (b) *PKBR*1$_{N150}$-KikGR (pDM358), (c) KikGR-G$\beta$ (pDM358), (d) PTEN-KikGR (KF3), (e) GFP-R(+8)-Pre (pDM358) which was also used in[9], (f) PKBR1-HaloTag (HK12neo), (g) CRY2PHR-mCherry-R+ (pCV5), (h) CRY2PHR-mCherry-R+ (pmCherryN1, mammalian) (i) cAR1-CIBN (pDM358). GFP-RasG (pDEXB) and Lifeact-Dendra2 (pDEXB) were kind gifts from A. Müller-Taubenberger (LMU Munich). The pCRY2PHR-mCherryN1 (Addgene Plasmid #26866) was from C. Tucker and Lyn11-CIBN-GFP (Addgene Plasmid #79572) was obtained from P. De Camilli and O. Idevall-Hagren. GFP-R(+8)-Pre (mammalian) was from S. Grinstein (Addgene, plasmid 17274). Following plasmids used in this study were obtained from the Devreotes or Ueda Lab stock: (a) PHCrac-mCherry (pDM358), (b) PHCrac-RFP (pDRH), (c)PHAkt-mCherry (mammalian), (d) RBD-YFP (pCV5), (e) RBD-RFP (pDM 181), (f) LimE$_{\Delta coil}$-mCherry (pDM181), (g) CynA-KikGR (KF2), (h) PTEN-YFP (pCV5).

### Drugs and reagents

F-actin polymerization inhibitor Latrunculin A (Enzo Life Sciences; BML-T119-0100) was dissolved in DMSO to prepare a stock solution of 5 mM. Caffeine (Sigma-Aldrich; C0750) was dissolved in ddH2O to result a stock solution of 80 mM. cAMP (Sigma-Aldrich; A6885) was dissolved in ddH2O to make a stock solution of 10 mM. LY294002 (ThermoFisher Scientific; PHZ1144) was dissolved in DMSO to prepare a stock solution of 40 mM. InSolution Y-27632 (688001; Calbiochem) was obtained from Sigma-Aldrich. The anti-BSA mouse monoclonal antibody was acquired from Sigma-Aldrich (SAB4200688, clone BSA-33). The C5a receptor agonist FKP-(D-Cha)-Cha-r (Anaspec; 65121) was dissolved in 1X PBS to prepare a 2.5 mM of stock solution. TMR-Halo-ligand (G8251; Promega) and Janelia Fluor HaloTag Ligands (GA1120; Promega) were dissolved in DMSO to prepare a stock solution of 200 μM which was stored at 4 °C and they were diluted 1000X in DB buffer before the experiments. Fibronectin (Sigma-Aldrich; F4759) was dissolved in 2mL sterile ddH2O and then 8 mL $Ca^{2+}/Mg^{2+}$-free PBS solution was added to it to prepare a stock solution of 200 $\mu g$/mL which was stored at 4 °C. The formylated Methione-Leucine-Phenylalanine or fMLP (Sigma-Aldrich; F3506) was dissolved in DMSO to make 10 mM stock solution. Unless otherwise mentioned, everything was stored as small aliquots at −20 °C.

### Transfection

AX2 and G$\beta^-$ *Dictyostelium* cells were transfected as per standard electroporation protocol. Briefly, $5 \times 10^6$ Ax2 cells were collected from the shaking culture and pelleted for each trasnfection. Then the cells were washed twice with ice-cold H-50 buffer (20 mM HEPES, 50 mM KCl, 10 mM NaCl, 1 mM MgSO4, 5 mM NaHCO3, 1 mM NaH2PO4, pH adjusted to 7.0). Subsequently, cells were resuspended in 100 μL ice-cold H-50 buffer, around 1–5 μg of each DNA species was added to it, and quickly transferred to an ice-cold 0.1cm gap cuvette (Bio-Rad, 1652089). Cells were then electroporated for two times at 0.85 kV voltage and 25 $\mu F$ capacitance, with a 5s interval between pulses (using Bio-Rad Gene Pulser Xcell Electroporation Systems). Next, the electroporated cells were incubated in ice for 5 min and then they were transferred from the cuvette to a 10-cm petri dish with 10 mL of HL-5 medium, supplemented with heat-killed *Klebsiella aerogenes* bacteria. After 1–2 days of recovery, drugs were added for antibiotic selection, as per the resistance of the vectors that contains genes of interest.

### Nucleofection in macrophages and global receptor activation assay

The RAW 264.7 macrophage cells were nucleofected with Amaxa Nucleofector II device and Amaxa Cell line kit V (Lonza, VACA-1003), by primarily following an existing protocol[9,107]. To perform each

nucleofection, 3 million cells were harvested and combined with 100 µL of supplemented Nucleofector Solution V and the appropriate amount of DNA (usually around 2 µg of each DNA construct). The combined solution is then promptly transferred to a Lonza cuvette and cells were electroporated with program D-032. Then 200 µL of pH and temperature adjusted culture medium was added to the cuvette. The nucleofected cells, along with the media, were then transferred to an eppendorf tube that contains 300 µL of pH and temperature adjusted culture medium. Cells were incubated at 37°C and 5% $CO_2$ for 10 min, keeping the tube uncapped. Next, 70–100 µL of these cells were added to each well of an 8-well Nunc Lab-Tek chambers and inubated for another 1 h. After cells have adhered to the substrate, 400 µL of pH and temperature adjusted culture medium was added to each well and cells were incubated for 4–6 h. The culture media was then replaced with 450 µL of 1g/L glucose-supplemented Hank's balanced salt solution (HBSS buffer). Cells were additionally incubated for 20–40 min before starting the image acquisition experiment. After acquiring images for 4–5 min, 10 µM (final concentration) of C5aR receptor agonist FKP-(D-Cha-r)-Cha-r was gently added to each well and image acquisition was continued, where 12 sec/frame imaging frequency was maintained throughout the experiment.

### Nucleofection and cell migration assay for neutrophils

Differentiated HL-60 cells were nucleofected with Amaxa Nucleofector II device and Amaxa Cell line kit V (Lonza, VACA-1003) and prepared for live-cell imaging using a slightly modified version of an existing protocol[108]. Briefly, $5 \times 10^6$ differentiated HL-60 cells were harvested from the suspension culture for each nucleofection and after removing the media, cells were resuspended in 100 µL supplemented Nucleofector Solution V. A total of ~1–1.5 µg of DNA mixture was added to it and everything was quickly transferred to a Lonza cuvette. Cells were electroporated using the program setting Y-001. Next, ~500 µL of recovery medium (IMDM with L-Glutamine and HEPES (Lonza; 12-722F), supplemented with 20% FBS, and equilibrated at 37 °C and 5% $CO_2$) was added immediately to the cuvette and the entire solution was transferred to an eppendorf tube. After 30 min incubation at 37 °C and 5% $CO_2$, the eppendorf tube was taken out and ~500 µL of cells were transferred to 1.5 mL of recovery medium in a 6-well plate. Subsequently, after 3–4 h, ~100–150 µL of nucleofected cells were added to an 8-well Nunc Lab-Tek chambers (which were pre-coated with 125 µL of fibronectin, as prepared earlier, for 1.5-2 hours and washed with RPMI culture media). Then the cells were incubated in chamber wells for 15 min, the media was aspirated, and fresh culture media was added. Before starting the optogenetics experiments, the cells were stimulated with 100 nM (final concentartion) fMLP and then were allowed to polarize for 15 more min. For ROCK inhibition experiments, cells were treated with 10 µM Y-27632 for 15 min before starting the image acquisition.

### Microscopy

All *Dictyostelium* experiments were performed on a 22 °C stage. All neutrophil experiments were performed inside a 37 °C chamber with 5% $CO_2$ supply. All time-lapse live-cell imaging experiments were performed using one of the following microscopes: Zeiss LSM 780-FCS Single-point, laser scanning confocal microscope (Zeiss Axio Observer with 780-Quasar; 34-channel spectral, high-sensitivity gallium arsenide phosphide detectors), (b) Zeiss LSM880-Airyscan FAST Super-Resolution Single-point confocal microscope (Zeiss AxioObserver with 880-Quasar (34-channel spectral, high-sensitivity gallium-arsenide phosphide detectors), c) Zeiss LSM800 GaAsP Single-point, laser scanning confocal microscope with wide-field camera, and (d) Nikon Eclipse Ti-E dSTROM Total Internal Reflection Fluorescence (TIRF) Microscope (Images were obtained using Photometrics Evolve EMCCD camera). The Zeiss 780 and Zeiss 880 Airyscan confocal microscopes were controlled using ZEN Black software, Zeiss 800

confocal microscope was controlled using ZEN Blue software, whereas Nikon TIRF was operated using NIS-Elements software. The 40X/1.30 Plan-Neofluar oil objective (with proper digital zoom) was used in Zeiss 780, 800, and 880 confocal microscopes, whereas 100x/1.4 Plan-Apo oil objective was used in Nikon TIRF. The 488 nm (Ar laser) excitation was used for GFP and YFP, whereas 561 nm (solid-state) excitation was used for RFP and mCherry in Zeiss 780 and 800 confocal microscopes. In case of Zeiss 880 Airyscan confocal microscope, 488 nm (argon laser) excitation was used for GFP, 514 nm (Ar laser) was used to excite YFP, whereas 594 nm (HeNe laser) excitation was used for mCherry. The 639 nm (diode laser) was used in Zeiss 780 confocal microscope to excite Janelia Fluor HaloTag 646. The 488nm (Ar laser) excitation was used for GFP and 561 nm (0.5W fiber laser) excitation was used for mCherry and RFP in Nikon TIRF.

### Frustrated phagocytosis and osmotic shock

To record the ventral wave propagation on the substarte-attached surface of RAW 264.7 cells, we have adapted a pre-existing protocol[9,55]. First, 8-well nunc Lab-Tek chambers were washed with 30% nitric acid and then coated with 1 mg/mL BSA (in 1X PBS) for 3 hours. Then the chamber coverslips were incubated with 5 µg/mL (1:200) anti-BSA antibody for 2 h. After incubation, chamber coverslips were washed twice with 1X PBS to remove excess antibodies. The nucleofected RAW 264.7 cells were collected (as described in "Nucleofection in macro-phages and global receptor activation assay" section) and after 4 h incubation in culture medium, cells were lifted from a 6-well plate to starve in suspension in 1X Ringer's buffer (150 mM NaCl, 5 mM KCl, 1 mM CaCl2, 1 mM MgCl2, 20 mM HEPES and 2g/L glucose, pH 7.4) for 30 min. Next the cells were allowed to spread over the opsonized coverslip chambers for 5 min and then hypotonic shock was applied using 0.5X Ringer's solution. For ROCK inhibition experiments, cells were exposed to 50 µM of Y-27632 while in suspension. For Supplementary Fig. 7b, 5 µM LatA was also added while cells were in suspension. For Supplementary Fig. 7c, the ROCK inhibited cells were allowed to perform frustrated phagocytosis over the opsonized coverslips first, then osmotic shock was applied, and then 5 µM LatA was added and incubated for 20 min before starting image acquisition (the Y-27632 concentration of 50 µM was maintained throughout).

### Electrofusion

Total $1.5 \times 10^8$ growth phase *Dictyostelium* cells were first harvested from shaking culture. Cells were then washed two times and resuspended in 10 mL SB (17 mM Soerensen buffer, 15 mM KH2PO4 and 2 mM Na2HPO4, pH 6.0). Next, the cells were put inside a conical tube and gently rolled for 30-40 min to induce cluster formation. Subsequently, 800 µL of rolled cells were transferred to a 0.4cm gap Bio-Rad cuvette, using pipette tips with cut off edges (to ensure clusters remain intact). Using a BioRad Gene Pulser (Model 1652098), the electroporation was performed at 1kV, 3 µF once, then with 1kV, 1 µF twice more to induce hemifusion[45]. A 3s time interval was maintained between two pulses. Next, ~35 µL of electrfused cells were taken from the cuvette and transferred to a Nunc Lab-Tek 8 well chamber. Cells were incubated for 5 min before adding 450 µL of SB buffer supplemented with 2 mM CaCl2 and 2 mM MgCl2. Cells were then allowed to recover, settle and adhere to substrate for ~1 h before starting the image acquisition.

### Live-cell imaging of subcellular symmetry breaking in different modes

To capture the ventral wave dynamics at the substrate attached surface of cell membrane, the electrofused "giant" *Dictyostelium* cells were used (please see previous "Electrofusion" section for details). Images were captured at 7 sec/frame, using either TIRF microscope, or using confocal laser scanning microscopes focusing at the very bottom surfaces of the cells. To capture protrusion dynamics at the randomly

migrating cells, growth phase *Dictyostelium* cells were transferred to an 8-well Nunc Lab-Tek coverslip chamber and allowed to adhere for ~15 min. In the next step, the HL-5 medium was aspirated and 450 µL of fresh DB buffer (Development buffer; 5 mM Na2HPO4 and 5 mM KH2PO4 supplemented with 2 mM MgSO4 and 0.2 mM CaCl2, pH 6.5) was added to the cells. Cells were incubated at 22 °C for around 45–60 min before staring the image acquisition in one of the confocal microscopes, at an imaging frequency of 5 sec/frame. To visualize cytoskeleton-independent symmetry breaking dynamics of signaling components, growth phase single *Dictyostelium* cells were prepared in an 8-well chamber as described, but they were incubated in DB for longer time (more than 2.5 h). For final 30 min, DB buffer was supplemented with Caffeine (final concentration 4 mM) which increases the symmetry breaking events on cell membrane[9,26,32]. Before starting the image acquisition, Latrunculin A was added to a final concentration of 5 µM. Cells were incubated in presence of Latrunculin A for around 20–25 min. To record protrusion and cytoskeleton-independent signaling activities, unlike ventral wave experiments, confocal laser scanning microscopes were focused in the middle z-planes of the cell.

## Cell differentiation and chemotaxis assay
For development of *Dictyostelium* cells, we used a previously established protocol[109]. Briefly, $8 \times 10^7$ cells of growth phase cells were collected from shaking culture and pelleted. The cells were washed twice with DB buffer and resuspended in 4 mL DB in a conical flask. It was shaken at 110 rpm for 1 h. After 1 h, the cells were pulsed with 50-100 nM of cAMP at a rate of 5 s pulse every 6 min, using a time-controlled programmed peristaltic pump for next 5–6 h, while the shaking was continued. Around 10–15 µL of differentiated cells (in DB media) was collected and transferred to a 1-well Nunc Lab-Tek chamber. Then ~2 mL DB buffer was added to the chamber and cells were dispersed by pipetting multiple times. A Femtotip microinjection needle (Eppendorf) was loaded with 10 µM of filtered cAMP solution and it was then connected to a FemtoJet microinjector (Eppendorf). The microinjector was operated in continuous injection mode with a compensation pressure of 15 hPa. To initiate the chemotaxis, the micropipette was brought to the (x,y,z) coordinate of cells using a programmed micromanipulator. The imaging was continued with a acquisition frequency of 10 sec/frame.

## Global receptor activation assay in *Dictyostelium*
After the *Dictyostelium* cells were properly differentiated (as described in previous section), around $5 \times 10^5$ cells were transferred to an 8-well coverslip chamber. Around 450 $\mu$L of fresh DB was added and cells were resuspended thoroughly to break the clusters. Then, the cells were incubated for around 20 min at 22 °C. Cells were subsequently incubated with 5 µM Latrunculin A for around 25 min before starting image acquisition for the global stimulation experiment. Using a confocal laser scanning microscope, a few frames were first acquired to record the basal activity of the proteins, then cAMP was added to the chamber (to a final concentration of 10 µM) to activate all the cAR1 receptors, and the image acquisition was continued. An imaging frequency of 2.5 sec/frame was maintained throughout the experiment.

## Photoconversion and protein movement assay
The photoconversion experiments were performed in a Zeiss LSM 780-FCS Single-point, laser scanning confocal microscope, with a frame rate 7 sec/frame. *Dictyostelium* cells expressing photoconvertible protein (Dendra2 or mKikGR) fused with proteins of interest (PKBR1, G$\beta$, CynA, or PTEN) were first electrofused. After settling, recovery, and adherence, electrofused cells were imaged using 488 nm (Argon) laser, by focusing the confocal microscope at the very bottom of the cell, for 5–10 frames, to visualize the wave dynamics and to determine the direction of wave propagation. After that, an area of photoconversion was drawn right in front of one the shadow wave regions (which shows

the activated/"front"-state regions of the membrane), in the direction of wave propagation, using the "region" module of Zeiss Black. Next, that particular area was photoconverted with 405 nm (diode) laser using the 'bleaching' module, usually utilizing 1–2% laser power. After single iteration, 405 nm laser was turned off and the photoconverted molecules were tracked for next 100-120 s. Throughout the experiment, both green and red channels were imaged simultaneously using proper microscope beam splitters and filters.

## Optogenetic manipulation of membrane binding affinities
Optogenetics experiments were performed using slightly modified protocols that we described in details earlier for *Dictyostelium* and HL-60 cells[9,110–112]. Briefly, differentiated and nucleofected HL-60 cells were collected as described in "Nucleofection and cell migration assay for neutrophils" section. The *Dictyostelium* cells were selected against both G418 as well as hygromycin to co-express cAR1-CIBN (pDM358), CRY2PHR-mCherry-R+ (pCV5) / CRY2PHR-mCherry (pCV5), along with LimE$_{\Delta coil}$-Halo (pCV5) / PH$_{Crac}$-YFP (pCV5). To visualize protrusion dynamics, normal growth phase cells were used whereas to visualize ventral wave dynamics, electrofused cells were used, and Zeiss LSM 780-FCS microscope was focused accordingly (please see 'Live-cell imaging of subcellular symmetry breaking in different modes' section for details). In the beginning, a few frames were acquired using 561 nm and 639 nm laser to visualize the normal cytosolic dynamics of CRY2PHR-mCherry-R+ and F-actin polymerization, respectively. Next, 488 nm laser was turned ON globally to recruit the CRY2PHR fused cytosolic protein to the membrane using CIBN-fused cAR1 (for *Dictyostelium*) or CIBN-fused Lyn11 (for HL-60) and then the image acquisition was continued. The 488 nm laser was intermittently turned on (for around 950 ms after each 5–8s) to maintain the optogenetic recruitment throughout the time period of experiment.

## Cell preparation for single-molecule imaging
Cultured cells were washed twice with DB buffer by centrifugation ($500 \times g$, 2 min) and resuspended in DB at a cell density of $3 \times 10^6$ cells/mL. In total, 1 mL of cell suspension was transferred to a 35-mm culture dish and incubated for 3–4 h at 21 °C. To observe PKBR1-Halo, Halo-Tag® TMR ligand (Promega) was added to the cell suspension at the final concentration of 1 nM during the last 30 min. The cells were washed twice with DB by centrifugation and suspended in DB at around $5 \times 10^6$ cells/mL. A 5 µL cell suspension was placed on a coverslip (25 mm diameter, 0.12–0.17 mm thick; Matsunami) that was washed by sonication in 0.1 N KOH for 30 min and rinsed with 100% ethanol prior to use. After a 10 min incubation, the cells were overlaid with an agarose sheet (5 mm × 5 mm, Agarose-II; Dojindo). After 20 min of incubation, the coverslip was set in an Attofluor™ cell chamber (Invitrogen) and observed by TIRF Microscope.

## Microscopy setup for single-molecule imaging
Objective-type TIRFM was constructed on an inverted fluorescence microscope (Ti; Nikon) equipped with two EM-CCD cameras (iXon3; Andor) for the detection of TMR and GFP signals separately[53]. They were excited with solid-state CW lasers (OBIS 488-150 LS and Compass 561-20; Coherent), which were guided to the back focal plane of the objective lens (CFI Apo TIRF 60X Oil, N.A. 1.49; Nikon) through a back port of the microscope. The excitation lights were passed through a dual-band bandpass filter (FF01-482/563-25; Semrock) and reflected by a dichroic beam splitter (Di01-R488/561-25 × 36; Semrock). The emission lights were passed through the dichroic beam splitter, separated by another dichroic beam splitter (Di01-R561-25 × 36; Semrock), passed through single-band bandpass filters (FF01-525/45-25 and FF01-609/54-25; Semrock) and 4 × intermediate magnification lenses (VM Lens C-4 × ; Nikon) before the detection with the cameras. The PKBR1-Halo and PHD-GFP images were acquired at 30 and 1 frames/s, respectively, with a software (iQ2; Andor).

## Image analysis

Most of the image processings were performed in MATLAB 2021a (MathWorks) and Fiji/ImageJ 1.53q (NIH) (with occasional use of iLastik 1.3.3post3 for segmentation). The results were plotted using MATLAB 2021a, OriginPro 9.0 (OriginLab) or GraphPad Prism 8 (GraphPad Software).

A. *Cell segmentation:* For most of the image analysis, cell segmentation was performed in the first step, in either MATLAB or Fiji/ImageJ. To segment in Fiji/ImageJ, first "*Threshold*" command was used to generate a binary image consisting of all the pixels of the cells ('*Don't reset range*' was checked and '*Calculate threshold for each image*' option was unchecked). '*Analyze Particles*' module was used to perform a size-based thresholding which excluded all non-cell particles. Next, different morphological operations, such as '*Fill holes*', '*Erode*' and '*Dilate*' options were applied judiciously (often multiple times), to obtain correct binarized masks of the cells. Overlapping cells were separated during segmentation either manually selecting ROIs or by *Trainable Weka Segmentation* plugin. To perform cell segmentation in MATLAB, first a user-defined ROI was selected using *roipoly* function which excluded any overlapping cell regions. Next, image was preprocessed first by performing morphological top-hat filtering, using proper structural elements. Next, image was processed by background subtraction, top-hat filtering, and Gaussian smoothing. Thresholds were checked using *multithresh* command and then cells were threholded and binarized. Small non-cell particles were removed by *bwareaopen* function and next proper morphological operations, such as *imfill*, *imdilate*, and *imerode* were applied judiciously. The holes in the cells were removed using a custom-written code involving *regionprops*, *imcrop*, *bwboundaries*, *polyarea*, and *poly2mask* functions. This finally generated the binarized mask of the cells.

B. *Colocalization study*: First, cells were segmented, either in MATLAB or in Fiji/ImageJ, as described above, to generate binary images of 16-bit unsigned integer arrays and subsequent processing was performed in MATLAB. The original images consisting of $PH_{Crac}$ channel and second marker channel were smoothed using Gaussian filtering. The $PH_{Crac}$ channel and second marker channel intensities of cell regions were selected by employing the *find* function (and by utilizing the previously generated masks) to exclude the background areas from the analysis. Colocalization study between two channels were performed by using *corrcoef* function of MATLAB, which determines the Pearson correlation coefficient (r) as follows:

$$r(A,B) = \frac{1}{N-1} \sum_{i=1}^{N} \left( \frac{A_i - \mu_A}{\sigma_A} \right) \left( \frac{B_i - \mu_B}{\sigma_B} \right)$$

where $\mu_A$ and $\sigma_A$ denotes mean and SD of A, respectively. Similar notation is true for B as well. Each variable has N observations.

Finally, the Pearson's r values for different cells for particular number of frames were plotted using *heatmap* function and 'parula' colomap was chosen where blue denotes anti-correlation and yellow denotes positive correlation. The Supplementary Fig. 1g inset scatterplot was generated using *heatmap_scatter* function version 1.1.1 from MATLAB Central File Exchange.

C. *Optical flow analysis*: The optical flow analysis[65,66,113] was performed with a custom-written program using the *Computer Vision Toolbox* (Mathworks) inside MATLAB 2021a. Briefly, first, the cell area was segmented as described above. Next, using similar top-hat filtering, Gaussian smoothing, and setting a proper threshold, along with judicious use of *imdilate, imerode, infill, bwareaopen, imcrop, bwboundaries*, and *poly2mask* functions, the shadow wave

areas (SW) and photoconverted areas (PC) were segmented and binarized. Next, the center and radius ($r_m$) of a minimal bounding circle was computed around PC (using *minboundcircle* function of '*A suite of minimal bounding objects*' 1.2.0.0 from MATLAB Central File Exchange). Then the shadow waves inside a circular area having the same center and a radius $1.2r_m$ - $1.3r_m$ was selected for tracking optical flow (since shadow waves which are further away possibly had insignificant effect on the movement of photoconversion area molecules). Intensity over different frames were calculated using a custom written program and plotted to generate Fig. 4e. To analyze whether the SW and PC moved together throughout the time period of the experiment, *opticalFlowHS* and *estimateFlow* functions of *Computer Vision Toolbox* was used for each of those. This solves for x-direction velocity $u$ and y-direction velocity $v$, in equation:

$$I(x,y,t) = I(x + \Delta x, y + \Delta y, t + \Delta t)$$

where, I(x,y,t) is the intensity at time frame t. Essentially, these programs employed Horn-Schunck method[65] to compute local flow driven transport between two frames. Horn-Schunck method effectively computes the velocity field for each pixel in the image, [u v], by Sobel convolution kernel, which minimizes the following equation:

$$E = \iint (I_x u + I_y v + I_t)^2 dxdy$$
$$+ \alpha \iint \left\{ \left( \frac{\partial u}{\partial x} \right)^2 + \left( \frac{\partial u}{\partial y} \right)^2 + \left( \frac{\partial v}{\partial x} \right)^2 + \left( \frac{\partial v}{\partial y} \right)^2 \right\} dxdy$$

where $\alpha$ is smoothness factor. After obtaining optical flow velocity vectors of SW and PC for each frame, the resultant vectors for SW and PC were computed. Then their dot products were computed to obtain the angel between them. All these values over different frames and different cells were plotted in polar histograms using *polarhistogram* function. Minimum number of bins were decided based on Sturges' formula. To generate flow vector diagrams, *plot* command of *Computer vision toolbox* was used while 'DecimationFactor' of [8 8] and 'ScaleFactor' of 60 was specified.

D. *Kymographs*: To generate line kymographs that accompanied ventral waves, a thick line having a width of 10-12 pixels were drawn in in Fiji/ImageJ and the entire stack was processed using the '*KymographBuilder*' plugin.

The process of generating membrane kymographs in MATLAB were described earlier[9,25]. Briefly, the cells were first segmented as described above. The kymographs were generated by linearizing the boundaries and stacking intensities over the boundaries for each frame. Average of top five brightest pixel along the perpendicular lines across the boundary was selected as membrane intensity. The consecutive lines over time were aligned by minimizing the sum of the Euclidean distances between the coordinates in two adjacent frames using a custom-written MATLAB program. For the first frame, it was realigned so that the desired angle corresponds roughly to the point that is at the center of the kymograph. For other frames, it was aligned to the points are closest to the previous frame, relative to the centroid. A linear colormap ('Turbo') was used for the normalized intensities in the kymographs.

E. *Linescan intensity profile*: Linescan intensity profiles accompanying ventral waves were obtained from Fiji/ImageJ. A thick line of 7-10 pixels were drawn (as shown in the figures) and using "*Plot Profile*" option, intensity values were obtained. The values were then imported to OriginPro 9.0 (OriginLab) and normalized. The intensity profiles were plotted first and then

smoothened using the Adjacent-Averaging method of OriginPro and by selecting proper boundary conditions. For a specific line scan, the green and red intensities were processed using the exact same parameters to maintain consistency.

F. *Time-series plots of cytosolic intensity*: To obtain time-series plots of cytosolic intensities (Fig. 3), first cells were segmented as described above. Next, it was eroded three times in Fiji/ImageJ to exclude the membrane and generate the binarized cytosolic mask. Next, using a custom-written macro, the cytosolic mask stacks were processed using *"Create Selection"* and *roiManager("Add")* commands. Subsequently, using those ROIs, the green and red channel cytosolic intensities were obtained using *"Measure"* options for all frames. Intensities were normalized by dividing by mean of intensity values in the frames before cAMP or C5aR agonist addition. Mean and SEM values of normalized intensities were then plotted in Graphpad Prism.

G. *Analysis of single-molecule imaging data*: The x- and y-coordinates of individual single molecules were determined semi-automatically using a laboratory-made software. The methods for the statistical analysis of the lateral diffusion and membrane-binding lifetime are described elsewhere[114,115]. Briefly, the lateral diffusion coefficient was estimated from the statistical distribution of the displacement, $\Delta r$, that a single molecule moved during a time interval, $\Delta t = 33.3$ ms[114]. The distribution was fitted to the following probability density function,

$$P_m(\Delta r) = \sum_{h=1}^{m} q_h \frac{\Delta r}{2D_h \Delta t + 2\epsilon^2} e^{\left(\frac{-\Delta r^2}{4D_h \Delta t + 4\epsilon^2}\right)}$$

where $D_h$, $q_h$, m, and $\epsilon$ denote the diffusion coefficient and fraction of the h-th mobility state, total number of mobility states, and standard deviation of the measurement error, respectively. The number of states was estimated using the Akaike Information Criterion (AIC).

Short-range diffusion analysis was performed as follows[68,115]. From a trajectory of the $i-th$ molecule, $(X_i(t), Y_i(t))$, where $t = 0, \Delta, \Delta t, 2\Delta t, ..., T_i \Delta t$, $T_i - 14$ fragments with a time duration of 0.5 s were extracted successively. For each fragmented trajectory, $(X_i(T), Y_i(T))$, where $T = 0, \Delta t, 2\Delta t, ..., 15\Delta t$, mean-squared displacement (MSD) was calculated as,

$$MSD(l\Delta t) = \overline{(X_l(T + l\Delta t) - X_l(T))^2 + (Y_l(T + l\Delta t) - Y_l(T))^2},$$

where $l\Delta t$ denotes the lag-time ($l = 1, 2, ..., 15$). MSD($1\Delta t$) to MSD($4\Delta t$) were fitted with the linear function,

$$MSD(l\Delta t) = 4D_{SRD} l\Delta t + c,$$

where $D_{SRD}$ is the short-range diffusion coefficient of the fragmented trajectory and c is a constant representing the measurement error. The distribution of $D_{SRD}$ of all fragments of all molecules was obtained.

The membrane-binding lifetime was quantified from the statistical distribution of the time a single molecule was detected on the membrane. The distribution in cumulative form was fitted to an exponential function as,

$$F(t) = \sum_{j=1}^{n} a_j e^{-(k_j + k_b)t},$$

where $k_j$, $a_j$, n, and $k_b$ denote the decaying rate constant of the j-th binding state, the fraction of the j-th binding state, the total number of binding states, and the rate constant of photobleaching of the fluorophore, respectively. The inverse of $k_j$, $\tau_j = 1/k_j$, corresponds to the lifetime of the j-th state.

The fraction of the molecules that adopt the j-th state at an arbitrary time point was calculated as,

$$f_j = \frac{a_j \tau_j}{\sum_{j=1}^{n} a_j \tau_j},$$

whereas a mean of the membrane binding lifetime was calculated as,

$$\bar{\tau} = \sum_{j=1}^{n} f_j \tau_j.$$

## Computational modeling

All computational modeling was performed in MATLAB 2022a (MathWorks, Natick, MA, USA) on a macOS (version 12), using URDME package (version 1.4). Original URDME package development was described in refs. 116,117.

A. *Excitable signal transduction network*: The core of the excitable signal transduction network was modeled using three interacting species: *F* (front), *B* (back), and *R* (refractory)[9,28,31,71]. The first two represent compositional states of the membrane in which there is a preponderance of front-associated species (e.g. RasGTP, RapGTP, PIP3, etc.) and back-associated species (e.g. PTEN, PI(4,5)P2, myosin II, etc.). Note that at any point in the membrane surface, one of these states typically dominates (i.e. $F \gg B$, or $B \gg F$) but the states are not mathematically mutually exclusive. This is accomplished by assuming that the *B* and *F* species mutually inhibit each other, and is based on observations that the membrane tends to segregate into these regions as well as evidence that there is such mutual inhibition between Ras and anionic lipids[9,26,28,31,32,49,118]. The refractory species, *R*, denotes the element that provides negative feedback to the front state. Excitable systems are typically found in a basal, quiescent state. Following a suprathreshold stimulus, the system transitions to an excitable state which shows high level of activity. Following this excitable state comes a *refractory* period during which further excitation is not possible. In the context of our three-state model, the quiescent states would have high *B* and low *F* and *R* values, the excitable state would have high *F* with low *B* and *R*. Finally, the refractory period would have high *R* and *B* but low *F*.

Figure 6a illustrates the interactions between the three species: *F* activates *R*, while *R* inhibits *F* via a delayed negative feedback; *F* and *B* mutually inhibit each other, creating a autocatalytic loop effect (i.e., a double negative feedback loop[119]). The system dynamics can be described by following three partial differential equations denoting reaction-diffusion terms.

$$\frac{\partial F}{\partial t} = -(a_1 + a_2 R)F + \left(\frac{a_3}{a_4^2 B^2 + 1} + u_b\right)(a_5 - F) + D_F \nabla^2 F \quad (1)$$

$$\frac{\partial R}{\partial t} = -c_1 R + c_2 F + D_R \nabla^2 R \quad (2)$$

$$\frac{\partial B}{\partial t} = b_1 - b_2 B - b_3 FB + D_B \nabla^2 B \quad (3)$$

In all three equations, the last term represents the diffusion of the respective species following Fick's law of diffusion. We now describe the reaction terms.

The first term of Eq. (1) denotes the combination of the constitutive ($-a_1 F$) and *R*-mediated inactivation of *F*-molecules ($-a_2 RF$), respectively. The second term captures the increase in levels of *F*. These can be constitutively: $u_b$ (which can also be

influenced by chemical, electrical, or mechanical stimuli) as well as a term arising from the doubly-negative feedback autocatalytic loop: $\frac{a_3}{a_4^2 B^2 + 1}$. Note that in this term, increases in $B$ lower the effect of this production. These last two terms multiply by $(a_5 - F)$ representing a finite amount of $F$ molecules (e.g., if the variable $F$ represents the level of RasGTP, then the term $(a_5 - F)$ would be the concentration of RasGDP molecules, and $a_5$ is the total number of Ras molecules.). In Eq. (2), the first and second reaction terms denote the inactivation of $R$ and $F$-mediated activation of $R$, respectively. In Eq. (3), the terms denote basal activation of B ($b_1$), basal inactivation of B ($b_2B$), and $F$-mediated inhibition of B ($b_3FB$), respectively. To highlight the presence of the various loops, first note that increasing $F$ raises the level of $R$ (through the $+c_2F$ term in Eq. (2)) and this increases the inactivation of $F$ (through the $-a_2RF$ term in Eq. (1)), thus closing a negative feedback loop. Concomitantly, increasing $F$ lowers the level of $B$ (through the $-b_3FB$ term in Eq. (3)) and this increases the activation of $F$ (through the $\frac{a_3}{a_4^2 B^2 + 1}$ term in Eq. (1)), thus closing a positive feedback loop that comes about from two negative interactions.

B. *Incorporating Lipid-anchored and peripheral membrane protein dynamics into the computational model*: To simulate the dynamics of lipid-anchored proteins/integral membrane proteins and peripheral membrane proteins, we considered two additional species, LP which can break symmetry by dynamic partitioning and PP which can break symmetry by recurrent recruitment and release (see Fig. 6a).

From the single molecule experimental data, we found contributions of four different diffusion coefficients (Supplementary Table 2). From the AIC values in Supplementary Table 2, we can conclude that no more than four diffusion coefficients are required to model the diffusion in either front or back state and optimal model is among the models investigated. In developing the computational model, we sought a simpler description. To this end, we fitted Gaussian mixture models (GMM) with varying number of components ($n$) to the experimentally obtained histogram data of short range diffusion (SRD) coefficients (back and front data combined, from Fig. 5f). We found the corrected Akaike information criterion (AICc) value was smallest for $n = 3$ (see Supplementary Table 3). Since the change ($\Delta = \text{AICc}_{2-\text{comp}} - \text{AICc}_{3-\text{comp}}$) in the AICc value from 2-component model to 3-component model was minimal ($\Delta = -6.80$), we ended up considering two different diffusion coefficients—$D_f = 0.45\,\mu\text{m}^2/$ s (fast) and $D_s = 0.05\,\mu\text{m}^2/\text{s}$ (slow). To explain the fast and slow diffusion constants, we assumed that LP can exist in two forms, membrane unbound LP$_u$(u:unbound) which can diffuse faster over the membrane (but cannot go to cytosol) and membrane bound which, due to its association with the back (B) and front state molecules (F) diffuses much slower. Due to the association of LP with both front and back states, we assumed two factions of LP − front-bound F:LP and back-bound B:LP. To estimate the relative proportions of the membrane bound and unbound fraction at the front and back of the cell, using `fminsearchbnd` function (MATLAB Central File Exchange) in MATLAB, we fitted 2-component GMM model with known means ($D_f, D_s$),

$$\underbrace{p_{11}\exp\left(-\frac{(\log_{10}x - \log_{10}D_s)^2}{p_{12}^2}\right)}_{:=f_1(x)} + \underbrace{p_{21}\exp\left(-\frac{(\log_{10}x - \log_{10}D_f)^2}{p_{22}^2}\right)}_{:=f_2(x)}$$

to the front ($P_f(x)$) and back state ($P_b(x)$) SRD histogram data, respectively. The fitting parameters obtained are—back state: $p_{11} = 0.028$, $p_{12} = 1$, $p_{21} = 0.133$, $p_{22} = 0.318$ and front state: $p_{11} = 0.023$, $p_{12} = 0.553$, $p_{21} = 0.150$, $p_{22} = 0.383$. For both cases,

high goodness of fit were obtained (Adjusted R-square values for front and back are $R_f^2 = 0.98$, $R_b^2 = 0.97$, respectively). The relative proportions of bound to unbound factions are—back state: B:LP/LP$_u$= 0.4/0.6 and back state: F:LP/LP$_u$= 0.18/0.82. For PP in addition to similar membrane associated states (PP$_u$, F:PP, B:PP) we assumed another state PP$_c$(c:cytosolic), which can freely diffuse to cytosol, representing its shuttling capability between cytosol and membrane.

The descriptions of the propensity functions for all the reactions and the corresponding parameters are listed in Supplementary Table 4. The parameter values were estimated to match the qualitative features as well as relative proportions of bound to unbound factions of LP from the experimental observations. It is important to note here that when we classified the nodes into "front" and "back" states and tracked the concentration of different species at front and back, over a simulation period of 90s, the ratio of the time integral of bound to unbound species closely matched the original ratio (0.4:0.6 and 0.18:0.82 in back and front, respectively).

For in silico photoconversion, we divided the simulation into two subdomain – intended domain for photoconversion (PC domain) and the rest. We also assumed seven additional species for the respective photoconverted form of PP and LP. During photoconversion all the molecules of LP and PP in the PC domain were irreversibly converted to the respective photoconverted factions. After the photoconversion, photoconverted species follows the same reaction and diffusion dynamics of their respective non-converted forms as described in Supplementary Table 4.

C. *Unstructured Reaction-Diffusion Master Equation (URDME)-based spatiotemporal simulation*: The Unstructured Reaction-Diffusion Master Equation (URDME) framework was used here to test the in silico spatiotemporal profile of the system states of excitable network and different membrane-associated proteins. This approach uses the Next Sub-volume Method[120,121] on an unstructured mesh. For our spatiotemporal simulations, we assumed a two-dimensional square domain of length 20 $\mu$m. The domain is discretized into 11146 nodes. To facilitate the reproducibility, we used the same random seed for all the stochastic simulations. We also used the same initial condition derived from a previous simulation. The detailed implementation of the reactions with respective propensity functions are discussed in supplementary method of Biswas et al.[71].

### Statistics and reproducibility

All the statistical analyses were performed either in MATLAB 2021a or in GraphPad Prism 8. Time-series data are shown as the mean ± s.e.m. or mean ± s.d., as indicated. Tukey's convention was used to plot all the box and whisker plots. Details of statistical tests are indicated in the figure captions. Sample sizes were chosen empirically as per the standard custom in the field and similar sample sizes were used for the experiment and control groups. Each micrograph, including the images presented in Figs. 1a, d, 5c–d, 7b–e, shows a representative image (or image series) from $N \geq 4$ independent experiments. The following convention was followed to show $P$ values: n.s. (not significant), $P > 0.05$ ; *$P \leq 0.05$; **$P \leq 0.01$; ***$P \leq 0.001$; and ****$P \leq 0.0001$.

### Reporting summary

Further information on research design is available in the Nature Portfolio Reporting Summary linked to this article.

## Data availability

All data needed to evaluate the conclusions in the paper are present in the main text or the supplementary materials. Any additional requests for information or data will be fulfilled by the corresponding authors upon reasonable request. Source data are provided with this paper.

## Code availability

Computational simulation and optical flow analysis codes are available on GitHub: https://github.com/tatsatb/Dynamic-Partitioning-of-Membrane-Proteins. The codes are also available on Zenodo (https://zenodo.org/doi/10.5281/zenodo.10072363)[122]. Any additional information will be available from the corresponding authors upon reasonable request.

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

## Acknowledgements

We thank all the members of the Devreotes, Iglesias, and Ueda laboratories as well as the members of the D. Robinson and M. Iijima research groups (Johns Hopkins University School of Medicine) for their helpful feedback. We thank R. R. Kay (MRC LMB), O. D. Weiner (UCSF), and N. Gautam (Washington University School of Medicine in St. Louis) for providing parental cell lines. We thank A. Müller-Taubenberger (LMU Munich) for sharing plasmids. We thank Addgene and dictyBase for providing the plasmids and resources. This work was supported by NIH grant no. R35 GM118177 (to P.N.D.), DARPA HR0011-16-C-0139 (to P.A.I. and P.N.D.) and AFOSR MURI FA95501610052 (to P.N.D.) as well as NIH grant S10 OD016374 (to S. Kuo of the JHU Microscope Facility). This work was also supported by funds from Japan Science and Technology Agency grant no. JPMJPR1879 (to S.M.) and JPMJCR21E1 (to M.U.), Japan Agency for Medical Research and Development grant no. JP20gm0910001 (to M.U.), JSPS KAKENHI grants no. 19H00982 (to M.U.), and no. 19H05798 (to S.M.).

## Author contributions

T.B., P.N.D., and P.A.I. conceptualized the overall study. T.B. designed and performed the majority of experiments. S.M. and M.U. designed single-molecule imaging experiments while S.M. performed them and conducted relevant analysis. Y.K. contributed to the single-molecule experiments. Y.M. provided resources and helped in designing experiments. D.S.P. carried out the neutrophil migration experiments. D.B. and T.B. built the optical flow analysis software and T.B. performed the analysis. D.B. and P.A.I developed the computational models, with inputs from T.B. and P.N.D. D.B. conducted all of the simulations and performed relevant analysis. T.B. quantified and analyzed all other results, with input from the other authors. T.B., P.N.D., and P.A.I. wrote the manuscript, with contribution from all other authors. M.U. supervised single-molecule imaging experiments. P.N.D. and P.A.I. supervised all other parts of the study.

## Competing interests

The authors declare no competing interests.
