## [Peer Review File · Nature Communications]

A dynamic partitioning mechanism polarizes membrane protein distributionREVIEWER COMMENTS

Reviewer #1 (Remarks to the Author):

The manuscript by Banerjee et al describes experiments on the localization of lipid-anchored signaling molecules in Dictyostelium. They interpret the results in the context of the localization of known signaling molecules associated with the front and the back of moving cells. First they define front state and back state regions, and then describe several localization experiments that yield results that have not seen before: lipid-anchored signaling molecules such as Gbg, PKBR1 and RasG are depleted from front state regions and enriched in back state regions. Previously localization of these proteins has been shown to be nearly uniform in moving cells. The new observation is supported by several localization experiments, single molecule diffusion measurements, and modelling. The proposed dynamic partitioning model is potentially very interesting. However several points needs to be addressed.

1. Style of the manuscript

The manuscript is well written linguistically, but unfortunately not scientifically. In old-fashion papers the result section presents just the data with minimal discussion and conclusions. In more modern papers each result section has a neutral title, introduces the scientific question for the experiments and ends with the conclusion related to the question raised, not more than a conclusion that every scientist would make. The purpose of this writing style is that the reader thinks and judges the evidence. In the current manuscript the titles of the results sections and the titles of the figures are formulated as conclusions. In this way the readers are framed before they can read and evaluate the data. And more importantly, the conclusions in the titles may not be correct or needed. Below the titles are indicated between <> with remarks below.

<Multiple lipid-anchored membrane proteins dynamically self-organized to back-state regions>

It is questionable whether this is self-organized and/or dynamically self-organized. Neutral and factual is "Different localization of multiple lipid-anchored membrane proteins in front- state and back-state regions."

<The asymmetric distribution of lipid-anchored proteins is independent of cytoskeleton dynamics>

Correct.

<Shuttling-type and lipid-anchored back membrane proteins responds differently to receptor activation>

Correct

<Lipid-anchored membrane proteins self-organize via a novel partitioning mechanism>

This is close to the final conclusion and title of the manuscript. It has no information on questions addressed and the experiments done.

<Single-molecule imaging establishes dynamic partitioning mechanism >

The single-molecule imaging experiments yield at best different components of the molecules with

different diffusion constants and different fractions in the regions of the cell measured. Neutral is “Single-molecule imaging in different regions of the cell”

<Stochastic simulation of an excitable system demonstrates that dynamic partitioning and shuttling can generate similar propagating wave patterns>

Correct

<Optogenetic alteration of membrane region-specific binding affinity can induce symmetry breaking of uniform lipid-anchored and integral membrane proteins>

Correct

None of the figure titles describes what has been done and what is demonstrated in the figure, but all are conclusions.

The problem with such a writing style is that the reader is pushed too much, some will follow the authors, but many others may get annoyed because they feel driven into a trap.

2. The first paragraph is mainly used to define front-state and back-state regions of the cell. They are defined according to the presence of PIP3 (front-state) or PTEN (back-state). Although authors can make their definitions, here front-state and back-state have strong associations with morphology of the cell, and may not be relevant (see also next point). A moving cell has a front and back, probably best defined by new area gained and old area lost, respectively. In polarized cells the front has extending protrusions and the back a retracting uropod. At a molecular level the front has branched F-actin and the side/back parallel F-actin/myosin. At a signaling level the front has PIP3 (and many others like Ras-GTP) and the back has PTEN (and many others). The definition of a PIP3-rich area as front-state region has several problems. First PIP3 is not required to make a front, second PIP3 is also present in other areas of the moving cell where it does not induce protrusions/front, thirdly PIP3 is associated with many other cellular processes such as pinocytosis cups, cell-cell contacts, cell-substrate contact, cell division, etc. Thus a PIP3-rich region of a cell may not be related to a front of that cell. So they are PIP3-rich regions. For the sake of clarity PIP3-regions are addressed below as front-stage regions.

3. It is surprising that the strong localization of several lipid-anchored proteins in back-state regions and depletion from front-state regions have not been seen before, also not by the authors (Gbg is homogeneously distributed in figure 2 of their Science paper). The authors mention this, but do not give any indication for an explanation. This point must be addressed. In previous reports mentioning uniform localization of Gbg and Ras often confocal images of moving cells are shown; often this is a slice a few micrometers above the surface with an extending protrusion. Many data in the current manuscript come from the ventral surface of the cell where the PIP3 regions look like sites of attachment with the substrate; these PIP3 regions do not look like protrusions and are also not in the front of the cell. Is that correct? How is the distribution of Gbg a few micrometers above the surface in a chemotaxing cell with a clear front and protrusions? Is their proposed dynamic partitioning model for “front-state” regions relevant for the front of the cell with extending protrusion?

4. The data of the single-molecule experiments are used twice in the manuscript, first by the

experimentalists and later by the modelers, yielding very different interpretation of diffusion constants. They should explain why they use different methods and come to different values. In both cases the data are probably analyzed with some optimization procedure, possibly using minimizing RSS. The experimentalists describe three (Table S1) or four components (Table S2); no statistical arguments are given for the model with the minimal number of parameters that these data allow (e.g. Akaike information Criterion). Also no indication is given for the accuracy of the fitted values (e.g. 95% confidential interval); this would permit to indicate which of the three or four components is significantly different between front-state and back-state regions; it may well be that of the four components in Table S2 only the fraction of D1 and D3 are different.

It is understandable that a modeler can not work with four components. They use a different method to find diffusion constants. They use two components for back-stage region and identify a fast and slow component with approximately the values of D1 and D3 of the experimentalist. But surprisingly, and not explained, they use only one component for the front-state region. It is obvious from Figure 5F,G and Table S2,3 that also the front-state region has a slow component in a very significant amount. On page 80 they mention high goodness of fits with one component, but relevant is whether a statistically better fit is obtained with two components. Incorporation of two components in the front-state region may strongly effect the outcome of the model.

5. The data show that the back-state has 2-3-fold more of the slow component than the front-stage region, with 30-fold lower diffusion rate constant than the fast component. This leads to several questions. First, recently the group of Ueda has presented (J Cell Sci. 2023 Feb 15;136(4):jcs260280. doi: 10.1242/jcs.260280) of membrane “domains” with slow , intermediate and fast diffusion of membrane proteins. How are those observations related to the current manuscript? Do back state regions have more of the slow diffusion domains? Secondly, this 30-fold difference in diffusion constant of slow and fast component is probably too large to be explained with changes in molecular weight or viscosity (membrane fluidity), but may be due to interaction with other proteins with slow diffusion (as discussed in the manuscript), and this interaction is more pronounced in the back-state that in the front-state region (more interacting proteins, higher affinity, etc). Is this difference in diffusion sufficient to explain the differences in observed fluorescence in front- state and back-state regions? Third, is the possible stronger binding in the back-stage region a cause or a consequence of the observed enriched fluorescence in the back-stage region. Thus is the potential binding partner already enriched in the back-stage region and thereby more fluorescent protein is present in the back-stage region. If this is the case, then the interacting protein has “symmetry breaking properties” there is not much evidence for the signaling molecules to actively induce “dynamic and self-organized symmetry breaking”, as is mentioned throughout the manuscript in the title of the result sections and captures of the figures.

Reviewer #2 (Remarks to the Author):

In their manuscript entitled “A dynamic partitioning mechanism polarizes membrane protein distribution”, Banerjee et al report the discovery that multiple lipid-anchored membrane proteins in Dictyostelium cells localize asymmetrically to regions associated with cell rear signaling programs. They

show that this localization arises from diffusion and interactions within the membrane, rather than through vesicular trafficking or translocation between the membrane and cytoplasm. Interestingly, it is also independent of the actin cytoskeleton. They use mathematical modeling to show that differences in diffusion within the membrane (due to interaction with other polarized components) is sufficient to explain the asymmetric distribution of these proteins. Finally, they use optogenetic experiments to show that inducing binding to a back component is sufficient to cause a membrane protein to localize to the rear domain. They name this mechanism “dynamic partitioning.”

The authors’ findings are interesting and will advance our understanding of subcellular signaling domains. The finding that lipid-anchored proteins including central components of chemotaxis signaling (PKBR1, Gbeta, and RasG) are asymmetrically localized is surprising. These proteins were thought to be uniformly distributed, and this discovery will be important for thinking about how chemotaxis and related signaling networks function. The imaging experiments are thoughtfully designed, and the movies are beautiful. The single molecule imaging experiment is impressive and compelling. Together, the modeling and imaging convincingly demonstrate that the asymmetric localization of these proteins occurs through diffusion within the membrane. However, I do have concerns about the claims that this represents a “new mechanism of plasma membrane organization” and the description of this mechanism as a “symmetry breaking” process. I am also not convinced that the mechanism demonstrated in HL60 cells is the same as the one observed in Dictyostelium.

Major points:

- 1.) The authors claim in the abstract that they have uncovered a “new mechanism of plasma membrane organization”. While the authors’ observations are interesting and the mechanism is well-described, this claim seems overstated for the following reasons:
 - a.) Proteins are already known to localize to membrane microdomains through interactions with the lipids enriched in these domains.
 - b.) Asymmetric accumulation of a protein within the membrane due to polarized interactions which slow its diffusion is an established concept. For example, it was the basis for the “Membrane Proximal Actin” biosensor from Tobias Meyer’s group (PubMed ID 32527825).
 - c.) The asymmetric localization of any protein due to interactions with other cellular components is also achieved by changing the diffusion properties of that protein. This is the case for a protein that localizes to the cell front through binding to branched actin or for a protein that localizes to the centriole by interacting with centriolar proteins.
 - d.) The exact mechanism(s) for polarization of PKBR1, Gbeta, and RasG are still unclear. (Is it due to interaction with lipid head groups or with something else?)

- 2.) I am not convinced that the rear localization in the HL60 optogenetic experiment is achieved through the “dynamic partitioning” mechanism. The results might be better explained by the rearward membrane flow that has been observed in fast migrating cells. This flow is known to carry large complexes to the cell rear, with accumulation at the uropod. Consistent with the flow mechanism, the accumulation of Lyn at the rear appears to be slower than the relocalization of cAR1 in the analogous Dicty experiments, and Lyn appears to accumulate at the uropod, rather than being excluded from the protrusive cell front domain. The control experiment would seem to argue against the flow hypothesis. However, the cryptochrome optogenetic system is known to cause clustering in addition to

heterodimerization. The resulting complexes would have multiple lipid-interacting polybasic domains, which might allow the flow to carry them to the rear. An additional experiment is needed to demonstrate that this rearward localization is not dependent on flow.

A simple experiment that could address this question would be to repeat the experiment in the presence of the ROCK inhibitor, which would prevent the cytoskeletal-driven flow, while still allowing the formation of protrusive fronts.

3.) It remains unclear to what degree this mechanism occurs in mammalian cells. For this claim, the authors rely on HL60 data which may reflect a different mechanism (see point 2 above). The authors' previous work indicates that R-Pre is depleted from PIP3-rich regions in cortical waves in macrophage cells. However, it is unclear if this would be the case in migrating HL60 cells or in other contexts in mammalian cells. More clarity about the role of this mechanism in mammalian cells would strengthen the paper.

4.) The use of "symmetry breaking" could confuse readers. Symmetry breaking usually refers to a system behavior where asymmetry is generated intrinsically by dynamic properties of the system. In the model presented in this work, the excitable network breaks symmetry in this sense. The dynamic partitioning allows the asymmetry to be transmitted to the LP protein.

Minor points:

1.) The "R-Pre" peptide should be described as being positively charged. This peptide is introduced on page 6 and described simply as a "prenylated peptide", which gives the impression that something about the prenyl group is driving the rear localization of this peptide. However, from the broader set of results in this paper and the authors' recent work, it seems clear that the +8 positive charge is driving the localization. This is (perhaps unintentionally) misleading.

2.) More generally, the authors should provide more discussion of why the analyzed lipid-anchored proteins are rear-localized. They previously showed that the localization of R-Pre is determined by membrane charge (PubMed ID 36202973). Similar to R-Pre, the PKBR1, PKBR1(N150), and RasG proteins all have polybasic motifs adjacent to their lipidation sites. G-protein subunits also have positively charged patches that are implicated in plasma membrane association (PubMed ID 23161140). Thus, all of the lipid-anchored proteins analyzed in this study may be rear-localized through interaction with negatively charged lipids. While the proposed mechanism could work with other types of interactions, it would improve clarity to discuss how membrane charge may contribute for these proteins.

3.) In contrast to positively charged peptides, the authors previously found that two more neutral anchoring signals (Lyn and Palm/Pre) localize homogeneously. Including data (or at least references) for one or both of these would be an important contrast to clarify that the rear-localization is not a generic property of lipid-anchored proteins.

4.) For the experiments such as those included in Figures 1 and 2, a cross-correlation analysis would be

very helpful. It is clear that the localization of PIP3 is anticorrelated with that of other proteins, but is there a temporal lag?

5.) On page 7, for the Latrunculin experiments, the authors state “When periodic circulating waves were induced in these cells”. How were waves induced?

6.) In Fig 3A-3H and Fig S5, the authors do not specify in the main text that global cAMP stimulation experiments were also performed in LatA treated cells. There is a brief mention of this in the Methods section but this seems like a detail that should also be provided in the main text.

7.) Though mentioned in the methods section, the authors should state in the main text that giant, electrofused cells were used for photoconversion and ventral wave experiments.

8.) This is a small point, but in Figure 7B and the associated text, the authors state that cAR1 was induced to localize to back-state regions. In fact, the localization of the R+ peptide was measured. Presumably, this also reflects cAR1, but that was not measured directly. The text could be reworded to be more precise.

9.) While movies were provided for most imaging experiments, they were not provided for the HL60 optogenetic experiments. Adding movies for the HL60 experiments would be helpful.

10.) How do the authors reconcile their observation that Gbeta is asymmetrically localized to cell rear regions with their group’s prior publication (Jin et al 2000, cited in the discussion) that Gbeta is localized with a polarization towards the cell front?

11.) Related to major point 4, on page 19 the authors refer to the “self-organizing dynamic partitioning events in plasma membrane”. The dynamic partitioning mechanism described does not appear to be self-organizing, but it could contribute to self-organizing signaling domains.

12.) There were a number of small typo or grammatical errors that should be corrected.

Reviewer #3 (Remarks to the Author):

Understanding sorting mechanisms of membrane proteins is central to many cellular processes. The current paper describes a possible new mechanism based on differences in diffusional dynamics of lipid anchored proteins between different regions in the plasma membrane which arise during formation and propagation of protrusions. A large array of experimental data is provided, which indeed indicate the presence of a sorting potential for this class of proteins. It is then argued that existing mechanisms such as the involvement of the actin filament network or shuttling of proteins between the membrane and cytosol cannot explain the observed partitioning behaviour, giving room to a novel type of mechanism.

Stochastic simulations are then used to provide evidence for a novel type of sorting mechanism, coined dynamic partitioning. However, before possibly accepting this mechanism, I have two major concerns.

First of all, it is not clear to me why a thermodynamic-based mechanism (instead of a dynamic one) cannot be invoked to explain the results. As acknowledged by the authors themselves, the back and front regions of the plasma membrane have very distinct compositions, both in terms of protein content and considering lipid types. It is also well known that proteins can preferentially sort to different membrane regions based on their thermodynamic preference, which could arise from global membrane properties as well as from detailed protein-lipid interactions. The RAFT hypothesis is based on this principle. Although it remains unclear to date how RAFT domains are precisely organized *in vivo*, membrane heterogeneity as an organizational principle is well established. Remarkably, the concept of RAFTS is only mentioned at the very end of the paper. If lipidated proteins simply sort to the back region of the membrane due to more favourable interactions with the lipid population in this region, this would provide a direct driving force for the observed partitioning behaviour. Note that RAFT-type of domains could couple to curvature gradients that might exist in the protruding front end, providing a driving force for RAFT coalescence and sorting itself.

My second concern is about the stochastic simulations, which I find hard to understand the way it is described in the current paper. For instance, it is unclear what the 'refractory' state is, as it was not mentioned before in the manuscript. It is also unclear why we need an 'excitable system' (what is being excited here?), where the 'mutually inhibitory action' comes from (how and why are the F and B states inhibiting each other?), and what the 'delayed negative feedback' represents (why/how do F and R state provide negative feedback?). Figure 6A is not very helpful to understand what went into the modelling approach (and why!). Things become even more confusing when reading the Method section where one reads sentences such as 'Basal and R-mediated inactivation of F-molecules' and 'non-linear B-dependent activations and inhibitions' (what are F-molecules? Why are they basal-inactivated? Why non-linear? What does all of this even mean in the context of the rest of the paper?). Without a clear rationale of invoking these terms, one is left with the impression that a lot of fiddling was done to get behaviour consistent with the desired outcome. Furthermore, it remains puzzling (and unexplained) what the 'bound' and 'unbound' states of the lipid anchored proteins (LPs) are - it was argued before that these proteins actually do NOT unbind, due to the strong association of the anchor with the membrane. Could it be that in fact these LPs still can unbind, even though at a much reduced rate compared to peripherally bound proteins (PPs), and that this fraction accounts for the observed effects (being similar to how PPs partition)? The difference between diffusion constants of the bound and unbound states is really large (40-fold), and cannot be easily explained by assuming a different membrane environment. The fact that, according to the data in Table S3, the diffusion constants for the bound and unbound states are the same for LPs and PPs supports this notion.

We thank the reviewers for their time and for providing overall positive and constructive feedback. The reviewers' comments and our responses (in red) are provided below.

REVIEWER COMMENTS

Reviewer #1 (Remarks to the Author):

The manuscript by Banerjee et al describes experiments on the localization of lipid-anchored signaling molecules in Dictyostelium. They interpret the results in the context of the localization of known signaling molecules associated with the front and the back of moving cells. First they define front state and back state regions, and then describe several localization experiments that yield results that have not seen before: lipid-anchored signaling molecules such as Gbg, PKBR1 and RasG are depleted from front state regions and enriched in back state regions. Previously localization of these proteins has been shown to be nearly uniform in moving cells. The new observation is supported by several localization experiments, single molecule diffusion measurements, and modelling. The proposed dynamic partitioning model is potentially very interesting. However several points needs to be addressed.

We would like to thank the Reviewer #1 for providing overall positive, helpful, and constructive feedback.

1. Style of the manuscript

The manuscript is well written linguistically, but unfortunately not scientifically. In old-fashion papers the result section presents just the data with minimal discussion and conclusions. In more modern papers each result section has a neutral title, introduces the scientific question for the experiments and ends with the conclusion related to the question raised, not more than a conclusion that every scientist would make. The purpose of this writing style is that the reader thinks and judges the evidence. In the current manuscript the titles of the results sections and the

titles of the figures are formulated as conclusions. In this way the readers are framed before they can read and evaluate the data. And more importantly, the conclusions in the titles may not be correct or needed. Below the titles are indicated between <> with remarks below.

<Multiple lipid-anchored membrane proteins dynamically self-organized to back-state regions>

It is questionable whether this is self-organized and/or dynamically self-organized. Neutral and factual is “Different localization of multiple lipid-anchored membrane proteins in front- state and back-state regions.”

Correct.

<Shuttling-type and lipid-anchored back membrane proteins responds differently to receptor activation>

Correct

This is close to the final conclusion and title of the manuscript. It has no information on questions addressed and the experiments done.

<Single-molecule imaging establishes dynamic partitioning mechanism >

The single-molecule imaging experiments yield at best different components of the molecules with different diffusion constants and different fractions in the regions of the cell measured.

Neutral is “Single-molecule imaging in different regions of the cell”

<Stochastic simulation of an excitable system demonstrates that dynamic partitioning and shuttling can generate similar propagating wave patterns>

Correct

<Optogenetic alteration of membrane region-specific binding affinity can induce symmetry breaking of uniform lipid-anchored and integral membrane proteins>

Correct

As per the suggestion of the reviewer, we have extensively edited our section titles to adhere to the neutral tone.

None of the figure titles describes what has been done and what is demonstrated in the figure, but all are conclusions.

The problem with such a writing style is that the reader is pushed too much, some will follow the authors, but many others may get annoyed because they feel driven into a trap.

We have also significantly changed our figure legends as per the reviewer's suggestion. We believe, in the current version, the figure legends do describe the experiments or simulations presented and they no longer sound like conclusions.

2. The first paragraph is mainly used to define front-state and back-state regions of the cell. They are defined according to the presence of PIP3 (front-state) or PTEN (back-state). Although authors can make their definitions, here front-state and back-state have strong associations with morphology of the cell, and may not be relevant (see also next point). A moving cell has a front and back, probably best defined by new area gained and old area lost, respectively. In polarized cells the front has extending protrusions and the back a retracting uropod. At a molecular level the front has branched F-actin and the side/back parallel F-actin/myosin. At a signaling level the front has PIP3 (and many others like Ras-GTP) and the back has PTEN (and many others). The definition of a PIP3-rich area as front-state region has several problems. First PIP3 is not required to make a front, second PIP3 is also present in other areas of the moving cell where it does not induce protrusions/front, thirdly PIP3 is associated with many other cellular processes such as pinocytosis cups, cell-cell contacts, cell-substrate contact, cell division, etc. Thus a PIP3-rich region of a cell may not be related to a front of that cell. So they are PIP3-rich regions. For the sake of clarity PIP3-rich regions are addressed below as front-stage regions.

For convenience, we have addressed this point and next point together (please see below).

3. It is surprising that the strong localization of several lipid-anchored proteins in back-state regions and depletion from front-state regions have not been seen before, also not by the authors (Gbg is homogeneously distributed in figure 2 of their Science paper). The authors mention this, but do not give any indication for an explanation. This point must be addressed. In previous reports mentioning uniform localization of Gbg and Ras often confocal images of moving cells are shown; often this is a slice a few micrometer above the surface with an extending protrusion. Many data in the current manuscript come from the ventral surface of the cell where the PIP3 regions look like sites of attachment with the substrate; these PIP3 regions do not look like protrusions and are also not in the front of the cell. Is that correct? How is the distribution of Gbg at a few micrometer above the surface in a chemotaxing cell with a clear front and protrusions? Is their proposed dynamic partitioning model for “front-state” regions relevant for the front of the cell with extending protrusion?

The reviewer is right that the waves and protrusions do look different geometrically. However, their molecular arrangement as well as their phenotypic responses to different perturbations (genetic, chemical, mechanical, or electrical) always closely matched each other, as extensively demonstrated by the work from Günther Gerisch’s group (*Gerisch G. et al., Biophys J.* 2012, 103(6): 1170–1178; *Gerisch G. et al., BMC Cell Biol.* 2011, 12:42; *Gerhardt M. et al., J Cell Sci.* 2014, 127: 4507–4517; *Lange M. et al., J Cell Sci.* 2016, 129: 3462–3472), as well as work from other groups, including ours (*Miao Y. et al, Nat. Cell Biol.* 2017, 19(4), 329–340; *Banerjee T. et al, Nat. Cell Biol.* 2022, 24(10), 1499–1515; *Miao Y. et al, Mol Syst Biol.* 2019, 15(3): e8585; *Yang Q. et al., eLife* 2022, 11: e73198; *Weiner O. D. et al., PLoS Biol.* 2007, 5(9):e221; *Matsuoka S. et al., Nat Commun.* 2018, 9(1):4481; *Fukushima S. et al, J. Cell Sci.* 2019, 132(5), jcs224121). Moreover, when a wave reaches the cell boundary, it creates a protrusion (*Weiner O. D. et al., PLoS Biol.* 2007, 5(9):e221; *Zhan H. et al., Dev Cell.* 2020; *Huang, C. H. et al., Nat Cell Biol.* 2013;15(11):1307–1316). Additionally, although the reviewer is correct that PIP3 independent protrusion formation is possible and PIP3 does appear in other structures, it is to be noted that PIP3 sensors have been widely used as a useful surrogate for front-activity in many of these and other papers. Nevertheless, to address these two issues that the reviewer has brought

up, we have included data from multiple new experiments in the revised version of the manuscript.

i) We co-expressed our lipid-anchored proteins (i.e., PKBR1, G β γ , RasG, and R-Pre) with newly polymerized F-actin biosensor LimE or activated Ras biosensor RBD. Then, in those cells, we recorded protrusion formation events (i.e., we imaged confocal slices a few micrometers above the substrate-attached surface). We found that during all the protrusion formation events, LimE and RBD were enriched in the protrusion (as expected), whereas PKBR1, G β γ , RasG, and R-Pre were depleted from those membrane domains and were dynamically localized to the back-state regions of the membrane (Figure S3A-S3E). These data are consistent with our previous data where we showed that, during protrusion formation (as well as ventral wave propagation) events, these proteins are depleted from protrusions or front-state regions where PIP3 levels were high (Figure S2A-S2C, Figure S2G).

ii) Although PIP3 sensors are generally used as a useful surrogate for front-activity, we are aware that PIP3 production is not essential to create a protrusion or front-state regions and can be involved in other physiological events, and hence, we wanted to demonstrate that the dynamic back-state localization of these newly identified lipid-anchored proteins are PIP3 independent. We treated the cells with potent PI3K inhibitor LY294002 and recorded the ventral wave propagation (on substrate-attached surface) and protrusion formation events (in a confocal slice). We observed that in all these cases, these lipid-anchored proteins shunned PIP3-independent front-state regions or protrusions and maintained a consistent complementarity with newly polymerized F-actins (Figure S3F-S3I). In other words, irrespective of our choice of front biosensor (PIP3/RasGTP/F-actin polymerization), the complementary localizations of our lipid-anchored proteins with respect to those biosensors were observed throughout.

iii) As per the reviewer's suggestion, we performed the chemotaxis assay as well with LimE co-expressing cells. We observed that as cells migrated towards the chemotactic gradients, LimE preferentially localized in the membrane domains facing the needle, while PKBR1 and G β γ localized to the back of the cell membrane (Figure S4A-S4C). This is consistent with complementary dynamics that we observed in case of ventral wave propagation and protrusion formation during random migration.

Finally, it is true that our current data contradicts our group's old data (*Jin T. et al., Science 2000, 287(5455):1034-1036*) regarding G β localization. We believe that can be attributed to the fact that those cells were imaged using epifluorescence microscopes, almost twenty-five years ago, in the early days of live-cell imaging when optical resolution and sensitivity were severely limited (compared to today's standard confocal images). Additionally, unlike the ventral wave images which contain no membrane folds (*Gerisch G. et al., Biophys J. 2004, 87(5):3493-503; Gerisch G. et al., Biophys J. 2009, 96(7):2888-2900*), the spatial gradients observed during protrusion formation, as it is usually recorded a few micrometer above the substrate-attached surface, sometimes appear weak. The data we present here using multimodal imaging techniques in different physiological scenarios reveal a more accurate picture of localization dynamics.

4. The data of the single-molecule experiments are used twice in the manuscript, first by the experimentalists and later by the modelers, yielding very different interpretation of diffusion constants. They should explain why they use different methods and come to different values. In both cases the data are probably analyzed with some optimization procedure, possibly using minimizing RSS. The experimentalists describe three (Table S1) or four components (Table S2); no statistical arguments are given for the model with the minimal number of parameters that these data allow (e.g. Akaike information Criterion). Also no indication is given for the accuracy of the fitted values (e.g. 95% confidential interval); this would permit to indicate which of the three or four components is significantly different between front-state and back-state regions; it may well be that of the four components in Table S2 only the fraction of D1 and D3 are different.

It is understandable that a modeler can not work with four components. They use a different method to find diffusion constants. They use two components for back-stage region and identify a fast and slow component with approximately the values of D1 and D3 of the experimentalist. But surprisingly, and not explained, they use only one component for the front-state region. It is obvious from Figure 5F,G and Table S2,3 that also the front-state region has a slow component in a very significant amount. On page 80 they mention high goodness of fits with one component, but relevant is whether a statistically better fit is obtained with two components. Incorporation of two components in the front-state region may strongly effect the outcome of the model.

In the revised version of the manuscript, we have updated the supplemental tables with additional 95% confidence interval and Akaike information Criterion values for the experimental data (Table S1 and Table S2).

The reviewer is correct that we made a simplifying assumption by reducing the 4-component description of the diffusion dynamics that was used to fit the experimental data to a simpler one that we used in the computational model to explain the possible interaction dynamics between the lipid-anchored molecules and membrane states. We also apologize for not providing enough explanation of the approximation method as well as the statistical evidence to do so. We have now included all these details in the revised manuscript and an excerpt from the revised method section is quoted here for convenience:

In developing the model, we sought a simpler description that had 1-2 components. To this end, we first checked the modality of the short-range diffusion constant (SRD) data corresponds to front and back states using the dip test of unimodality (*Hartigan J.A. and Hartigan P.M., Ann. Stat. 1985, 13(1): 70-84*) and found the statistical evidence in favor of unimodality for the “front” data in contrast to the multimodality in the “back” data (p-values from dip test: back = 0.001, front = 0.99, null hypothesis, H_0 : distribution is unimodal). Additionally, from the computation of Sarle’s bimodality coefficient (BC) (*Knapp T. R., J. Mod. Appl. Stat. Methods 2007, 6(1):3*) for the back data we found statistical evidence in favor of the bimodality (BC = 0.7151 > 0.555). This motivated us to fit a two-component model to back data whereas using only one component to approximate the front data. The diffusion constants (slow-moving: $D_{slow} = 0.0215 \mu\text{m}^2/\text{s}$, and fast-moving: $D_{fast} = 0.416 \mu\text{m}^2/\text{s}$, proportion: $a_{slow} = 0.34$, $a_{fast} = 0.66$) derived from the fitting of two-component model to the back data were close to D_1 and D_3 (Supplementary Table S3, Supplementary Table S3 Figure 1). Also, the Gaussian mixture model fit to the experimental back data with means D_1 and D_3 gave the lowest AIC score among all the paired combinations ($AIC_{D_1,D_2} > AIC_{D_2,D_3} > AIC_{D_1,D_3}$; Supplementary Table S3). Now, the diffusion coefficient ($0.46 \mu\text{m}^2/\text{s}$) derived from fitting the single component to the front data was close to the diffusion constant of the fast-moving fraction. From biological perspective, this likely indicates that the fast-moving fraction does not interact with either of membrane states whereas the slow-moving fraction resulted from the

interaction with the back membrane state. Thus, for the computational model, we assumed the diffusion constants for the two fractions as follows slow-moving $LP_b = 0.0215\mu m^2/s$, and fast-moving $LP_u = 0.416\mu m^2/s$.

5. The data show that the back-state has 2-3-fold more of the slow component than the front-stage region, with 30-fold lower diffusion rate constant than the fast component. This leads to several questions. First, recently the group of Ueda has presented (J Cell Sci. 2023 Feb 15;136(4):jcs260280. doi: 10.1242/jcs.260280) of membrane “domains” with slow, intermediate and fast diffusion of membrane proteins. How are those observations related to the current manuscript? Do back state regions have more of the slow diffusion domains? Secondly, this 30-fold difference in diffusion constant of slow and fast component is probably too large to be explained with changes in molecular weight or viscosity (membrane fluidity), but may be due to interaction with other proteins with slow diffusion (as discussed in the manuscript), and this interaction is more pronounced in the back-state than in the front-state region (more interacting proteins, higher affinity, etc). Is this difference in diffusion sufficient to explain the differences in observed fluorescence in front- state and back-state regions? Third, is the possible stronger binding in the back-stage region a cause or a consequence of the observed enriched fluorescence in the back-stage region. Thus is the potential binding partner already enriched in the back-stage region and thereby more fluorescent protein is present in the back-stage region. If this is the case, then the interacting protein has “symmetry breaking properties” there is not much evidence for the signaling molecules to actively induce “dynamic and self-organized symmetry breaking”, as is mentioned throughout the manuscript in the title of the result sections and captures of the figures.

In the study from Ueda group mentioned by the reviewer (Takebayashi K. et al., DOI: 10.1242/jcs.260280), three membrane domains were suggested where transmembrane proteins exhibit lateral diffusion with different diffusion coefficients. Each of the three diffusion coefficients showed a weak inverse relationship with a radius of the protein, which is estimated from the number of transmembrane helices, obeying Saffman-Delbruck model (*Saffman P.G. and Delbrück M., Proc Natl Acad Sci U S A 1975, 72(8):3111-3*). Based on this model, it was suggested that the lateral diffusion of the transmembrane proteins depends largely on the

viscosity of the membrane rather than the radius or the molecular weight of the protein. On the other hand, it remained an open question how the viscosity affects the diffusion of peripheral and lipid-anchored membrane proteins, which often exhibit various kinds of interactions with the membrane, including different electrostatic interactions (such as binding to specific anionic lipids such as PI(4,5)P2, phosphatidic acid, and phosphatidylserine) and binding to other integral membrane proteins. Therefore, it is more likely that the lipid composition or existence of specific integral membrane proteins, rather than the viscosity of the membrane, affects the behaviors of these proteins. It is important to note here that, compared to membrane domains reported by Takebayashi et al., the propagating membrane domains that we reported in this manuscript are very dynamic and relatively larger-scale structures that often encompass multiple square micrometer areas. The back-state regions may contain more of the slow diffusion domains. In turn, the slow diffusion domain can contribute to the retention of some of the negatively charged phospholipids or specific transmembrane proteins (which may serve as binding partner) to achieve the partitioning of lipid-anchored proteins. We incorporated this point in the revised version of our manuscript.

Lipid-anchored and peripheral membrane proteins exhibit large variation in the diffusion coefficient, as the Ueda group has previously revealed in PTEN, alpha subunit of heterotrimeric G-protein, and a PH-domain-containing protein. Such large variation is often caused by the difference in the binding partner molecules (*Matsuoka S. et al., Nat Commun. 2018, 9(1):4481; Matsuoka S. et al., J Cell Sci. 2006, 119:1071-9; Miyanaga Y. et al., Biochem Biophys Res Commun. 2018, 507(1-4):304-310; Yasui M. et al., PLoS Comput Biol. 2014, 10(9):e1003817*). For example, PTEN exhibits slow diffusion coefficient ($D = 0.01 \mu\text{m}^2/\text{sec}$), when it presumably binds to some membrane protein. It also exhibits much faster diffusion ($D = 0.52 \mu\text{m}^2/\text{sec}$) when it undergoes “hopping” diffusion, in which the very instant association (~ 30 msec) due to the electrostatic interaction with anionic phospholipids such as PI(4,5)P2 and PS is repeated, thus showing a 50-fold difference in diffusion coefficient depending on to which it binds. Based on these facts, it is possible that the slow diffusion components of the lipid-anchored molecules that we reported here reflects those molecules that bind to some specific transmembrane proteins or anionic lipids (latter is possibly more probable case for the R-Pre and PKBR1). Our simulations demonstrate that the spatially heterogeneous diffusion profiles that we obtained from single-molecule imaging experiments can sufficiently generate familiar propagating wave patterns.

The reviewer brings up the issue of cause and effect. Partitioning driven by the difference in lateral diffusional mobility most likely requires prepositioning of certain molecules in specific membrane domains. Nevertheless, the molecules in this study may contribute to feedback loops that amplify these spatial heterogeneities. For example, localization of PKBR1, G $\beta\gamma$, and RasG to back-state regions possibly counteract activation at the front-state regions, thereby controlling the amplification of signaling activities which further collectively define the state of the membrane and the enrichment or depletion of molecules therein. Under these scenarios, we believe, it is difficult to label the cause and consequence.

We would like to clarify here that when we mentioned “symmetry breaking”, we essentially meant the symmetry breaking or compartmentalization of specific lipid-anchored or integral membrane proteins to a specific domain of the membrane, which does not necessarily need to correlate with the overall symmetry breaking of the plasma membrane. For example, even when membrane symmetry is broken, as it happens during protrusion formation and ventral wave propagation (among other processes), certain proteins such as cAR1 or the C-terminal of HRas (Palm/Pre) still remain uniform (which was, in fact, the expected phenomena). We concur with the reviewer that our language was confusing in some cases. So, as per the reviewer’s suggestions, we have edited the section titles, figure legends, and also modified the introduction and discussion section of the manuscript thoroughly to convey the message more directly.

Reviewer #2 (Remarks to the Author):

In their manuscript entitled “A dynamic partitioning mechanism polarizes membrane protein distribution”, Banerjee et al report the discovery that multiple lipid-anchored membrane proteins in *Dictyostelium* cells localize asymmetrically to regions associated with cell rear signaling programs. They show that this localization arises from diffusion and interactions within the membrane, rather than through vesicular trafficking or translocation between the membrane and cytoplasm. Interestingly, it is also independent of the actin cytoskeleton. They use mathematical modeling to show that differences in diffusion within the membrane (due to interaction with other polarized components) is sufficient to explain the asymmetric distribution of these proteins. Finally, they use optogenetic experiments to show that inducing binding to a back component is sufficient to cause a membrane protein to localize to the rear domain. They name this mechanism “dynamic partitioning.”

The authors’ findings are interesting and will advance our understanding of subcellular signaling domains. The finding that lipid-anchored proteins including central components of chemotaxis signaling (PKBR1, Gbeta, and RasG) are asymmetrically localized is surprising. These proteins were thought to be uniformly distributed, and this discovery will be important for thinking about how chemotaxis and related signaling networks function. The imaging experiments are thoughtfully designed, and the movies are beautiful. The single molecule imaging experiment is impressive and compelling. Together, the modeling and imaging convincingly demonstrate that the asymmetric localization of these proteins occurs through diffusion within the membrane. However, I do have concerns about the claims that this represents a “new mechanism of plasma membrane organization” and the description of this mechanism as a “symmetry breaking” process. I am also not convinced that the mechanism demonstrated in HL60 cells is the same as the one observed in *Dictyostelium*.

We would like to thank the Reviewer #2 for showing enthusiastic response to our study and for appreciating the strength of our overall work. We also sincerely thank Reviewer #2 for providing constructive feedback.

Major points:

- 1.) The authors claim in the abstract that they have uncovered a “new mechanism of plasma membrane organization”. While the authors’ observations are interesting and the mechanism is well-described, this claim seems overstated for the following reasons:
 - a.) Proteins are already known to localize to membrane microdomains through interactions with the lipids enriched in these domains.
 - b.) Asymmetric accumulation of a protein within the membrane due to polarized interactions which slow its diffusion is an established concept. For example, it was the basis for the “Membrane Proximal Actin” biosensor from Tobias Meyer’s group (PubMed ID 32527825).
 - c.) The asymmetric localization of any protein due to interactions with other cellular components is also achieved by changing the diffusion properties of that protein. This is the case for a protein that localizes to the cell front through binding to branched actin or for a protein that localizes to the centriole by interacting with centriolar proteins.
 - d.) The exact mechanism(s) for polarization of PKBR1, Gbeta, and RasG are still unclear. (Is it due to interaction with lipid head groups or with something else?)

As per the reviewer’s suggestion, we have changed the wordings in the abstract and removed the “plasma membrane organization” part. We believe that the mechanism that we reported in this paper is distinctly different from previously reported mechanisms as it deals with formation of large-scale domains, extending across tens of micrometers, which are spatially and temporally dynamic and organized properly even in the absence of cytoskeleton. While multiple reports do exist which attribute the clustering of lipidated proteins to specific lipids, the reported cluster size is usually not bigger than a couple of hundred nanometers. Also, those nanoclusters are not known to propagate coordinately in time and space. Again, diffusion-limited polarization was reported in different physiological scenarios, but it always strongly depended upon actin cytoskeleton dynamic and/or vesicular trafficking. On the other hand, the novel phenomena and mechanism that we are describing here depends only on the intrinsic property of the protein itself and the membrane-domain it is binding to. However, we agree with the reviewer that to truly establish it as a new mechanism of plasma membrane organization, we possibly needed to identify all the specific interactions that PKBR1, Gβγ, and RasG make with different membrane

proteins and lipids to generate these differential diffusion states. Hence, we have modified our language in abstract and in the main body of the manuscript. We think that the current version of the manuscript more aptly describes the picture.

2.) I am not convinced that the rear localization in the HL60 optogenetic experiment is achieved through the “dynamic partitioning” mechanism. The results might be better explained by the rearward membrane flow that has been observed in fast migrating cells. This flow is known to carry large complexes to the cell rear, with accumulation at the uropod. Consistent with the flow mechanism, the accumulation of Lyn at the rear appears to be slower than the relocalization of cAR1 in the analogous Dicty experiments, and Lyn appears to accumulate at the uropod, rather than being excluded from the protrusive cell front domain. The control experiment would seem to argue against the flow hypothesis. However, the cryptochrome optogenetic system is known to cause clustering in addition to heterodimerization. The resulting complexes would have multiple lipid-interacting polybasic domains, which might allow the flow to carry them to the rear. An additional experiment is needed to demonstrate that this rearward localization is not dependent on flow.

A simple experiment that could address this question would be to repeat the experiment in the presence of the ROCK inhibitor, which would prevent the cytoskeletal-driven flow, while still allowing the formation of protrusive fronts.

We would like to thank the reviewer for this suggestion. We performed the optogenetic recruitment experiment in HL-60 cells which were pre-treated with ROCK inhibitor Y-27632. We observed that even when cytoskeleton-driven membrane flow was prevented, Lyn11, along with recruited R⁺, consistently localized to the back of the cell membrane (Figure S14E, S14F). We would also like to point out that, in some instances, the Lyn11, just like cAR1 (in *Dictyostelium*), was just excluded from the protrusive cell front and did not specifically localize to uropods (Figure S14F). To further establish the cytoskeleton-independent dynamic partitioning mechanism in mammalian cells, we also performed multiple new experiments (please see below).

3.) It remains unclear to what degree this mechanism occurs in mammalian cells. For this claim, the authors rely on HL60 data which may reflect a different mechanism (see point 2 above). The authors' previous work indicates that R-Pre is depleted from PIP3-rich regions in cortical waves in macrophage cells. However, it is unclear if this would be the case in migrating HL60 cells or in other contexts in mammalian cells. More clarity about the role of this mechanism in mammalian cells would strengthen the paper.

As we mentioned above, we believe our new data in HL-60 cells indicate that lipid-anchored proteins can possibly be partitioned by a cytoskeletal-independent mechanism. To further test the generality of the mechanism that we established in *Dictyostelium*, we performed a few new experiments in RAW 264.7 macrophage cells: First, in accordance with our previous report (which reviewer has mentioned here), we found that R-Pre does exhibit consistent complementarity with respect to PIP3-rich regions during ventral wave propagation in these cells (Figure S7A, Video S6). Secondly, we tested whether this dynamic compartmentalization depends on cytoskeleton dynamics. Obviously, when the RAW 264.7 cells were treated with both Latrunculin A and Y-27632 before initiating the frustrated phagocytosis, the cells could not spread and we could not observe any cortical patterning (Figure S7B). However, when we first pre-treated the cells with 50 μ M of Y-27632 alone, allowed the cells to spread, and after that added 5 μ M of Latrunculin A, we observed that cells could no longer move or produce any ruffles or protrusions, but importantly, the ventral wave did propagate (as it did in cytoskeleton-inhibited *Dictyostelium* cells, as shown in Figure S6E) and PIP3 and R-Pre maintained dynamic complementarity (Figure S7C, Video S7). Either when big waves (Figure S7C first 4 time points) or small wavelets (Figure S7C last 4 time points) of PIP3 have propagated, R-Pre remained consistently depleted from those regions. Finally, we also performed global receptor activation assay in these cells. When we globally stimulated the cells with C5a receptor agonist, we observed that while PH_{AKT} got uniformly recruited over the membrane (showing signaling activation) and then returned to cytosol (showing adaptation), R-Pre remained membrane bound throughout the entire experiment (Figure S8I, S8J, Video S13). These experiments, together indicate that R-Pre, even in macrophage cells, does not shuttle, and generates its asymmetric patterns via a cytoskeleton-dynamics independent dynamic partitioning mechanism.

4.) The use of “symmetry breaking” could confuse readers. Symmetry breaking usually refers to a system behavior where asymmetry is generated intrinsically by dynamic properties of the system. In the model presented in this work, the excitable network breaks symmetry in this sense. The dynamic partitioning allows the asymmetry to be transmitted to the LP protein.

We apologize for any confusion. When we used “symmetry breaking” phrase previously, we wanted to refer to the symmetry breaking or compartmentalization events that specific lipid-anchored or integral membrane protein molecules underwent. The overall symmetry breaking of the plasma membrane does not always result in asymmetric or polarized distribution of various membrane proteins. For example, even when membrane symmetry is broken, as it happens during protrusion formation and ventral wave propagation, cAR1 or the C-terminal of HRas (Palm/Pre) still remain uniform (which was, in fact, the expected phenomena). We do understand that this was not conveyed clearly (another reviewer also mentioned this point). So, as per the reviewers’ suggestions, we have edited the manuscript thoroughly to clarify this.

Minor points:

1.) The “R-Pre” peptide should be described as being positively charged. This peptide is introduced on page 6 and described simply as a “prenylated peptide”, which gives the impression that something about the prenyl group is driving the rear localization of this peptide. However, from the broader set of results in this paper and the authors’ recent work, it seems clear that the +8 positive charge is driving the localization. This is (perhaps unintentionally) misleading.

We thank the reviewer for pointing this out. We have edited this in the revised manuscript.

2.) More generally, the authors should provide more discussion of why the analyzed lipid-anchored proteins are rear-localized. They previously showed that the localization of R-Pre is determined by membrane charge (PubMed ID 36202973). Similar to R-Pre, the PKBR1, PKBR1(N150), and RasG proteins all have polybasic motifs adjacent to their lipidation sites. G-

protein subunits also have positively charged patches that are implicated in plasma membrane association (PubMed ID 23161140). Thus, all of the lipid-anchored proteins analyzed in this study may be rear-localized through interaction with negatively charged lipids. While the proposed mechanism could work with other types of interactions, it would improve clarity to discuss how membrane charge may contribute for these proteins.

We thank the reviewer for this suggestion. We have updated the discussion section of the revised manuscript accordingly.

3.) In contrast to positively charged peptides, the authors previously found that two more neutral anchoring signals (Lyn and Palm/Pre) localize homogeneously. Including data (or at least references) for one or both of these would be an important contrast to clarify that the rear-localization is not a generic property of lipid-anchored proteins.

We have mentioned this point and included the reference in the revised version of manuscript.

4.) For the experiments such as those included in Figures 1 and 2, a cross-correlation analysis would be very helpful. It is clear that the localization of PIP3 is anticorrelated with that of other proteins, but is there a temporal lag?

As per the reviewer's suggestion, we have performed the cross-correlation analysis for all our lipid-anchored or peripheral back proteins with respect to PIP3 profiles, to determine whether there is any time-lag. Please see the representative data below for four cells. First two cells (A and B) were co-expressing PTEN and PH_{crac} (PIP3 sensor). Last two cells (C and D) were co-expressing PKBR1 and PH_{crac}. As it is clear from the plots, we did not find any temporal lag here. In fact, we did not observe any temporal lag in cross-correlation value for any of our back proteins. We would like to clarify here that our data and analysis does not rule out the existence of time lags between the localization of different proteins. However, it is demonstrating that the temporal lag is not detectable at the rate we have acquired images (i.e., 7 sec/frame).

Figure 1. Pearson's r (top panel heatmap, in “parula” colormap) and cross-correlation vs. time-lag plots (bottom panels). Mean \pm SD are shown in bottom panels.

5.) On page 7, for the Latrunculin experiments, the authors state “When periodic circulating waves were induced in these cells”. How were waves induced?

We pre-treated the cells with 4mM caffeine before starting the experiment. It was demonstrated in earlier literature that such treatment, even in the absence of cytoskeleton dynamics, lowers the threshold of the signaling network in *Dictyostelium* cells and hence more symmetry breaking events are observed during the experiment (Arai Y. et al., *Proc. Natl. Acad. Sci. U.S.A.* 2010, 107(27):12399–12404; Matsuoka, S et al., *Nat. Commun.* 2018, 9(1):4481; Fukushima S. et al, *J. Cell Sci.* 2019, 132(5), jcs224121). We mentioned this point inside the Methods section of our manuscript.

6.) In Fig 3A-3H and Fig S5, the authors do not specify in the main text that global cAMP stimulation experiments were also performed in LatA treated cells. There is a brief mention of this in the Methods section but this seems like a detail that should also be provided in the main text.

We have mentioned this point inside the main text of the revised manuscript.

7.) Though mentioned in the methods section, the authors should state in the main text that giant, electrofused cells were used for photoconversion and ventral wave experiments.

We have included this point inside the main text of the revised manuscript.

8.) This is a small point, but in Figure 7B and the associated text, the authors state that cAR1 was induced to localize to back-state regions. In fact, the localization of the R+ peptide was measured. Presumably, this also reflects cAR1, but that was not measured directly. The text could be reworded to be more precise.

We have updated the figure legend. In the main text, we have also mentioned that we have used recruited R+ profile as a proxy for cAR1 localization.

9.) While movies were provided for most imaging experiments, they were not provided for the HL60 optogenetic experiments. Adding movies for the HL60 experiments would be helpful.

We have included movies of HL-60 as well as RAW 264.7 experiments (Videos S6-S7, Video S13, and Videos S22-S23).

10.) How do the authors reconcile their observation that Gbeta is asymmetrically localized to cell rear regions with their group's prior publication (Jin et al 2000, cited in the discussion) that Gbeta is localized with a polarization towards the cell front?

Another reviewer also mentioned this point. We are quoting our response below:

“It is true that our current data contradicts our group’s old data (*Jin T. et al., Science 2000, 287(5455):1034-1036*) regarding G β γ localization. We believe that can be attributed to the fact that those cells were imaged using epifluorescence microscopes, almost twenty-five years ago, in the early days of live-cell imaging when optical resolution and sensitivity were severely limited (compared to today’s standard confocal images). Additionally, unlike the ventral wave images which contain no membrane folds (*Gerisch G. et al., Biophys J. 2004, 87(5):3493-503; Gerisch G. et al., Biophys J. 2009, 96(7):2888-2900*), the spatial gradients observed during protrusion formation, as it is usually recorded a few micrometer above the substrate-attached surface, sometimes appear weak. The data we present here using multimodal imaging techniques in different physiological scenarios reveal a more accurate picture of localization dynamics.”

11.) Related to major point 4, on page 19 the authors refer to the “self-organizing dynamic partitioning events in plasma membrane”. The dynamic partitioning mechanism described does not appear to be self-organizing, but it could contribute to self-organizing signaling domains.

As per the reviewer’s suggestion, we have removed the “self-organizing” part from that sentence.

12.) There were a number of small typo or grammatical errors that should be corrected.

We have corrected multiple typos and grammatical errors in the manuscript.

Reviewer #3 (Remarks to the Author):

Understanding sorting mechanisms of membrane proteins is central to many cellular processes. The current paper describes a possible new mechanism based on differences in diffusional dynamics of lipid anchored proteins between different regions in the plasma membrane which arise during formation and propagation of protrusions. A large array of experimental data is provided, which indeed indicate the presence of a sorting potential for this class of proteins. It is then argued that existing mechanisms such as the involvement of the actin filament network or shuttling of proteins between the membrane and cytosol cannot explain the observed partitioning behaviour, giving room to a novel type of mechanism. Stochastic simulations are then used to provide evidence for a novel type of sorting mechanism, coined dynamic partitioning. However, before possibly accepting this mechanism, I have two major concerns.

We thank the Reviewer #3 for the comments on our work. Please find our reply to specific points below.

First of all, it is not clear to me why a thermodynamic-based mechanism (instead of a dynamic one) cannot be invoked to explain the results. As acknowledged by the authors themselves, the back and front regions of the plasma membrane have very distinct compositions, both in terms of protein content and considering lipid types. It is also well known that proteins can preferentially sort to different membrane regions based on their thermodynamic preference, which could arise from global membrane properties as well as from detailed protein-lipid interactions. The RAFT hypothesis is based on this principle. Although it remains unclear to date how RAFT domains are precisely organized in vivo, membrane heterogeneity as an organizational principle is well established. Remarkably, the concept of RAFTS is only mentioned at the very end of the paper. If lipidated proteins simply sort to the back region of the membrane due to more favourable interactions with the lipid population in this region, this would provide a direct driving force for the observed partitioning behaviour. Note that RAFT-type of domains could couple to curvature gradients that might exist in the protruding front end, providing a driving force for RAFT coalescence and sorting itself.

As we understand it, the reviewer is asking whether the distribution of lipid rafts in the plasma membrane can account for the compartmentalization of all the signaling and cytoskeletal molecules that we are describing here. While we do not completely discount the contribution from raft formation in our dynamic partitioning mechanism, our data suggests that the rafts are not the main regulator.

First, in *Dictyostelium* cells, the plasma membrane surface receptor cAR1 is completely localized in detergent resistant membrane fractions and thereby should be regarded as an ideal marker of lipid rafts (Xiao Z. *et al.*, *Mol Biol Cell*. 1997, 8(5):855-69). However, cAR1 remains totally uniformly distributed on the membrane and does not exhibit any specific spatiotemporal dynamics, even in highly polarized cells (Xiao Z. *et al.*, *J Cell Biol*. 1997, Oct 20;139(2):365-74; Jin T. *et al.*, *Science* 2000, 287(5455):1034-1036; Ueda M. *et al.*, *Science* 2001, 294(5543): 864-867). Together, this raised question whether rafts served any purpose in organizing compartmentalization on membrane (at least in the timescale of signaling cascade driven polarization).

Second, although there are different opinions, literature generally suggests that rafts size is limited under 300 nm (Pike L.J., *J Lipid Res*. 2003, 44(4):655-67; Hancock J.F., *Nat Rev Mol Cell Biol*. 2006,7(6):456-462; Sezgin E. *et al.*, *Nat Rev Mol Cell Biol*. 2017, 18(6):361-374; Lingwood D. *et al.*, *Science*. 2010, 327(5961):46-50), whereas our domain structures are much larger, often spreading over tens of square micrometers. While rafts can still exist and contribute to the dynamic partitioning driven compartmentalization process, they have to be organized to coordinate across large micron scale domains and have to be disassemble quickly as the domain switches states.

Third, while the reviewer is right that raft domains can couple to curvature to organize membrane domains during protrusion formation, we would like to point out that, as we have shown in the manuscript, we can see dynamic compartmentalization of lipid-anchored and integral membrane proteins very well, even during the ventral wave propagation at the substrate-attached surface of the cell. Moreover, it was demonstrated earlier that such ventral wave propagation is independent of any protrusion formation or membrane folds (Gerisch G. *et al.*,

Biophys J. 2004, 87(5):3493-503; Gerisch G. et al., Biophys J. 2009, 96(7):2888-2900; and Figure S5D, S5F of this manuscript). Hence, we can rule out the possibility of curvature gradient driven raft organization in our partitioning mechanism.

Finally, it was shown earlier that the raft-like nanocluster formation for lipid-anchored proteins on the membrane, unlike classical rafts, violates the law of mass action and such nanocluster formation is actually maintained away from equilibrium thermodynamic phase (*Goswami D. et al., Cell 2008, 135(6):1085-1097; Sharma P. et al., Cell. 2004, 116(4): 577-589; Plowman SJ et al., Proc Natl Acad Sci U S A. 2005;102(43):15500-15505*). It was further demonstrated that actin remodeling is required for larger-scale spatiotemporal organization of such raft-like nanoclusters (*Goswami D. et al., Cell 2008; Gowrishankar K, et al., Cell. 2012 Jun 8;149(6):1353-67; Lingwood D. et al., Science. 2010, 327(5961):46-50*). Since our paper primarily deals with lipid-anchored proteins whose compartmentalization is actin cytoskeleton independent and the domains are spread across even larger length-scale, we speculated that these do not stem simply from raft-like structures. The definition of “raft” is always evolving and hence it is impossible to completely rule out the possibility of the contribution of rafts in our partitioning mechanism. However, if rafts do contribute, they would need to be coordinated so that they spread across multiple square micrometers, assembling then disassembling quickly as soon as wave passes, and then reassembling spatially again as soon as signaling cascade turns on again.

My second concern is about the stochastic simulations, which I find hard to understand the way it is described in the current paper. For instance, it is unclear what the 'refractory' state is, as it was not mentioned before in the manuscript. It is also unclear why we need an 'excitable system' (what is being excited here?), where the 'mutually inhibitory action' comes from (how and why are the F and B states inhibiting each other?), and what the 'delayed negative feedback' represents (why/how do F and R state provide negative feedback?). Figure 6A is not very helpful to understand what went into the modelling approach (and why!). Things become even more confusing when reading the Method section where one reads sentences such as 'Basal and R-mediated inactivation of F-molecules' and 'non-linear B-dependent activations and inhibitions' (what are F-molecules? Why are they basal-inactivated ? Why non-linear ? What does all of this

even mean in the context of the rest of the paper ?). Without a clear rationale of invoking these terms, one is left with the impression that a lot of fiddling was done to get behaviour consistent with the desired outcome.

We apologize for the confusion regarding the description of the stochastic simulations. These equations describe an “excitable system” which is well-studied class of dynamical system that first arose in Hodgkin’s and Huxley’s seminal studies of action potentials in the 1950s (*Hodgkin A.L., Huxley A.F., "A quantitative description of membrane current and its application to conduction and excitation in nerve", J. Physiology., 1952, 117 (4): 500–44*) and gained traction with the dynamical descriptions of activator-inhibitor systems following the work of FitzHugh and Nagumo et al. (*FitzHugh R., Bull. Math. Biophysics, 1955, 17:257-78; FitzHugh R., Biophysical J., 1961, 1:445–466; Nagumo J. et al., 1962, Proc. IRE, 50(10):2061-2070*).

Excitable systems are nonlinear systems that have a single stable equilibrium. When the system is at this equilibrium it is said to be quiescent. In this case, the system response to stimuli shows quite different behaviors depending on the size of that perturbation. Whereas subthreshold perturbations lead to small responses, larger supra-threshold perturbations lead to large scale deviation before the system returns to its stable equilibrium. During these large-scale deviations, often referred to as “firings,” the system is said to be excited, and displays a stereotypical behavior. Following this excitable state, the system reaches a refractory period whereby it is unable to fire again until it returns to that basal level. In systems where the interacting species can diffuse, these firings propagate in characteristic waves. That the systems regulating cell motility shows excitable behavior is now well established experimentally, and numerous models by both our group and others have been proposed to describe it. In recent years, the biochemical excitable networks, in fact, were shown to be able to correctly describe and predict the behaviors of different physiological and developmental processes, such as random amoeboid migration, chemotaxis, electrotaxis, macropinocytosis, cytokinesis, somitogenesis, bone regeneration, oncogenic transformation, etc., in various cells and organisms (*Huang, C. H. et al., Nat Cell Biol. 2013;15(11):1307-1316; Xiong Y. et al., Proc Natl Acad Sci U S A. 2010;107(40):17079-17086; Weiner O.D. et al., PLoS Biol. 2007, 5(9):e221; Miao Y. et al., Nat. Cell Biol. 2017, 19(4), 329-340; Miao Y. et al, Mol Syst Biol. 2019, 15(3): e8585; Banerjee T. et al, Nat. Cell Biol. 2022, 24(10), 1499-1515; Yang H.W. et al., Nat. Cell Biol. 2016, 18(2):191-201; Yang Q. et al., eLife*

2022, 11: e73198; Bement W.M. et al., *Nat Cell Biol.* 2015, 17(11):1471-83; De Simone A et al., *Nature* 2021, 590(7844):129-133; Zhan H. et al., *Dev Cell.* 2020, 54(5):608-623.e5; Biswas D. et al., *PLOS Comput. Biol.* 2021, 17(7): e1008803; Hubaud A. et al., *Cell* 2017, 171(3):668-682.e11; Devreotes P.N. et al., *Annu Rev Cell Dev Biol.* 2017, 33:103-125). The model that we use to describe this excitable behavior in this manuscript follows these earlier works.

Nevertheless, we have expanded on our description of the model to make the manuscript more self-contained (please see subsection A. in *Computational modelling* section under Methods for the details). In particular, we have moved away from describing the three species F, B and R as membrane “states” but rather refer to them as species. Note, however, that the relative concentrations of these species specify regions of the membrane. In particular, the membrane partition into regions with distinct F, B and R profiles: it is quiescent when B is high, but F and R low; excited when F is high, but B and R low; and refractory when B and R are high, but F is low.

Furthermore, it remains puzzling (and unexplained) what the 'bound' and 'unbound' states of the lipid anchored proteins (LPs) are - it was argued before that these proteins actually do NOT unbind, due to the strong association of the anchor with the membrane. Could it be that in fact these LPs still can unbind, even though at a much reduced rate compared to peripherally bound proteins (PPs), and that this fraction accounts for the observed effects (being similar to how PPs partition) ? The difference between diffusion constants of the bound and unbound states is really large (40-fold), and cannot be easily explained by assuming a different membrane environment. The fact that, according to the data in Table S3, the diffusion constants for the bound and unbound states are the same for LPs and PPs supports this notion.

Our model assumes that the lipid-anchored protein, LP, and the peripheral membrane protein, PP, molecules are different. LP molecules bind and unbind to other (unmodeled) membrane-bound proteins with the binding mediated by the “Back” species, B. Thus, we have:
 $LP_u + B \rightarrow LP_b + B$; $LP_b \rightarrow LP_u$. We hypothesize that the unbound molecule (LP_u) can move freely on the membrane and thus assign to it the faster of the two diffusion coefficients ($D_{slow} =$

0.0215 $\mu\text{m}^2/\text{s}$). In contrast, the bound molecule (LP_b) is anchored which severely slows down the diffusion, which we assign as ($D_{\text{fast}} = 0.416 \mu\text{m}^2/\text{s}$). Our use of word “bound” and “unbound” possibly created some confusion – we would like to clarify here (we have also added this clarification in the manuscript) that both LP_u and LP_b molecules are restricted to remain on the membrane (LP_u can diffuse freely only on membrane and cannot move to cytosol, as our photoconversion assay and global receptor assay has showed). In contrast, PP molecules exist in three states. They can be cytosolic (PP_c), membrane-unbound (PP_u) and membrane-bound (PP_b). For the last two cases, they are on the membrane, similarly to the LP molecules. Thus, we have the following binding/unbinding reactions: $\text{PP}_u + B \rightarrow \text{PP}_b + B$, $\text{PP}_b \rightarrow \text{PP}_u$, as well as two shuttling reactions from the cytosol to the unbound state and back: $\text{PP}_c \rightleftharpoons \text{PP}_u$. Diffusion coefficients for the membrane bound forms of PP are the same as those of LP (fast: PP_u ; slow PP_b) with even faster coefficients for the cytosolic component.

Note that this means that the *observable* diffusion coefficients are not determined exclusively by composition/state of the membrane per se. The *intrinsic* diffusion coefficients are considered to be the same in regions of the membrane that are predominantly back (B high, F low) or front (B low, F high), as noted by the reviewer. However, the observable coefficients are also determined by whether the molecules are bound (LP_b and PP_b) or not (LP_u and PP_u). Please note that the diffusion coefficients used in the model were taken from the measured experimental values (please see subsection B. of *Computational modelling* section under Methods for the details on how we computed these values from the single-molecule imaging experiments). We agree that there is a large difference in different diffusion coefficients, but this is what was seen experimentally, and that observation formed the basis of modelling.

REVIEWER COMMENTS

Reviewer #1 (Remarks to the Author):

The manuscript has been improved considerably. points 1-3 and 5 were addressed well. Fig S4 with localization of Gbg during chemotaxis will take away a lot of confusion. A movie of Fig S4A or B would be very useful.

I still have very serious concerns on model selection as was addressed in point 4. As mentioned in the previous review, point 4 has two parts: Part A model selection for experiments and Part B, model selection for computational modelling. The first part is important to resolve, but largely technical. The second part is much more fundamental as it comes close to how the model is defined.

A. Model selection for experiments.

95% CI and AIC were added. Thank you, but no information was given how it was calculated (n, method/equation used etc)

It is striking that the AIC values in table S2 are very large negative, AND that they have not yet obtained a minimal value.

Therefore AIC of higher order models must be calculated to find the minimum AIC value. I estimate from how the AIC values evolve at increasing number of diffusion states that the minimal AIC will be at 5 or 6 states in front and back. There are two possibilities: 5 or 6 is real, or -more likely- it is an artefact. An artefact with this very high number of states is easily obtained due to data oversampling as is suggested by the very large negative AIC values. Although the number of data points is not mentioned, it was easily deduced from the presented probabilities of Fig 5F in the excel sheet that must have been derived from integers, yielding 30,803 data points for back and 10,822 for front. This are many more data points than needed for model selection.

Why does oversampling lead to selection of too complicated models. There is not much good literature on this topic, so some explanation.

The background of model selection using AIC is that the process of interest is described by different models. Data are collected that should be a reasonable representation of these models, that is they cover the width and complexity of the models and resemble synthetic data sets drawn randomly from the models. The collected data set should not be larger than needed to represent the models, to prevent overfitting (see below). Then for each model the maximum likelihood is calculated and the models are compared using $AIC = 2k - 2\ln(L)$, where k is the number of parameters of the model and L is the likelihood. AIC values for models with increasing number of parameters (here more diffusion states) are calculated and the model with the lowest AIC value is the preferred model. The idea is that when the number of diffusion states increases the number of parameters is a penalty that must be overcome by a "significant" increase of the likelihood. The log-likelihood $\ln(L)$ is proportional to the number of data points; thus when the data set is heavily oversampled, it is very difficult for the $2k$ penalty to override $\ln(L)$, requiring very high values of k . A normal probability distribution is reasonably well represented by 50-100 data points, so a sum of three or four normal distributions can be sampled with 300 data points, indicating that the datasets used are oversampled up to 100-fold for back and 30-fold for front. As a

result the $\ln(L)$ values are far too high, and a too complex model is selected. This can be easily illustrated with the data for 3- and 4-state model in the back. It is deduced that the $\ln(L)$ for 3-state is 33,001 and 4-state is 33,140, a gain of 139 points that is not overcome by the penalty of 4 points for increasing with one state, and the 4-state model is better than 3-state model. If 300 data points were taken the $\ln(L)$ might be 330.0 for 3-state and 331.4 for 4-state, an increase of 1.4 points that is less than the penalty, and 4-state is NOT better than 3-state. It is generally concluded that in case of possible overfitting the model with minimal AIC value is the maximal model, and the "true" model is among the models investigated.

A convenient alternative to fitting the original data points for finding the optimal model is to use the data of the histogram of figure 5F. These data are frequencies and no longer sensitive to oversampling; actually this is the easiest way to do model selection. Model selection using the histogram data needs to be done anyway for finding a model for computational modelling (see below).

B. Model selection for computational modelling

As mentioned in my first report, it is understandable to reduce the number of diffusion components for computational modeling. Looking to the histogram of figure 5E, there are at least two components in BOTH front and back. Also the AIC values in table S2 unequivocally reveal for the front AT LEAST two diffusion states, also after applying the corrections described above. The authors do first a modality test (was it done on the original 30803 data points in back and 10822 data points in front, or on the data of the histogram of Fig 5E?), they find a test that suggests one modality for front, conclude that the front has only one state and then ignore completely the possibility provided by all the other statistical analysis that the front may have more states. The method chosen by the authors to come to ONLY one component is the front is not justified in the context of the rest of the manuscript. It belongs to the brute statement "for every desired outcome there is a statistical test". It does do no right to all the efforts by the experimentalists to describe the differences between diffusion in the front and back; that work convincingly demonstrates that if a minimal model is selected it has two states BOTH in front and back. In a computation model with two diffusion components the authors must incorporate a slow and fast diffusion state in both front and back. For convenience they can use the same diffusion constant in front and back. There are several possibilities how the slow component can appear in the front. It may not only bind to back but also (with much lower affinity) to front. It may be formed only in back but diffuses to front (where it dissociates).

Suggestions for a practical solution

table S2 add AIC values for more states till AIC values increase. Add comments in legend to table or in method section that models may have been overfitting due to oversampling of the data, indicating that the optimal model is among the models investigated.

Table S3 analyze histograms of Fig 5E with AIC for models with 1,2,3,4..... states till AIC values increase. Present the statistical data for two states (for computational modelling), and statistical data for the optimal model

In computational modelling use the outcome for 2-state model of front and back.

Reviewer #2 (Remarks to the Author):

In their revised manuscript, Banerjee et al. address reviewer comments to support their characterization of a “dynamic partitioning” mechanism that can polarize the localization of membrane proteins. In general, the authors addressed my major concerns.

In particular, in their new Figure S14 (E,F) they show that a normally uniform Lyn11 domain can polarize upon optogenetic recruitment of R+, even in cells pretreated with ROCK inhibitor (Y-27632). This strengthens their argument that polarization is occurring independent of rearward membrane flows in HL60 leukocytes. The authors also added experiments in RAW 264.7 macrophages that indicate that the mechanism can also cause spatial patterning of lipid-anchored membrane proteins in these cells. Lastly, the authors effectively adjusted their text to be precise in describing the polarization of protein localizations, to discuss the potential role of membrane charge, and to relate dynamic partitioning to other characterized mechanisms.

I have only one remaining concern, which is that the data for HL-60 cells only includes representative images for 1-2 cells per experiment. Quantification including all cells analyzed, similar to that in Figure 7F, should be added to support the conclusions from these experiments. I understand that these experiments did not have a second marker for reference such as LimE, but the protruding region of the membrane could be determined from successive frames, and this could be used for a Front vs Back comparison.

I am also including here a few small constructive comments for the final version:

1.) In the abstract (3rd sentence), "the signaling network" is unclear. While it has a clear meaning for Dictyostelium chemotaxis, it might be confusing for a broader audience. Something like "a polarity signaling network" might be better, or even just "a signaling network".

2.) There are still a number of small grammatical mistakes throughout. A couple of examples are: in a few places, "cytoskeleton dynamics" should be "cytoskeletal dynamics"
Page 9 middle, "where signal transduction network is activated" should be "where the front-state signal transduction network is activated" or similar.

3.) In Figure S7A, for the 0:00 timepoint, it looks like the PIP3 image is a duplicate of the R-Pre image. The wrong image may have been inserted here by mistake.

Reviewer #3 (Remarks to the Author):

My concerns have been adequately addressed in the rebuttal letter, and the manuscript has been improved accordingly. I am happy to recommend publication of this interesting work.

We would like to thank the reviewers once again for their time and for providing constructive and overall positive feedback. The reviewers' comments and our responses (in red) are provided below.

REVIEWER COMMENTS

Reviewer #1 (Remarks to the Author):

The manuscript has been improved considerably. points 1-3 and 5 were addressed well. Fig S4 with localization of Gbg during chemotaxis will take away a lot of confusion. A movie of Fig S4A or B would be very useful.

We would like to thank the Reviewer #1 for appreciating the overall strength of our revised manuscript. As per the reviewer's suggestion, we have included a movie of Fig. S4A in the current version of our manuscript (Video S5).

I still have very serious concerns on model selection as was addressed in point 4. As mentioned in the previous review, point 4 has two parts: Part A model selection for experiments and Part B, model selection for computational modelling. The first part is important to resolve, but largely technical. The second part is much more fundamental as it comes close to how the model is defined.

A. Model selection for experiments.

95% CI and AIC were added. Thank you, but no information was given how it was calculated (n, method/equation used etc)

We have included all these details in the current version of the manuscript (Table S2 legend).

It is striking that the AIC values in table S2 are very large negative, AND that they have not yet obtained a minimal value.

Therefore AIC of higher order models must be calculated to find the minimum AIC value. I estimate from how the AIC values evolve at increasing number of diffusion states that the minimal AIC will be at 5 or 6 states in front and back. There are two possibilities: 5 or 6 is real, or -more likely- it is an artefact. An artefact with this very high number of states is easily obtained due to data oversampling as is suggested by the very large negative AIC values. Although the number of data points is not mentioned, it was easily deduced from the presented probabilities of Fig 5F in the excel sheet that must have been derived from integers, yielding 30,803 data points for back and 10,822 for front. This are many more data points than needed for model selection.

Why does oversampling lead to selection of too complicated models. There is not much good literature on this topic, so some explanation.

The background of model selection using AIC is that the process of interest is described by different models. Data are collected that should be a reasonable representation of these models, that is they cover the width and complexity of the models and resemble synthetic data sets drawn randomly from the models. The collected data set should not be larger than needed to represent the models, to prevent overfitting (see below). Then for each model the maximum likelihood is calculated and the models are compared using $AIC = 2k - 2\ln(L)$, where k is the number of parameters of the model and L is the likelihood. AIC values for models with increasing number of parameters (here more diffusion states) are calculated and the model with the lowest AIC value is the preferred model. The idea is that when the number of diffusion states increases the number of parameters is a penalty that must be overcome by a “significant” increase of the likelihood. The log-likelihood $\ln(L)$ is proportional to the number of data points; thus when the data set is heavily oversampled, it is very difficult for the $2k$ penalty to override $\ln(L)$, requiring very high values of k . A normal probability distribution is reasonably well represented by 50-100 data points, so a sum of three or four normal distributions can be sampled with 300 data points, indicating that the datasets used are oversampled up to 100-fold for back and 30-fold for front. As a result the $\ln(L)$ values are far too high, and a too complex model is selected. This can be easily illustrated with the data for 3- and 4-state model in the back. It is deduced that the $\ln(L)$ for 3-state is 33,001 and 4-state is 33,140, a gain of 139 points that is not overcome by the penalty of 4 points for increasing with one state, and the 4-state model is better than 3-state

model. If 300 data points were taken the $\ln(L)$ might be 330.0 for 3-state and 331.4 for 4-state, an increase of 1.4 points that is less than the penalty, and 4-state is NOT better than 3-state. It is generally concluded that in case of possible overfitting the model with minimal AIC value is the maximal model, and the “true” model is among the models investigated.

A convenient alternative to fitting the original data points for finding the optimal model is to use the data of the histogram of figure 5F. These data are frequencies and no longer sensitive to oversampling; actually this is the easiest way to do model selection. Model selection using the histogram data needs to be done anyway for finding a model for computational modelling (see below).

B. Model selection for computational modelling

As mentioned in my first report, it is understandable to reduce the number of diffusion components for computational modeling. Looking to the histogram of figure 5E, there are at least two components in BOTH front and back. Also the AIC values in table S2 unequivocally reveal for the front AT LEAST two diffusion states, also after applying the corrections described above. The authors do first a modality test (was it done on the original 30803 data points in back and 10822 data points in front, or on the data of the histogram of Fig 5E?), they find a test that suggests one modality for front, conclude that the front has only one state and then ignore completely the possibility provided by all the other statistical analysis that the front may have more states. The method chosen by the authors to come to ONLY one component is the front is not justified in the context of the rest of the manuscript. It belongs to the brute statement “for every desired outcome there is a statistical test”. It does do no right to all the efforts by the experimentalists to describe the differences between diffusion in the front and back; that work convincingly demonstrates that if a minimal model is selected it has two states BOTH in front and back.

In a computation model with two diffusion components the authors must incorporate a slow and fast diffusion state in both front and back. For convenience they can use the same diffusion constant in front and back. There are several possibilities how the slow component can appear in

the front. It may not only bind to back but also (with much lower affinity) to front. It may be formed only in back but diffuses to front (where it dissociates).

We thank the reviewer for this detailed comment. To address these issues, we have incorporated multiple changes which we have described below.

Suggestions for a practical solution

Table S2 add AIC values for more states till AIC values increase. Add comments in legend to table or in method section that models may have been overfitting due to oversampling of the data, indicating that the optimal model is among the models investigated.

As per the reviewer's suggestion, we have calculated AIC values assuming 5 or 6 states and confirmed that the AIC value that obtained by assuming 4 states is minimum among those of 6 models irrespective of the membrane regions (Table S2). We have also found that even without incorporating penalty terms, that is even if we focus on the values of the maximum log-likelihood alone, it is maximum when 4 states were assumed. We have not explicitly mentioned that the model was overfitted since, as the reviewer pointed out, it is difficult to accurately assess the impact of oversampling on AIC calculations. We, however, have definitively mentioned in the methods section that more than four diffusion coefficients are not required to model the data in either front or back state and optimal model is among the models investigated.

Table S3 analyze histograms of Fig 5E with AIC for models with 1,2,3,4..... states till AIC values increase. Present the statistical data for two states (for computational modelling), and statistical data for the optimal model

As per the reviewer's suggestion, we have fitted Gaussian mixture models with varying number of components (1 to 5) to the histogram data of short-range diffusion coefficients (we combined back and front histogram data from Figure 5F for fitting, since that would yield same diffusion coefficients in back and front regions, as the reviewer suggested above). As it is clear from the updated Table S3, both AIC and corrected AIC (AICc) values are smallest when we considered three diffusion states, after which it started increasing again. Since the $\Delta = AIC_{c2-comp} - AIC_{c3-comp}$ change is comparatively smaller and since the two faster diffusion coefficients in the three-component model differ by much less than an order of magnitude, we decided to

implement a simplified 2-state diffusion model for both front and back region associated molecules in the simulations. After fitting data to obtain the slow and fast diffusion coefficients, we further computed their relative proportions (separately in front and back regions) which essentially demonstrate the proportion of bound and unbound forms, respectively. When we segmented the simulation results and obtained the bound:unbound fractions in back and front regions, we were able to successfully obtain the same values. A graphical summary of our approach is provided here. Please see the revised subsection B of the *Computational modelling* section, Table S3, and Table S4 for additional details on the model development.

Graphical Summary

Simulated Model

F:X = front-bound, slow-moving fraction of X on membrane
 B:X = back-bound, slow-moving fraction of X on membrane
 u = unbound, rapidly diffusing fraction on membrane
 c = cytosolic fraction

In computational modelling use the outcome for 2-state model of front and back.

As per the reviewer's suggestion, we have extensively revised our model to incorporate 2-state diffusion model in both front and back regions (Figure 6A). The results from our revised model have demonstrated that the dynamic partitioning of lipid anchored proteins and the recurrent shutting of peripheral proteins can generate familiar propagating wave patterns, as our original model was able to generate. As mentioned above, the details of the changes are available in the revised subsection B of the *Computational modelling* section, Table S3, and Table S4. The new simulation results are provided in Figure 6, Figure S12, Figure S13, Video S20, and Video S21.

Reviewer #2 (Remarks to the Author):

In their revised manuscript, Banerjee et al. address reviewer comments to support their characterization of a “dynamic partitioning” mechanism that can polarize the localization of membrane proteins. In general, the authors addressed my major concerns.

In particular, in their new Figure S14 (E,F) they show that a normally uniform Lyn11 domain can polarize upon optogenetic recruitment of R⁺, even in cells pretreated with ROCK inhibitor (Y-27632). This strengthens their argument that polarization is occurring independent of rearward membrane flows in HL60 leukocytes. The authors also added experiments in RAW 264.7 macrophages that indicate that the mechanism can also cause spatial patterning of lipid-anchored membrane proteins in these cells. Lastly, the authors effectively adjusted their text to be precise in describing the polarization of protein localizations, to discuss the potential role of membrane charge, and to relate dynamic partitioning to other characterized mechanisms.

We sincerely appreciate the positive comments from the Reviewer #2.

I have only one remaining concern, which is that the data for HL-60 cells only includes representative images for 1-2 cells per experiment. Quantification including all cells analyzed, similar to that in Figure 7F, should be added to support the conclusions from these experiments. I understand that these experiments did not have a second marker for reference such as LimE, but the protruding region of the membrane could be determined from successive frames, and this could be used for a Front vs Back comparison.

As per the reviewer’s suggestion, we have performed this quantification and added a front vs back intensity ratio plot for HL-60 experiments (Figure S14I).

I am also including here a few small constructive comments for the final version:

1.) In the abstract (3rd sentence), "the signaling network" is unclear. While it has a clear meaning for Dictyostelium chemotaxis, it might be confusing for a broader audience. Something like "a polarity signaling network" might be better, or even just "a signaling network".

We thank the reviewer for this suggestion. We have now clearly mentioned that it is “the Ras/PI3K/Akt/F-actin network”.

2.) There are still a number of small grammatical mistakes throughout. A couple of examples are:

in a few places, "cytoskeleton dynamics" should be "cytoskeletal dynamics"

Page 9 middle, "where signal transduction network is activated" should be "where the front-state signal transduction network is activated" or similar.

We thank the reviewer for these suggestions. We have corrected all these issues.

3.) In Figure S7A, for the 0:00 timepoint, it looks like the PIP3 image is a duplicate of the R-Pre image. The wrong image may have been inserted here by mistake.

We thank the reviewer for pointing this out. We have now corrected this.

Reviewer #3 (Remarks to the Author):

My concerns have been adequately addressed in the rebuttal letter, and the manuscript has been improved accordingly. I am happy to recommend publication of this interesting work.

We would like to sincerely thank the Reviewer #3 for the enthusiastic and positive response to our revised manuscript.

REVIEWERS' COMMENTS

Reviewer #1 (Remarks to the Author):

The manuscript has been improved considerably, with a balanced use of model selection for experiments and computational analysis. I am happy to recommend publication of this interesting work.

We would like to take this opportunity to thank the reviewers once again for their valuable comments. The reviewer's comments and our response (in red) are provided below.

REVIEWER COMMENTS

Reviewer #1 (Remarks to the Author):

The manuscript has been improved considerably, with a balanced use of model selection for experiments and computational analysis. I am happy to recommend publication of this interesting work.

We sincerely thank the Reviewer #1 for enthusiastic and positive response to our revised manuscript.